# Rapid concerted switching of the neural code in the inferotemporal cortex

Yuelin Shi[1,2]✉, Dasheng Bi[2], Janis K. Hesse[2], Frank F. Lanfranchi[1,2], Shi Chen[2] & Doris Y. Tsao[2,3]✉

A fundamental paradigm in neuroscience is that neurons represent the world through fixed tuning functions, with stable mappings from stimulus features to firing rates[1]. Here, we report that tuning can instead shift rapidly and coherently across a neural population, enabling a dynamic transition from detecting a broad category to discriminating individual exemplars. We set out to address a longstanding debate in visual neuroscience about whether the inferotemporal cortex uses a specialized code for specific object categories or a general-purpose code that applies to all objects. We found that face-selective cells in macaque inferotemporal cortex initially adopted a general code optimized for face detection. However, after a rapid concerted population event lasting less than 20 ms, the neural code transformed into a face-specific one, with two striking features: response gradients to principal detection-related dimensions reversed direction, and new tuning emerged for multiple higher-dimensional features that support fine face discrimination. These dynamics in face patches were specific to face stimuli and did not occur in response to non-face objects. Thus, for faces, face cells transition from detection to discrimination by switching from an object-general code to a face-specific one. More broadly, our findings indicate that there is a previously unknown mechanism for neural representation: concerted stimulus-dependent switching of the neural code used by a cortical area.

An important challenge in visual neuroscience is to understand how populations of neurons encode visual stimuli. In the past decade, considerable progress on this question has been made in macaque inferotemporal cortex[2–6], a large brain region dedicated to high-level object recognition[7–9]. Anatomically, the inferotemporal cortex is organized into parallel networks, each specialized for encoding distinct aspects of object shape and colour[6,10–13]. Among these networks, the macaque face patch system has become a powerful model for testing competing theories of high-level object representation[14–16]. Here, we leverage this system to address a central debate in the field: whether the inferotemporal cortex relies on specialized mechanisms for processing specific object categories[17–19], or whether it relies on general-purpose mechanisms that process all object types in the same way[20–22].

According to one theory, face processing is 'domain specific', being reliant on specialized mechanisms dedicated to faces[17,23]. This idea was motivated by the observation that humans are best at discriminating face parts presented in the context of a whole, upright face[24,25] (Fig. 1a, top-right inset). This observation led to the suggestion that a face-detection gate—a switch determining whether an image is a face or not—is implemented before detailed feature processing begins[23] (Fig. 1a, top). The selectivity of face patches for faces[15,22,26,27], the exquisite sensitivity of cells in face patches to facial geometry and texture[3,14,28], and the striking perceptual distortions of faces but not non-face objects induced by face-patch stimulation[29] all point to a special role for face patches in representing faces.

However, a competing theory proposes that the computations performed by face patches are domain general rather than face specific[20,22]. Supporting this view, several studies have shown that general-purpose deep neural networks (DNNs) trained on object categorization effectively model neural responses across the inferotemporal cortex, including in face patches[2,6,22,30–32] (Fig. 1a, bottom). Specifically, inferotemporal cell responses can be modelled as linear combinations of features represented in late layers of these networks: $r = \mathbf{c} \cdot \mathbf{f}$, where $r$ is the cell's response, $\mathbf{f}$ is a feature vector representing the stimulus in the DNN space, and $\mathbf{c}$ defines the cell's 'preferred axis'[2,6,22,30] (Fig. 1b). According to this picture, the DNN-derived feature space forms a general object space in which all stimuli, including faces, reside (Fig. 1c). A neuron's preferred axis can be thought of as the direction in this space pointing from stimuli that elicit weak responses to those that elicit strong responses. In this framework, face cells encode stimuli, just like other inferotemporal cells, by projecting them onto their preferred axes, regardless of object category[6,22]. Consistent with this domain-general view, the inferotemporal cortex has been shown to contain a topographic map of object space defined by the top two principal components of DNN space[6,31] (Fig. 1c). Face patches occupy one quadrant of this map, alongside other category-selective regions, indicating that face cells participate in a shared object-coding framework.

According to the domain-general theory of face-patch coding, each cell should use a single preferred axis, and mapping with either faces or non-face objects should yield the same axis[6,22] (Fig. 1d, top). By contrast,

[1]Division of Biology and Biological Engineering, Caltech, Pasadena, CA, USA. [2]Department of Molecular and Cell Biology and Helen Wills Neuroscience Institute, University of California, Berkeley, Berkeley, CA, USA. [3]Howard Hughes Medical Institute, University of California, Berkeley, Berkeley, CA, USA. ✉e-mail: ysshi@caltech.edu; tsao.doris@gmail.com

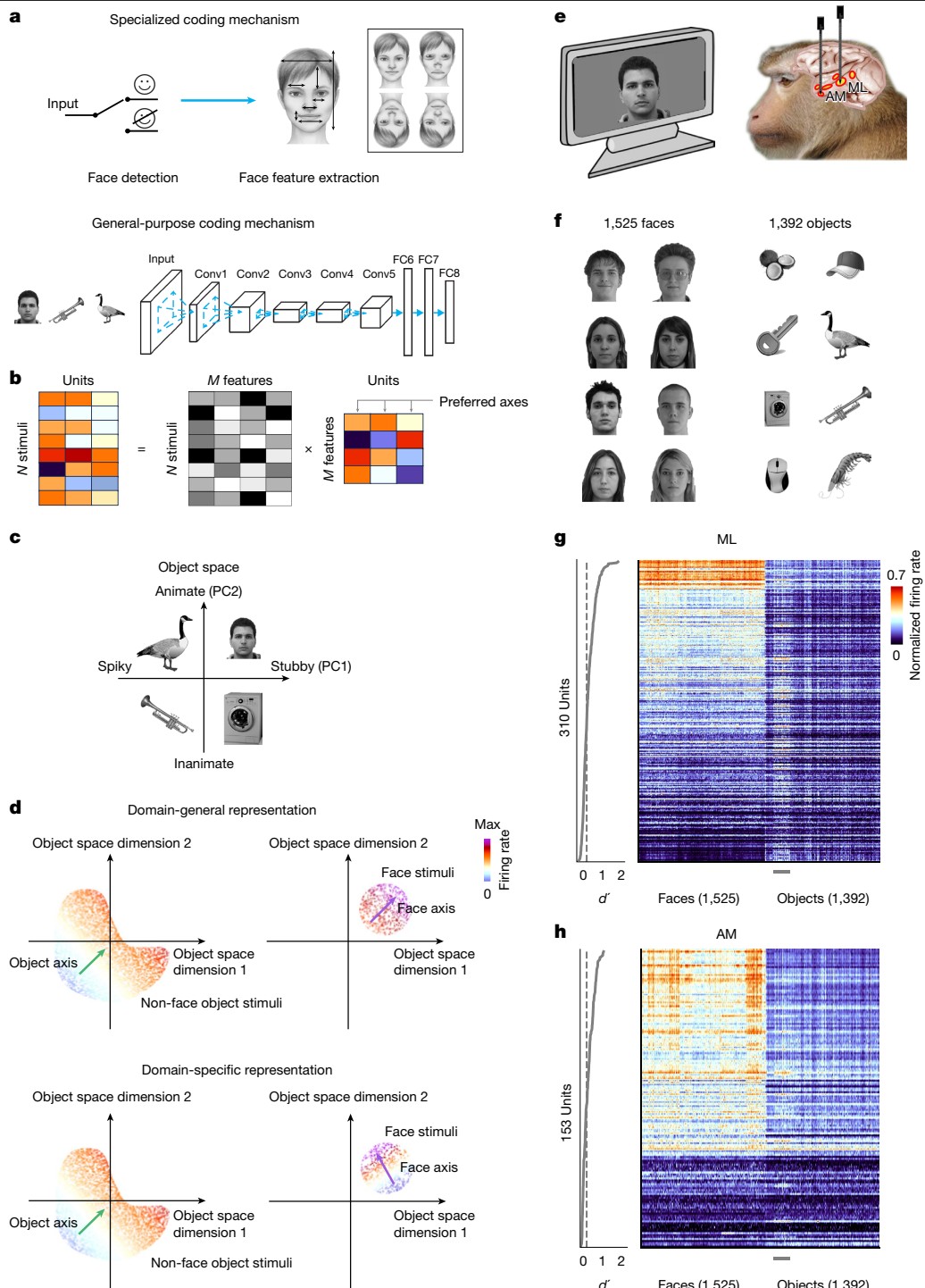

**Fig. 1 | Two contrasting models for face processing. a**, Top, a specialized coding mechanism. Extraction of detailed face features is preceded by a face-detection gate, ensuring that only upright faces undergo detailed processing. Top right, face features are perceived much better when embedded in an upright face[24]. Bottom, a general-purpose coding mechanism. Face cells are modelled by units in a late layer of a DNN trained on general object classification. Conv, convolutional layers; FC, fully-connected layers. **b**, The response of a unit to a stimulus is modelled by the projection of the features of the stimulus onto the unit's preferred axis. $N$ denotes the number of stimuli, and $M$ denotes the number of features. **c**, A 2D general object space can be computed by PCA of a DNN representation of faces and objects[6]. **d**, Schematic illustrating the predictions of domain-general (top) and domain-specific (bottom) models of face processing. Left, each dot corresponds to the projection of one object image onto the first two dimensions of object space, and the colour of the dot indicates the firing rate of a cell elicited by the image. The green arrow is the cell's object axis. Right,

each dot represents the projection of one face image onto the first two dimensions of the same object space. The purple arrow indicates the cell's face axis. If cells use a domain-general mechanism (top), the face and object axes should align. If cells use a domain-specific mechanism (bottom), the face and object axes should differ. **e**, Schematic of the experiment. **f**, In the main experiment, a large set of human faces (left) and non-face objects (right) were presented. **g**, Right, mean responses of all visually responsive cells in ML ($n = 310$). The small grey horizontal bar at the bottom indicates images of animals with greyed-out heads. Left, face selectivity $d'$ for each cell. The dotted vertical line marks $d' = 0.2$. **h**, As in **g** but for AM cells ($n = 153$). Credits: object images in **a**, **c** and **f** are from ref. 6, Springer Nature Limited; face images in **a**, **c**, **e** and **f** are reproduced from the FEI database (https://fei.edu.br/-cet/facedatabase.html)[47]; face images in **f** are reproduced from the CVL Face Database (http://lrv.fri. uni-lj.si/facedb.html)[48].

the domain-specific view posits that fine feature discrimination is selectively engaged for faces, so mapping with faces should produce a unique axis for each cell supporting fine-grained face identification (Fig. 1d, bottom).

To investigate these competing theories, we recorded responses of face cells in macaque face patches to a large set of faces and non-face objects. We parameterized the stimuli using layer fc6 of AlexNet[32], whose performance in modelling inferotemporal cells is representative of a large class of feedforward DNNs[6,30,33]. Comparison of the encoding axes for faces versus objects revealed an unexpected phenomenon: stimulus-dependent switching of the neural code. Multiple analyses converged on this conclusion. First, the preferred axes derived from responses to non-face objects were uncorrelated with those derived from responses to faces: object axes consistently pointed towards the face quadrant of feature space, supporting detection, whereas face axes were more diverse. Second, time-resolved analysis revealed that the face-encoding axis initially aligned with the object-encoding axis, consistent with domain-general processing. However, about 100 ms after the stimulus onset, a clear switch occurred: the face-encoding axis reversed direction in low dimensions of the general object space, neural responses became sparse and tuning to multiple face-specific features rapidly emerged. Strikingly, these dynamics were specific to faces and absent for non-face objects. This late-emerging tuning improved the reconstruction of face identity from neural responses, demonstrating the computational importance of the switch. Together, these results show that face cells achieve domain specificity through a rapid, concerted change in their neural code, resolving a longstanding debate on domain specificity in the inferotemporal cortex[17–22]. More broadly, the results reveal a new mechanism for neural representation: stimulus-dependent switching of the neural code.

We identified face patches ML (middle lateral) and AM (anterior medial) in three macaque monkeys by using functional MRI (fMRI)[15] (Methods). We then targeted NHP Neuropixels probes[34] to the ML and AM face patches of the three monkeys (Fig. 1e and Extended Data Fig. 1a–d). Neural responses were recorded while the animals passively fixated 1,525 images of real human faces and 1,392 images of diverse non-face objects (Fig. 1f). Each stimulus was shown for 150 ms followed by an interstimulus interval of 150 ms. In the figures, we show results from the ML face patch of monkey A. Corresponding data from the ML face patch in monkey J are shown in Extended Data Fig. 2, and results from the AM face patch in monkeys A and M are shown in Extended Data Fig. 3.

## Distinct axes for faces and objects

Cells in both the ML and AM face patches responded more to faces than to objects (Fig. 1g,h and Extended Data Fig. 1e,f), with 54.5% of ML cells ($n = 310$) and 51.6% of AM cells ($n = 153$) having a face selectivity $d' \geq 0.2$ (Extended Data Fig. 1g,h and Methods).

We first passed a combined set of face and object stimuli through AlexNet and performed principal component analysis (PCA) on the fc6 layer embedding to extract a general 60-dimension feature space capturing features of both faces and objects[6]. For each face cell, we used responses to the non-face object stimuli (averaged over 50–220 ms) to compute a preferred axis, the object axis, by linearly regressing responses of the cell to the 60-dimension feature vectors corresponding to different objects (Fig. 1b). Similarly, we used responses to face stimuli to compute a preferred face axis. We confirmed that the object and face axes could explain significantly more variance in responses to objects and faces, respectively, compared with a shuffle control (Extended Data Fig. 4a,b).

We wanted to know whether single face cells encode the entire object space using a single axis, or whether they use multiple axes (Fig. 1d). We first compared the face and object axes by visualizing them in a 2D space corresponding to the first two PCs of the 60-dimension feature

space. In agreement with a previous study[6], we found that the object axes of face cells consistently pointed towards the upper right quadrant of the PC1–PC2 space (Fig. 2a, left; standard deviation (s.d.) of angles = 37.4°), where faces reside (Extended Data Fig. 4c). By contrast, the face axes of the same cells pointed in diverse directions (Fig. 2a, right; s.d. of angles = 81.9°; comparing face and object axis variance, $F(1,150) = 214.7$, $P < 4.9 \times 10^{-37}$). To ensure that the consistency of the object axes was not due to any inherent non-uniform distribution of the object stimuli, we selected a subset of object stimuli that were Gaussian distributed (Extended Data Fig. 4d,e). This did not change the qualitative difference in angular distributions for object compared with face axes (Extended Data Fig. 4f,g), which was confirmed to be destroyed by stimulus shuffling (Extended Data Fig. 4h,i). A scatter plot of face and object axis weights for PC1 confirmed the lack of a clear correlation (Fig. 2b, left; $r = 0.16$, $P = 0.04$), and similarly for PC2 (Fig. 2b, right; $r = -0.07$, $P = 0.37$). Further confirming the lack of correlation between face and object axes, the face axis could not explain any variance in the object responses, and vice versa (Fig. 2c).

The lack of correlation between the face and object axes was evident in the raw responses of single cells. For example, Fig. 2d shows a cell with diametrically opposite face and object axes. Responses of this cell to objects were well captured by the object axis (Fig. 2d, top left), and responses to faces were well captured by the face axis (Fig. 2d, top right), but the two axes clearly pointed in opposite directions in the PC1–PC2 space (Fig. 2d, bottom). Thus, it is clear from the raw responses that a single axis cannot explain the selectivity of this cell for faces and objects (Fig. 1d).

Cells in the AM face patch showed a similar pattern (Extended Data Fig. 3a–d), with object axes consistently pointing towards the face quadrant and face axes pointing in diverse directions.

The difference between the face and object axes was not dependent on prior expectations or adaptation, because we observed the same results when the face and object stimuli were presented in separate blocks (Fig. 2 and Extended Data Fig. 3a–d,q) or randomly interleaved (Extended Data Figs. 2a–d and 3t) (Methods). The axis divergence also could not be attributed to out-of-distribution generalization failure[22], because artificial AlexNet fc6 units, which have single axes in the object space by construction, showed correlated face and object axes (Extended Data Fig. 4j–m).

## Face axis dynamics

If a face cell's face and object axes are distinct, we wanted to know when the cell decides which axis to use. We wondered whether it was evident in the first spikes, or whether it would emerge dynamically over time. Single face cells showed rich response dynamics that differed between faces and objects (Fig. 3a). The axis-encoding framework allowed us to directly address whether such firing-rate changes over time reflect dynamic changes in the neural code for faces and objects. To examine axis-change dynamics, we computed face and object axes using a 20-ms sliding window. Figure 3b shows matrices of mean similarities across the population for (object, object), (face, face) and (face, object) axis pairs at different latencies. The object axis stabilized early at 60 ms, whereas the face axis stabilized later, at 118 ms (Fig. 3b, left and middle). Surprisingly, the (face, object) matrix showed a positive correlation very early, in the 50–100 ms interval (Fig. 3b (right),c). This interval overlapped with the peak of face-evoked population activity (93 ms; Fig. 3d). After 100 ms, however, the face–object axis correlation became negative (Fig. 3b (right),c). Thus, the face axis is transiently aligned to the object axis at short latency before later diverging (see ref. 22, which compared face and object coding but did not analyse tuning dynamics).

To examine axis dynamics in single cells, we computed the cosine similarity between the overall object axis for each cell (computed using a time window of 50–220 ms) and its time-varying face and object axes (Fig. 3e). The face axis showed strong alignment with the object axis at

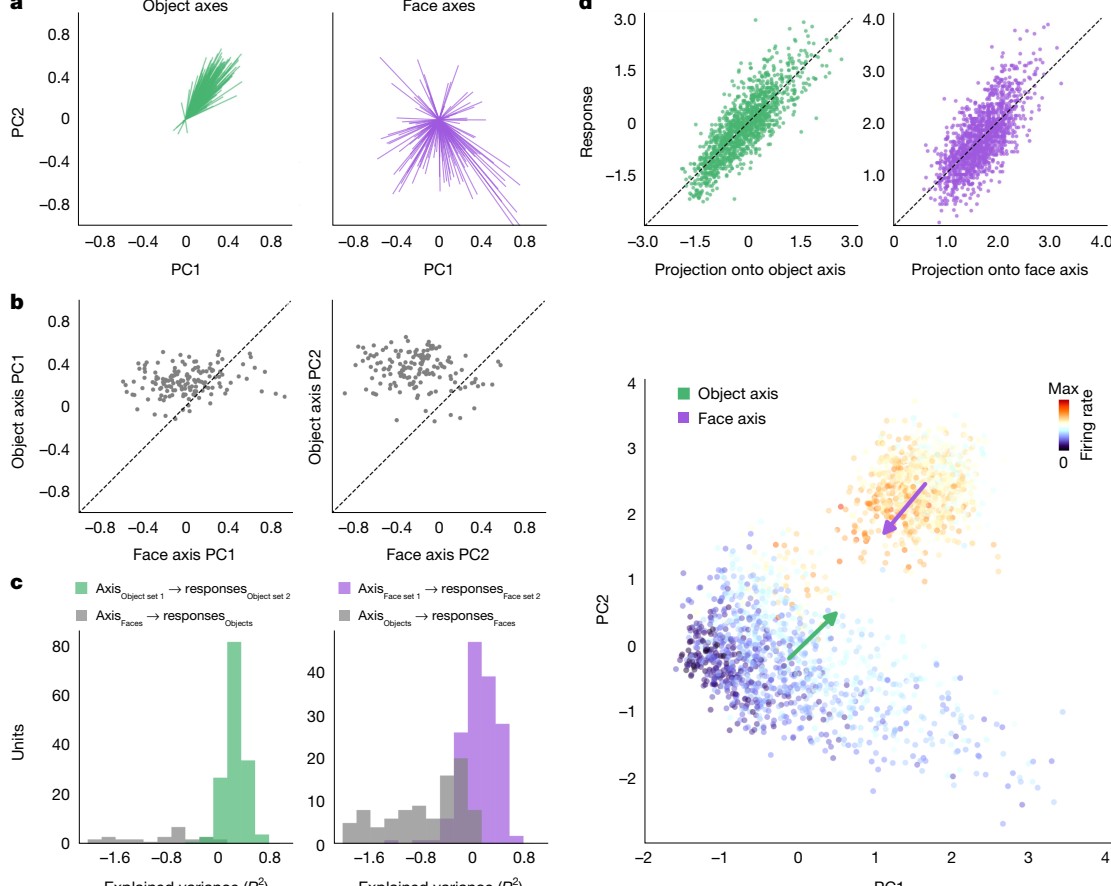

**Fig. 2 | Face cells use distinct axes to represent faces and objects. a**, Distribution of object (left, green) and face (right, purple) axes, projected onto the top two dimensions of the 60-dimension object space (note that faces occupy the upper right quadrant of this space; Extended Data Fig. 4c) ($n$ = 151 cells; only cells with $R^2$ > 0 on the test set for both object and face axes are included; Methods). **b**, Scatter plots of object versus face axis weights for PC1 (left) and PC2 (right). **c**, Cross-prediction accuracies. Left, distribution of explained variances when using face axes to explain object responses (grey), or object axes computed using responses to a random object subset to explain responses to the left-out object subset (green). Right, as in the left, except to explain face responses. Units with $R^2$ < −2 are not shown. **d**, Responses of an example cell with opposite tuning to objects and faces. Top left, scatter plot showing actual responses to objects versus predicted responses computed by projecting images onto the object axis ($R^2$ = 0.62). Top right, scatter plot showing actual responses to faces versus predicted responses computed by projecting images onto the face axis ($R^2$ = 0.54). Bottom: responses of this cell to objects (large cluster in the lower left) and faces (small cluster in the upper right), colour coded by response magnitude (Extended Data Fig. 4c). The object (face) axis of the cell is indicated by the green (purple) arrow.

early latencies (50–100 ms) but then diverged (Fig. 3e, top). The axis divergence times were concerted across the population (Extended Data Fig. 5a). The object axis, by contrast, showed high correlation throughout the entire presentation duration (Fig. 3e, bottom). The cells in Fig. 3e are sorted according to their face selectivity $d'$; divergence was more prominent for highly face-selective cells.

Inspection of raw responses of an example cell projected onto PC1–PC2 space (Fig. 3f) revealed that, over time, the face axis did not merely diverge from the object axis; it actually reversed direction. In this cell, face and object axes were aligned at 80–100 ms, both pointing towards the face quadrant of PC1–PC2 space. Thereafter, the face axis suddenly reversed direction at 120–140 ms. By contrast, the object axis did not change over time.

This pattern of face-axis reversal in the PC1–PC2 subspace of the general object space at around 100 ms was extremely common across the population (Extended Data Fig. 5b,c). Figure 3g illustrates this phenomenon in eight further example cells. Across the population, 62% of cells clearly flipped tuning in PC1–PC2 space (Extended Data Fig. 5d). Notably, this reversal was accompanied by increased response sparseness across the face-cell population (Fig. 3h). This reversal is consistent with the earlier observation of divergent face axes using a time-averaged response (Fig. 2a), which probably reflected temporal

blurring, underscoring the value of time-resolved analysis. The early alignment of face and object axes in a direction pointing towards faces followed by subsequent specific reversal of face axes is consistent with a transition from face detection to face discrimination (Fig. 1a, top).

Cells in the AM face patch showed a largely similar pattern of dynamics, but with longer latency overall (Extended Data Fig. 3e–j,r,s,u,v). As with ML, face axes in the AM face patch initially aligned with the object axis (at around 100 ms) before reversing in low dimensions (interestingly, there was also a weak, brief second period of re-alignment at about 160 ms that was not observed in ML). Overall, 57% of AM cells exhibited a clear tuning reversal (Extended Data Fig. 5e). The fact that face axis reversal in AM occurred later than in ML indicates that the ML reversal was due to local recurrence, rather than to long-range feedback from AM.

Direct assessment of tuning from the raw neural responses yielded results consistent with the axis-based analyses (Extended Data Fig. 5f,h). Given that face axis reversal is evident in the raw responses and occurs consistently across face-selective cells, we reasoned that the response rank across the population should reverse when the monkey sees a face. If true, this would offer a powerful way to assess axis change in response to single images. We computed a single-stimulus axis-change score by tracking shifts in population rank over time (Extended Data

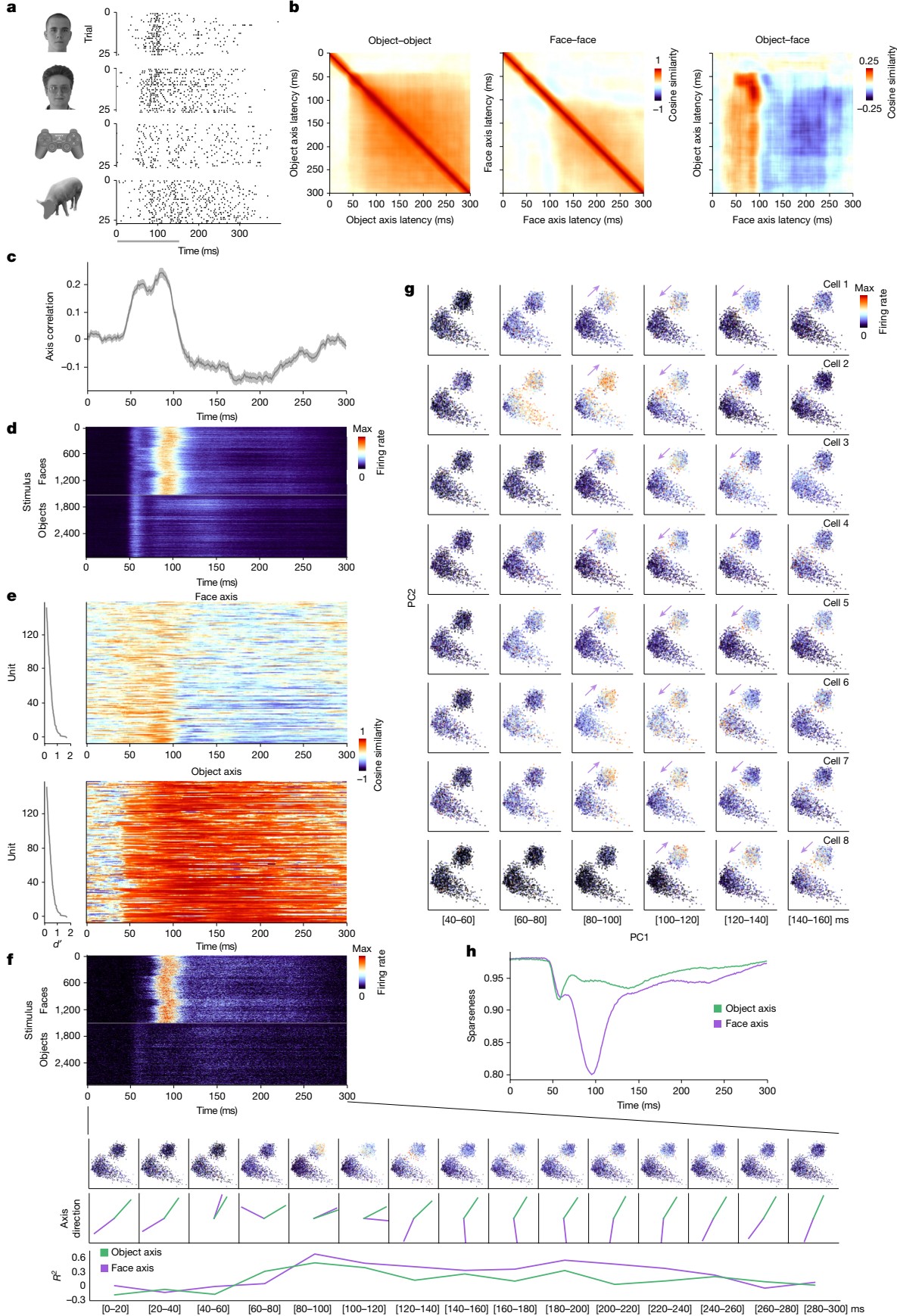

**Fig. 3 | See next page for caption.**

**Fig. 3 | Face and object axes transiently align before diverging. a**, Raster plot of responses of an example ML cell to two face and two non-face stimuli. The grey bar at the bottom indicates stimulus on time. **b**, Left to right, matrices of mean cosine similarities across the population between (object, object), (face, face) and (face, object) axes (computed using all 60 dimensions) for different pairs of latencies ($n = 151$ cells; only cells with $R^2 > 0$ on the test set for both object and face axes are included; Methods). **c**, Correlation between face and object axes as a function of time (diagonal values in the right-most similarity matrix in **b**). The shaded region indicates s.e.m. **d**, Mean response time course to each face and object stimulus, averaged across cells and trials. **e**, Cosine similarity between each cell's overall object axis (computed using a time window of 50–220 ms) and its time-varying face (top) and object (bottom) axes, sorted from top to bottom according to face selectivity, $d'$ (left). **f**, Axis change dynamics of a single cell. Top, mean response time course of this cell to each face and object stimulus, averaged across trials. Expanded row 1: time-resolved scatter plots of face and object stimuli projected onto PC1 and PC2 of object space, colour coded by the cell's response magnitude. Expanded row 2: time-resolved face and object axis directions computed using a time window of 20 ms. Expanded row 3: explained variance ($R^2$) on the test set for object axis (green) and face axis (purple) as a function of time. **g**, Eight example cells showing axis reversal in PC1–PC2 space for faces but not for objects. Purple arrows indicate the approximate face axis direction in each time window. **h**, Response sparseness as a function of time for faces and objects (Methods). Credits: object images in **a** are from ref. 6, Springer Nature Limited; face images in **a** are reproduced from the CVL Face Database (http://lrv.fri.uni-lj.si/facedb.html)[48].

Fig. 5j and Methods). Applied to human faces, monkey faces, dog faces and pareidolia, the metric revealed robust axis reversal for individual human and monkey faces, variable effects for dog faces and weaker effects for pareidolia (Extended Data Fig. 5k,l). This approach bypassed the need for human-defined face and object categories[4], providing an unbiased, data-driven measure of whether a face patch treats an image as a face based solely on neural population dynamics.

Finally, we investigated whether the observed axis-change behaviour is unique to neurons within face patches in the inferotemporal cortex. We recorded from neurons outside face patches while we presented a large, diverse set of objects (Extended Data Fig. 6 and Methods). Many neurons outside face patches, including isolated neurons selective for faces, also exhibited axis change for their preferred stimuli, although this was typically weaker and less consistent than that observed in face-selective populations. This indicates that stimulus-gated axis change may be a general computational motif in the inferotemporal cortex, with the face system representing a particularly robust instance.

We next wanted to investigate the mechanism responsible for stimulus-gated axis change. Control analyses ruled out 'cell-intrinsic' mechanisms that depend only on a neuron's intrinsic input–output properties, including explanations based on mean response differences to faces versus objects (Extended Data Fig. 7a–i), distinct response time courses to high- versus low-responding faces (Extended Data Fig. 7j–l), adaptation (Extended Data Fig. 7m,n) and differences in spatial frequency content of faces versus objects (Extended Data Fig. 7o–v). In particular, we found that faceless animal bodies drove even stronger mean responses than some faces did[35], yet they failed to trigger axis change, whereas those faces did (Extended Data Fig. 7b–i). Conversely, some degraded face stimuli elicited relatively weak responses but still triggered axis change (Extended Data Fig. 8a–e). These analyses, together with our earlier finding ruling out top-down feedback based on later occurrence of axis change in the AM face patch compared to ML (Extended Data Fig. 3e–j), indicate a mechanism for axis change driven by local recurrent dynamics.

## Emergent tuning in higher dimensions

To investigate whether new tuning to face features emerges over time, we first generated a 60-dimension face space by passing face stimuli through AlexNet and performing PCA on the fc6 representation. This new AlexNet face space emphasized differences between face identities, rather than between faces and objects. A plot of face and object axis weights computed in this new space using short (80–100 ms) and long (120–140 ms) latency windows revealed a markedly different pattern for faces (Fig. 4a, top and middle) compared with objects (Fig. 4b, top and middle). For faces, short- versus long-latency axis weights were anti-correlated in low dimensions and decorrelated in higher ones, resulting in a distribution of correlations skewed towards negative values when computed over all 60 dimensions (Fig. 4c, purple; one-sample Student's $t$-test, $t(150) = -7.78$, $P < 1.1 \times 10^{-12}$). By contrast,

for objects, the weights were clearly correlated between the two time windows across all dimensions (Fig. 4c, green; one-sample Student's $t$-test, $t(150) = 16.01$, $P < 2.8 \times 10^{-34}$). Importantly, we confirmed that axis weights for faces computed using short and long time windows in higher dimensions (6–60) were not positively correlated (Fig. 4d, purple; one-sample Student's $t$-test, $t(150) = -3.13$, $P = 0.002$), in contrast to axis weights for objects (Fig. 4d, green; one-sample Student's $t$-test, $t(150) = 14.56$, $P < 1.7 \times 10^{-30}$); thus, the changes in tuning were not confined to low dimensions of the face space. Following the drastic change in face axis weights between 80–100 ms and 120–140 ms, the face axis then stabilized (Fig. 4a, middle and bottom, and Fig. 4e; face: one-sample Student's $t$-test, $t(150) = 29.3$, $P < 5.8 \times 10^{-64}$; object: one-sample Student's $t$-test, $t(150) = 19.1$, $P < 5.5 \times 10^{-42}$).

Another signature of new tuning to higher dimensions would be an increase in the dimensionality of the neural state space over time. To test this, we performed PCA on raw neural population responses to faces at short (80–100 ms) and long (120–140 ms) latencies. Figure 4f shows that more dimensions were required to explain 90% of the response variance at long compared to short latency (mean ± s.d.: 91 ± 5.28 dimensions for long versus 79 ± 2.85 dimensions for short). This difference was significant (paired Student's $t$-test across 100 bootstrap samples, $t(99) = -21.52$, $P < 0.05$), supporting the emergence of new tuning dimensions at later stages of processing.

To directly visualize the new tuning dimensions emerging at long latency, for each cell, we first computed its preferred face axis at 80–100 ms ($v_1$) and at 120–140 ms ($v_2$). We then derived the component of $v_2$ orthogonal to $v_1$ ($v_\perp$) (Fig. 4g). This component represents a new tuning direction orthogonal to the cell's previous tuning direction. We projected the features of each face onto $v_\perp$ ($x$ axis) and the principal orthogonal direction to $v_\perp$ ($y$ axis) and plotted neural responses at 80–100 ms, 120–140 ms, 160–180 ms for four example cells. In each of the four example cells shown, new tuning is apparent along $v_\perp$ at 120–140 ms (Fig. 4h–k, top). To further illustrate what these new dimensions encode, we generated faces varied along $v_\perp$ (Fig. 4h–k, bottom). Each cell showed new tuning to different features, for example $v_\perp$ for cell 1 varied from small inter-eye distance, sharp chin, masculine appearance and light complexion to large inter-eye distance, round chin, feminine appearance and dark complexion.

Cells in the face patch AM showed a very similar pattern, with reversal in low dimensions, new feature tuning in higher dimensions for faces and no change in tuning for objects (Extended Data Fig. 3k–p).

Similar results were obtained using a face space derived from an alternative deep network (ResNet-50 trained on VGGFace2; ref. 36), demonstrating that the axis dynamics are robust to the choice of feature representation (Extended Data Fig. 8f–i). Analyses of tuning directly from raw neural responses further corroborated the axis-based findings (Extended Data Fig. 5g,i).

Overall, these results show that for faces, a marked change in tuning occurs across all dimensions of face space at about 100 ms, involving reversal in low dimensions and loss or gain of tuning in higher dimensions.

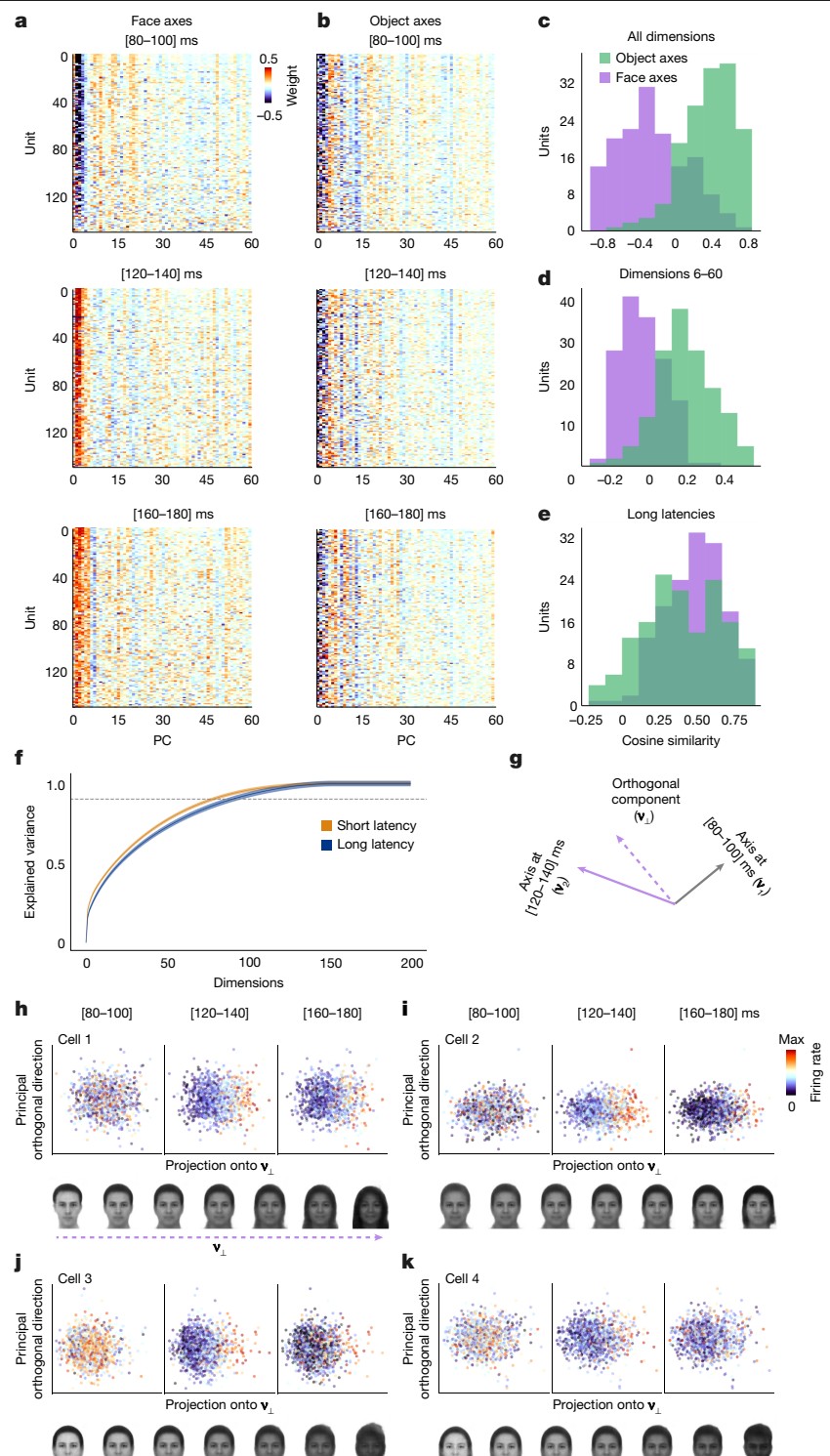

**Fig. 4 | Face axis reversal in low dimensions is accompanied by the emergence of diverse new tuning to higher dimensions of face space. a**, Matrix of face axis weights in the AlexNet face space for each cell, computed using short (80–100 ms, top), long-early (120–140 ms, middle) and long-late (160–180 ms, bottom) latency windows ($n = 151$ cells; only cells with $R^2 > 0$ on the test set for both object and face axes are included; Methods). **b**, Matrix of object axis weights computed at three different latencies; conventions are the same as in **a**. **c**, Purple (green) distribution shows cosine similarities across units between face (object) axis weights at short and long-early latency. All the dimensions (1–60) were used to compute the cosine similarities. **d**, As in **c**, but using higher face space dimensions 6–60 to compute cosine similarities for each cell.

**e**, As in **c**, but showing cosine similarities across units between face (object) axis weights at long-early and long-late latencies (120–140 ms and 160–180 ms), using all 60 dimensions. **f**, Explained variance in z-scored population responses to faces at short (80–100 ms, orange) and long (120–140 ms, blue) latencies, as a function of the number of PCs. Data are shown as mean ± s.d. **g**, Schematic showing the projection of a cell's encoding axis at 120–140 ms ($\mathbf{v}_2$) onto the component orthogonal ($\mathbf{v}_\perp$) to the cell's encoding axis at 80–100 ms ($\mathbf{v}_1$). **h**–**k**, Top, scatter plots of responses to faces projected onto $\mathbf{v}_\perp$ (top) at three different time windows (80–100 ms, 120–140 ms and 140–160 ms) for four example cells. Bottom, faces spanning $\mathbf{v}_\perp$ for each cell, sampled at [−4, −2, −1, 0, 1, 2, 4] $\sigma$, where $\sigma$ is the s.d. of the projections of stimuli along $\mathbf{v}_\perp$ (Methods).

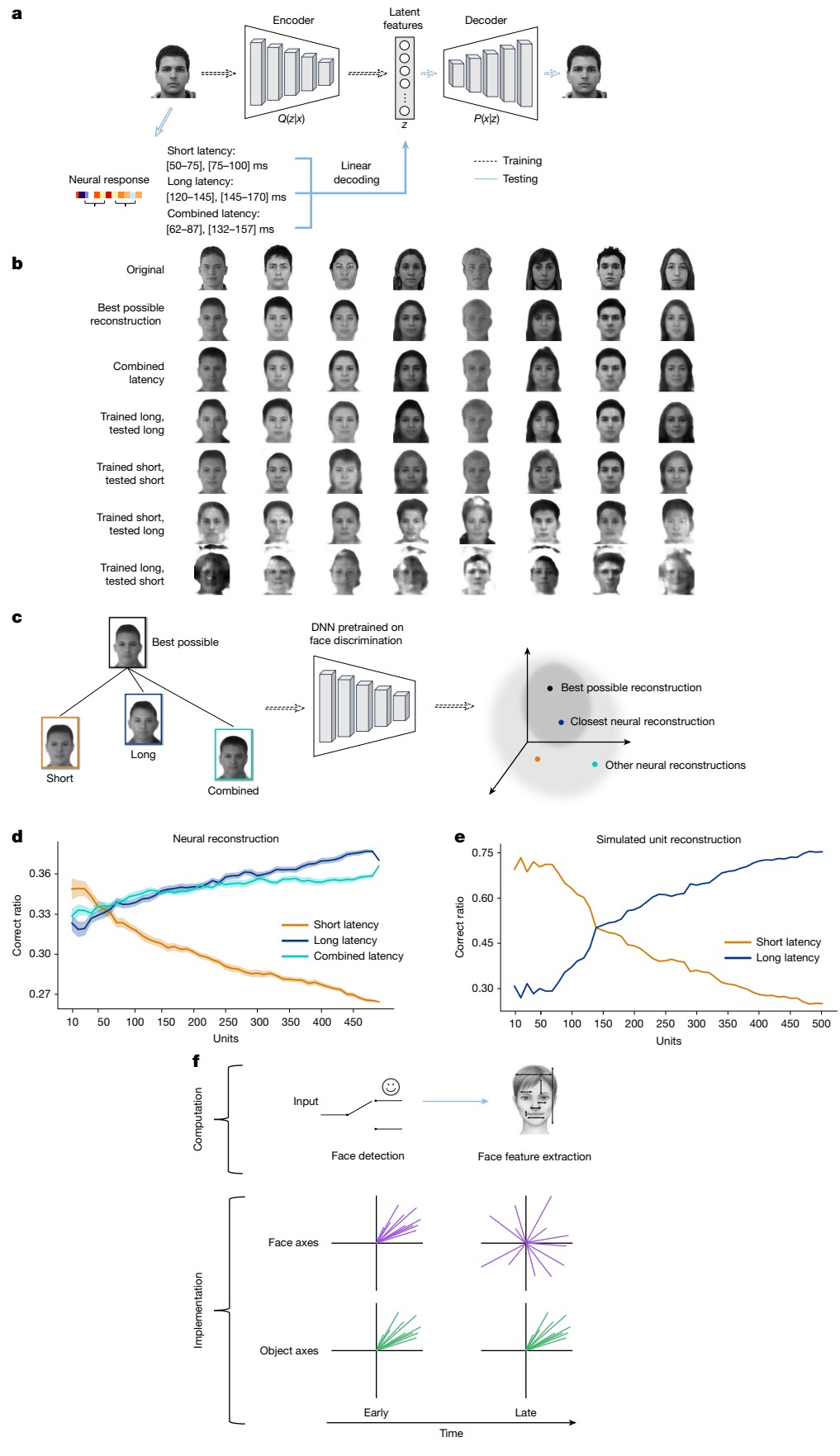

**Fig. 5 | See next page for caption.**

**Fig. 5 | Newly emergent face-encoding axes improve face discrimination. a**, The reconstruction pipeline. We trained an image generator (encoder: images → 512D latent features; decoder: latent features → images; Methods). Latent features were linearly decoded from neural responses in three windows (short, long and combined) to reconstruct faces. Each window comprised two sub-windows, ensuring equal degrees of freedom. Data from the ML (*n* = 332 cells) and AM (*n* = 154 cells) face patches were pooled, with AM windows delayed by 20 ms. **b**, Row 1: eight example faces. Row 2: best possible reconstructions using ground-truth latent features. Row 3: reconstructions from combined-window activity using a decoder trained and tested on combined responses. Row 4: decoder trained and tested on long-latency responses. Row 5: decoder trained and tested on short-latency responses. Row 6: decoder trained on short- and tested on long-latency responses. Row 7: decoder trained on long- and tested on short-latency responses. **c**, A face-discrimination DNN was used to identify reconstructions; a reconstruction was correct if it was closest (in DNN space) to the optimal reconstruction of the same face. **d**, Face identification performance on faces reconstructed from neural activity in different time windows as a function of the number of cells, following the scheme in **c**. The shaded area denotes the s.e.m. **e**, As in **d**, but computed using simulated units in which short-latency responses had similar tuning across units and were concentrated in a small number of low (high-variance-capturing) dimensions (5 of 60), whereas long-latency responses had more varied tuning across units and spanned many dimensions (60 of 60) (Methods). **f**, Summary schematic showing how a face-detection gate can be implemented through axis dynamics. Early in the response, face and object axes align, both pointing towards the face quadrant, supporting detection. Later in the response, face axes reverse in low dimensions while new tuning directions emerge in face space, enabling fine discrimination. Credits: face images in **a** and **b** are reproduced from the FEI database (https://fei.edu.br/~cet/facedatabase.html)[47]; face images in **b** are reproduced from the CVL Face Database (http://lrv.fri.uni-lj.si/facedb.html)[48] and the Chicago Face Database (chicagofaces.org)[52].

## Late axes improve face discrimination

We wanted to know what the newly emergent encoding axes for faces accomplish. We therefore investigated whether changes in tuning improve face discrimination. To assess the population-level impact, we reconstructed faces from neural activity using linear decoders trained on responses from three different windows: a short latency window; a long latency window; and a combined latency window. Each decoder estimated face features ($f = M_{face}r_{face}$), which were then passed to a variational autoencoder trained to generate faces from latent features (Fig. 5a and Methods).

For the long and combined windows, this approach gave reasonable reconstructions compared with the best possible reconstruction (the reconstruction from veridical face features; Methods) (Fig. 5b, rows 3 and 4). Reconstructions were less accurate using the short window (Fig. 5b, row 5). When we attempted to perform cross-window decoding by training a decoder using short (long) window responses and then applying the decoder to long (short) window responses, face reconstruction completely failed (Fig. 5b, rows 6 and 7). This underscores the complete change in neural code occurring at about 100 ms.

We quantified reconstruction quality by identifying the reconstructed faces using a DNN trained to discriminate faces[36–38]. We compared faces reconstructed using short, long and combined windows, to see which reconstruction was closest to the best possible reconstruction in the DNN space (Fig. 5c). For small numbers of cells, the short-latency response performed best (Fig. 5d, orange curve). However, as cell number increased, the long and combined responses outperformed the short-latency response (Fig. 5d, blue and cyan curves).

This crossover pattern aligns with the idea that short-latency responses are optimized for face detection, whereas long-latency responses support face discrimination through more diverse tuning. Cells showed stronger tuning to low dimensions of face space at short latency than at long latency (Extended Data Fig. 9a,b). These low face-space dimensions, which account for the most variance in faces, are well-explained by low object-space dimensions (Extended Data Fig. 9c), which support face detection. When only a few cells, and thus a limited number of dimensions, are available, it is advantageous to use these low, face-detection dimensions from short latency, because they are also the ones that explain the most variance in faces. By contrast, as more cells become available, more dimensions can be leveraged to encode subtle differences between individual faces, improving discrimination. A simulation of this trade-off, between using a few high-variance, face-detection dimensions and many diverse dimensions, produced the same pattern observed in the actual cell population (Fig. 5e).

We also evaluated face categorization and identity discrimination performance directly from the population responses using a 20-ms sliding window (Methods), without relying on reconstruction. This analysis confirmed that face discrimination accuracy increased at later time points, rising after the peak in categorization accuracy (Extended Data Fig. 9d), consistent with previous studies[15,39,40].

To what extent does the improvement in face discrimination accuracy at long latency depend on the emergence of new tuning to higher dimensions of face space? To test this, we designed an experiment to knock out the contribution of low-dimensional features and assess whether the performance gain at later latencies persisted. We created synthetic faces whose variation along low-dimensional features was fixed (Methods). Despite this, discrimination performance still improved at later time points. This supports the idea that the observed improvement is, at least in part, driven by new tuning to higher-dimensional features essential for fine face discrimination (Extended Data Fig. 9e,f).

Overall, these results show that the marked changes in feature tuning occurring at about 100 ms have a functional consequence, greatly improving the ability of the neural population to perform fine face discrimination.

## Discussion

In this study, we resolve a longstanding debate over whether coding mechanisms in the inferotemporal cortex are domain specific[17–19] or domain general[20–22]. We show that face-selective cells initially use a domain-general code, but following a rapid and concerted population-level event lasting less than 20 ms, the neural code transitions into a domain-specific one. Our findings further indicate that the early, domain-general phase supports face detection, whereas the subsequent domain-specific phase supports face discrimination, offering a concrete neural implementation of the long-hypothesized face-detection gate proposed to explain domain-specific face processing[23] (Fig. 5f). At short latencies, face cells encode stimuli using a shared axis for both faces and objects, oriented towards faces in a DNN-derived feature space, optimally subserving face detection. At longer latencies, however, the same population switches to new axes selectively for faces to optimally distinguish face identities. This transition is marked by a reversal of the face axis in low dimensions of the object space (effectively subtracting the 'DC component' of faces to shift processing away from detection[41]), a change to sparser responses, and emergence of tuning to new facial feature dimensions that improve discrimination between individual faces.

These findings demonstrate that the neural code for faces is not fixed but can change completely in just 20 ms (Fig. 3b–h), complementing a previous report of a long-latency axis change triggered by stimulus familiarity[42]. Crucially, we are not merely observing different latencies for different feature representations, which would be expected given the multiple synaptic pathways through which visual input can reach the inferotemporal cortex. Rather, we have identified a wholesale switch in neural code, from axes mediating face detection to axes mediating

face discrimination. This switch is stimulus gated: it occurs only for faces, while non-face objects continue to evoke responses along fixed object axes throughout the stimulus presentation.

Our results challenge a prevailing view that 'core object recognition'—the ability to rapidly recognize objects in 200 ms, despite substantial variation in appearance—can be explained primarily by feedforward processes[43]. They also call into question the dominant view that deep networks, which are widely considered to be the best models of the inferotemporal cortex, are sufficient to capture core object recognition processes in the inferotemporal cortex. The recognition of clear, isolated faces in the absence of any background clutter is a textbook example of core object recognition. Yet we find that even this process is consistently accompanied by a rapid switch in neural code in the key brain structures mediating face recognition[29]. We speculate that the dynamic switch in encoding axis is mediated by the rich local and long-range recurrent connections in the inferotemporal cortex[5,42]. Supporting this idea, a recurrent neural network model showed that a simple form of lateral inhibition can lead to axis reversal (Extended Data Fig. 10). Lateral inhibition was one of the earliest circuit motifs to be identified[44,45], but we are only beginning to understand its full range of computational functions[46].

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

## Methods

Three male rhesus macaques (*Macaca mulatta*) were used in this study. All procedures conformed to local and US National Institutes of Health guidelines, including the US National Institutes of Health Guide for the Care and Use of Laboratory Animals. All experiments were performed with the approval of the UC Berkeley Animal Care and Use Committee.

No statistical methods were used to predetermine sample size. The experiments were not randomized and investigators were not blinded to allocation during experiments and outcome assessment.

### Visual stimuli

**Face-patch localizer.** The fMRI localizer stimulus contained five types of blocks, consisting of images of faces, hands, technological objects, vegetables and fruits, and bodies. Face blocks were presented in alternation with non-face blocks. Each block lasted 24 s (each image lasted 500 ms). In each run, the face block was repeated four times and each of the non-face blocks was shown once. A block of grid-scrambled noise patterns was presented between each stimulus block and at the beginning and end of each run. Each run lasted 408 s.

**Stimuli for electrophysiology experiments. Human faces.** We acquired 2,000 frontal views of faces, as in ref. 42, from various face databases: FERET[49,50]; CVL (Peter Peer, CVL Face Database, Computer Vision Laboratory, University of Ljubljana, Slovenia; http://www.lrv.fri.uni-lj.si/facedb.html)[48]; MR2, ref. 51; Chicago Face Database[52]; CelebA[53]; FEI (fei.edu.br/~cet/facedatabase.html)[47]; PICS (pics.stir.ac.uk); Caltech Face Dataset 1999 (Caltech DATA, 2022; https://doi.org/10.22002/D1.20237); Essex (Face Recognition Data, University of Essex, UK); and MUCT (www.milbo.org/muct)[54]. The faces were aligned using facial landmarks as in ref. 42 with an open-source face aligner (github.com/jrosebr1/imutils).

For Extended Data Fig. 4c (right), we used synthetic face images from Syn-Vis-v0[55].

**Non-face objects.** We used 1,392 different images of isolated objects, previously used in ref. 6.

For monkey A, we presented a subset of 1,525 faces and 1,392 objects. For monkey J, we presented all 2,000 faces and 1,392 objects. For monkey M, we presented a subset of 1,050 faces and 1,392 objects. We presented a smaller number of faces to monkeys A and M to increase the number of repetitions per stimulus; all the analyses for monkey J used only the same subset of 1,525 faces that was presented to monkey A. For all experiments using monkey A and Extended Data Fig. 3q–s for monkey M, we showed the human faces and objects in separate blocks. In each block, we showed a training set (a random subset of 1,425 faces or 1,292 objects) for one repetition, followed by the test set (the remaining 100 faces or 100 objects) for three repetitions. The face and object blocks were interleaved and repeated until enough repetitions were acquired (around 10 repetitions per training image and 30 repetitions per test image). For all experiments using monkey J and Extended Data Fig. 3t–v for monkey M, stimuli were not separated into blocks; instead, images were drawn randomly from the combined pool of faces and objects. All images were rendered in greyscale and subtended 3.9° × 3.9° of visual angle.

**Mixed image pool.** To characterize neurons outside face patches, we presented a combination of six image sets: stubby objects (*n* = 600), spiky objects (*n* = 600), monkey bodies (*n* = 600), animals (*n* = 1,087), faces (*n* = 600) and other general objects (*n* = 1,593). The stubby, spiky, animal and general-object images were previously used in ref. 6. The monkey body images were sourced from Flickr (https://www.flickr.com/) and edited manually to mask visible faces. The face images were randomly selected from the human face set described above.

**Non-human and ambiguous face-like stimuli.** To probe responses to non-human and ambiguous face stimuli, we used monkey faces (*n* = 1,100, previously used in ref. 42), dog faces (*n* = 600, from the AFHQ (Animal Faces-HQ) dataset, https://www.kaggle.com/datasets/andrewmvd/animal-faces) and pareidolia images with matched controls (*n* = 400, from ref. 56).

The illustrative dog photograph in Extended Data Fig. 5l and the illustrative monkey-body photographs in Extended Data Fig. 6a,b were obtained from StockSnap (stocksnap.io; CC0).

**Degraded faces.** To test responses to degraded faces, we rendered occluded, noisy (blended with 50% spectrally equivalent noise) and Mooney versions of 200 images from the human face set described above.

**Synthetic faces.** We generated 2,000 synthetic faces using the face model introduced in ref. 3, which defines a 50D face feature space consisting of 25 shape and 25 appearance dimensions. To constrain variation along low-level features, we restricted the first five shape and first five appearance dimensions (a 10D low-level subspace) to five discrete vectors and generated 400 faces per low-level position. The remaining higher-dimensional features were allowed to vary freely, sampled from Gaussian distributions, ensuring that identity differences in this set were driven primarily by high-dimensional variation.

**Face cell screening set.** We presented a screening stimulus set consisting of six object categories (faces, bodies, hands, gadgets, fruits and scrambled patterns) with 16 identities each (Extended Data Fig. 1e–h).

### Behavioural task

For electrophysiology experiments, monkeys were head-fixed and passively viewed a screen in a dark room. Stimuli were presented on an LCD monitor (Asus ROG Swift PG43UQ). Screen size measured 26.0° × 43.9° visual angle. Gaze position was monitored using an infrared-camera eye-tracking system (Eyelink, SR Research) sampled at 1,000 Hz.

All monkeys performed a passive fixation task for both fMRI scanning and electrophysiological recording. Juice reward was delivered every 2–4 s in return for monkeys maintaining fixation on a small spot (0.2° in diameter).

### MRI scanning and analysis

Subjects were scanned in a 3T PRISMA (Siemens) magnet. First, anatomical scans were done using a single-loop coil at isotropic 0.5 mm resolution. Then functional scans were done using a custom eight-channel coil (MGH) at isotropic 1 mm resolution, while the monkeys performed a passive fixation task. Contrast agent (Molday ION) was injected to improve the signal-to-noise ratio. Further details about the scanning protocol can be found in ref. 28.

Analysis of functional volumes was done using the FreeSurfer Functional Analysis Stream[57] and FSL[58]. Volumes were corrected for motion and undistorted based on the acquired field map. Runs in which the norm of the residuals of a quadratic fit of displacement during the run exceeded 5 mm and the maximum displacement exceeded 0.55 mm were discarded. The resulting data were analysed using a standard general linear model. The face contrast was computed as the average of all face blocks compared with the average of all non-face blocks.

### Electrophysiological recording

We used Neuropixels 1.0 NHP (non-human primate) probes[34] (probes 45 mm long with 4,416 contacts along the shaft, of which 384 are selectable at any time) to perform electrophysiology targeted to the face patches ML and AM. To insert the probes, we developed a custom insertion system composed of a linear rail bearing and 3D-printed fixture enabling a precise insertion trajectory. Face patches were initially targeted with single tungsten electrodes before the Neuropixels recordings, following methods for MRI-guided electrophysiology described previously[59]. All Neuropixels data were acquired using SpikeGLX or OpenEphys[60] acquisition software and spike sorted using Kilosort 3 or 4[61,62] with the threshold parameter set to (10, 4).

We included data from monkey A for 13 sessions, from monkey J for one session and from monkey M for one session. Results across all sessions were consistent. Data shown in Figs. 1–5 and Extended Data Fig. 3a–p were from one session using monkey A; data shown in Extended Data Fig. 2 were from one session using monkey J; and data shown in Extended Data Fig. 3q–v were from one session using monkey M.

The percentages of face-selective cells recorded with Neuropixels probes are lower than those reported previously using single tungsten recordings[15,26]. This is probably because Neuropixels probes capture activity from many nearby neurons, including smaller or less well-isolated units, as well as neurons that are not strongly visually driven, and because portions of the probe often extended beyond the face-patch boundaries (Fig. 1g,h and Extended Data Fig. 1g,h).

## Data analysis

Python v.3.10 and MATLAB R2023a were used for analysis. Only visually responsive cells were included for analysis. To determine visual responsiveness, a two-sided Student's $t$-test was done comparing activity between −50 ms and 0 ms with that of 50–300 ms after the stimulus onset. Cells with $P < 0.05$ were included. Wherever further cell selection was done (for example, to cull cells whose activity could be well explained by an axis model, as determined by $R^2$), it is indicated in the relevant section of the paper.

Here, we summarize these inclusion criteria. In Figs. 2–4, we analysed the same subset of cells that satisfied all the following criteria: significant visual responsiveness to our main stimulus set consisting of 1,525 faces and 1,392 objects; non-zero response variance across stimuli; peak $d' \geq 0.2$ within 80–140 ms (see 'Face selectivity index' section); and the presence of positive $R^2$ on a held-out test set for both face and object axes, in both the general object space and the face space (see 'AlexNet general-object space and face space' section), at 80–140 ms. This resulted in 151 of 563 units in ML of monkey A, 76 of 248 units in AM of monkey A, 131 of 467 units in ML of monkey J, and 84 of 353 units in AM of monkey M. For Fig. 5, to include a larger population of cells for face reconstruction, we used a different inclusion criterion. Specifically, we included cells that were visually responsive to the face cell screening set (see 'Visual stimuli' section) and showed significantly different responses to faces versus non-face objects in that set (two-sided Student's $t$-test, $P < 0.05$).

## Face selectivity index

The face selectivity, $d'$, was defined for each cell and each 1 ms time bin as:

$$d'_t = \frac{E(r_{\text{face},t}) - E(r_{\text{object},t})}{\sqrt{\frac{1}{2}(\sigma^2_{\text{face},t} + \sigma^2_{\text{object},t})}}$$

Where $E(r_t)$ and $\sigma^2_t$ are the mean and variance of responses of the cell over stimuli at time $t$, respectively. We further calculated the peak $d'$ over a 20-ms sliding window between 80 ms and 140 ms (corresponding to the time interval in which we observed the main face-selective responses in raw time courses; Fig. 3d):

$$\text{peak } d' = \max_{T \in [80,120]} E_{t \in [T,T+20]}(d'_t).$$

We then selected cells with peak $d' \geq 0.2$.

In Extended Data Fig. 1f–h, the face selectivity index (FSI) was defined for each cell as:

$$\text{FSI} = \frac{r_{\text{face}} - r_{\text{non−face}}}{r_{\text{face}} + r_{\text{non−face}}}$$

where $r$ is the average neuronal response in a 50–220 ms window after stimulus onset.

## Average response profile

Mean responses of each cell to each stimulus were computed in a 50–220 ms window after stimulus onset. Responses were then normalized for each cell to the range [0, 1], where the minimum response was assigned 0 and maximum was assigned 1.

## AlexNet general-object space and face space

We used AlexNet[32] to embed stimulus images into a high-dimensional latent space. Specifically, images were passed through a pretrained version of the model in MATLAB (Mathworks) and 4,096-dimension features were extracted from layer fc6. To further reduce dimensionality, we performed PCA. We built two feature spaces using this approach: a 60-dimension general-object space and a 60-dimension face space.

To build the 60-dimension general-object space, we performed PCA on fc6 responses to a set of 100 face images (randomly selected from the 2,000 faces in our face database) and 1,292 object images. This general-object space could explain 80.9% (61.6%) of the variance in the fc6 features of the object (face) stimuli. We normalized each dimension such that the projection of all stimuli along the dimension had a mean of 0 and an s.d. of 1. This general-object space was used for the analyses shown in Figs. 2 and 3 and Extended Data Figs. 2a–j, 3a–j, q–v, 4, 5a–e. For visualization purposes, we also used the 2D subspace consisting of the first two PCs of this space, as in ref. 6.

To build a 60-dimension face space capturing variation in face-specific features, we performed PCA on fc6 responses to a set of 1,425 face images. The face space explained 86.4% of the variance in the fc6 features of the face stimuli. This face space was used for the analyses shown in Fig. 4 and Extended Data Figs. 2k–p, 3k–p and 9a–c.

To build a 60-dimension face feature space using a ResNet-50 model trained on the VGGFace2 dataset[36], we used a PyTorch implementation of the VGGFace2 model available at https://github.com/cydonia999/VGGFace2-pytorch. We extracted responses from the final fully connected (fc) layer to a set of 1,425 face images and performed PCA. The resulting 60 principal components defined the face feature space used in the analyses shown in Extended Data Fig. 8f–i.

## Preferred axis of cells

The preferred axis of each cell was computed using linear regression as follows:

$$P_{\text{lin}} = (\mathbf{r} - \bar{r})F(F^T F)^{-1}$$

where $\mathbf{r}$ is a $1 \times n$ vector of the firing-rate response to a set of $n$ face stimuli, $\bar{r}$ is the mean firing rate and $F$ is an $n \times d$ matrix, where each row consists of the $d$ parameters representing each image in the feature space. As mentioned above in 'Stimuli for electrophysiology experiments', we split the full image set into training and test sets, and linear regression models were trained on the training set and tested on the held-out test set. In all figures, $R^2$ for the held-out test set is shown.

## Stimulus distribution control

To evaluate whether the shape of our stimulus distribution had any effect on the axis directions (Extended Data Fig. 4d–i), we fitted multivariate Gaussian distributions to the face and object training sets, and then identified subsets of 500 faces and 500 objects with the highest probability density under the face and object Gaussian distributions, respectively.

## Cross prediction between face and object axes

To evaluate how well the face axis generalizes to object responses and vice versa, we trained a linear regression model with the training set for faces (objects) and then tested on the test set for objects (faces) (Fig. 2c and Extended Data Figs. 2c, 3c and 4l).

## Normalized face–object axis correlation

To quantify the correlation between the face and object axes of each cell while accounting for the out-of-distribution effect, we first measured the within-category axis correlation by calculating the correlation between axes estimated from two random subsets of images from the same category. We then used the mean of the face subset–face subset correlation and the object subset–object subset correlation for each cell as the upper bound imposed by the out-of-distribution effect. Finally, we calculated the correlation between the face and object axes of each cell and divided this raw correlation by the upper bound to obtain the normalized face–object axis correlation (Extended Data Figs. 2d, 3d and 4m).

## Artificial-unit comparison

To observe how a unit with a single axis across both face and object space would behave, we identified artificial face-selective units from the AlexNet fc6 layer and redid our analyses for real neural units on them (Extended Data Fig. 4j–m). Because the general-object space was built by PCA from the activities of AlexNet fc6-layer unit activities, a unit in the same layer should respond linearly to the features in the general-object space, thus providing a model neuron with a single encoding axis. Given that there are features not perfectly explained by the feature space, the artificial units should preserve effects from the out-of-distribution generalization. We first identified face-selective AlexNet fc6 units by calculating the FSI for each unit using its response to the face cell screening set. We then repeated the analyses of Fig. 2a–c using the responses of the 126 most face-selective units (FSI ranging from 0.23 to 0.46).

## Time-varying axis analysis

We fitted axes to the average neural responses in 20-ms windows over the trial duration of 0–300 ms after stimulus onset (Figs. 3 and 4 and the related Extended Data figures). To quantify the alignment between axes at different latencies, we computed cosine similarity between the time-varying face or object axis at each latency and the trial-wide object axis computed across 50–220 ms (Fig. 3e and Extended Data Figs. 2h, 3h and 8a–e, rightmost column).

## Population response sparseness

To characterize the sparseness of face-cell population responses to faces and objects, we computed a modified Treves-Rolls population sparseness[63]. Specifically, we calculated

$$S_{i,t} = 1 - \frac{\left(\sum_{j=1}^{N} r_{j,i,t}\right)^2}{N \sum_{j=1}^{N} r_{j,i,t}^2}$$

where $N$ is the number of neurons and $r_{j,i,t}$ is the response of neuron $j$ to stimulus $i$ at time $t$, for all stimuli and times. To compare population response sparseness to faces and objects, we took the mean over all face and all object stimuli, respectively, at each time point.

## Identifying top- and bottom-projected images for response PC1 and PC2

We performed PCA on the time-averaged population response matrix across all stimuli and treated the first two PCs (PC1 and PC2) as pseudo-units. For each response PC, we computed its face and object axes using a 20-ms sliding time window. We then projected all the stimuli onto each axis and identified the top five and bottom five images based on projection values, representing the most and least preferred stimuli for each axis over time.

## Single-stimulus axis-change score

To assess whether an individual face image elicits a change in the population code, we avoided any per-cell normalization and worked with raw firing rates. Because large between-cell firing rate differences can mask population rank-order changes, we excluded extreme-firing-rate cells before analysis.

For each image, we computed the Pearson correlation between the population response vectors at an early (60–80 ms) and a late (100–120 ms) window across all face-selective cells. We then mapped the correlation values to face probabilities using a simple single-feature logistic regression classifier trained to discriminate between face and object labels from the correlations. The classifier output is reported as the single-stimulus axis-change score: higher values indicate a greater 'faceness' of an image (more negative early-versus-late population correlation).

## Recording outside face patches and quantifying image selectivity

We used fMRI localizers[6] to target the middle stubby, spiky and body patches in the monkeys as in ref. 6. Neuropixels probes were inserted into these regions and neural activity was recorded while the monkeys viewed a mixed image pool (see 'Visual stimuli' section). For each neuron, we quantified discriminability (peak $d'$) for its most responsive stimulus category using the same procedure described above for quantifying face selectivity. Neurons were pooled across sessions for subsequent analyses (Extended Data Fig. 6).

## Computing axis correlation between short and long latencies

To quantify changes in face and object axes (Fig. 4 and related Extended Data figures), we computed the cosine similarity between the face axes at three different latencies (80–100 ms, 120–140 ms and 160–180 ms), and similarly for object axes. We chose to focus on these three time windows because they straddled the population response peak of face cells (Fig. 3d). We note that individual cells sometimes showed salient changes in response to objects over time; our analyses and conclusions are intended to capture population-wide changes in tuning.

In Fig. 4c,d, to better account for different latencies in responses of different units to faces and objects (see the example cell in Fig. 3f, top), for each unit, we took the short latency to be the first 20-ms window in which the mean response of that unit was at least 2 s.d. above the mean baseline (−25–25 ms) response of that unit to faces and objects, respectively.

## Control analysis to rule out cell-intrinsic axis-change hypotheses

In hypothetical scenario one, axis dynamics is always accompanied by a high mean response magnitude (Extended Data Fig. 7a). To address this, we identified the 100 most-effective non-face stimuli, as well as the 100 least-effective face stimuli (Extended Data Fig. 7b). The former evoked a greater mean response, averaged over 50–220 ms, than the latter (Extended Data Fig. 7c; mean response ratio = 1.29). However, the two types of stimuli evoked very different response dynamics (Extended Data Fig. 7d), and comparison of axis weights across different time windows revealed axis change only in response to the face stimuli (Extended Data Fig. 7e–i). This shows that the presence of discriminable face features is necessary to trigger axis change; high mean response magnitude is insufficient.

In hypothetical scenario two, axis-change dynamics could be due to delayed responses to weaker stimuli leading to a change in tuning (Extended Data Fig. 7j). Specifically, if weaker responses are more delayed, then at longer latencies, stronger responses may already have subsided while weaker responses persist. This temporal offset could result in weaker responses appearing stronger than the diminished stronger responses, thereby giving the phenomenon of a reversed tuning profile. To address this, we first identified the 50% most-effective and 50% least-effective face stimuli for each cell, determined by the mean response in the early time window 80–100 ms. We then computed face axes separately using these two groups of stimuli, at both short (80–100 ms) and long (120–140 ms) latency. If axis-change

dynamics were driven solely by delayed onset of weak stimuli, then one would predict: a lack of correlation between axes computed using the most-effective and least-effective faces at long latency; and positive correlation between axes at short and long latency computed using the most-effective faces (determined by response at short latency). Neither of these predictions was supported by the data (Extended Data Fig. 7k,l). In particular, the negative correlation that we observed experimentally between axes computed using the most-effective faces at short and long latency rules out the possibility that axis reversal is driven solely by responses to non-optimal, low-contrast stimuli (Extended Data Fig. 7l).

In hypothetical scenario three, axis-change dynamics could be due to a cell-intrinsic increase in firing threshold following a strong transient response (adaptation; Extended Data Fig. 7m). To investigate this possibility, we measured the axis tuning of model cells encoding a single axis with and without application of a raised threshold. We found that the resulting axes were still highly correlated (Extended Data Fig. 7n). This contrasts with our actual results (Fig. 4a–d), ruling out cell-intrinsic adaptation as the source of the change in neural code.

### Generating low and high spatial frequency-filtered images
Each image from our original set of 1,525 face images was first convolved with a Gaussian filter ($\sigma = 0.044°$) to generate a low spatial frequency-filtered image. This was then subtracted from the original to compute a high spatial frequency image (Extended Data Fig. 7o–v).

### Identifying new tuning directions for each cell
In Fig. 4c,d, to better identify new tuning directions that emerge at long latency (Fig. 4g–k and related Extended Data figures), we compared the face axes of each cell at 80–100 ms, denoted as $\mathbf{v}_1$, to the axes at 120–140 ms, denoted as $\mathbf{v}_2$ (AM time windows were delayed by 20 ms). We decomposed $\mathbf{v}_2 = \mathbf{v}_\parallel + \mathbf{v}_\perp$, where $\mathbf{v}_\parallel = \langle \mathbf{v}_1, \mathbf{v}_2 \rangle (\mathbf{v}_1 / |\mathbf{v}_1|_2)$ is the component of $\mathbf{v}_2$ parallel or antiparallel to $\mathbf{v}_1$, and $\mathbf{v}_\perp$, the component of $\mathbf{v}_2$ orthogonal to $\mathbf{v}_1$, is given by $\mathbf{v}_\perp = \mathbf{v}_2 - \mathbf{v}_\parallel$. For visualizations in Fig. 4h–k, we computed the principal orthogonal direction for each $\mathbf{v}_\perp$. Specifically, for each stimulus with embedding $\mathbf{u}$ in the face space, we orthogonalized $\mathbf{u}$ with respect to $\mathbf{v}_\perp$ by computing $\mathbf{u}_\perp = \mathbf{u} - \langle \mathbf{u}, \mathbf{v}_\perp \rangle (\mathbf{v}_\perp / |\mathbf{v}_\perp|_2)$. Then we performed PCA over all $\mathbf{u}_\perp$ and took the first PC as the principal orthogonal direction to $\mathbf{v}_\perp$.

### Face reconstruction
To reconstruct faces from neural activity (Fig. 5 and Extended Data Fig. 2q,r), we leveraged a probabilistic generative model[64]. The model followed the design of variational autoencoders[65], with an encoder mapping the input images (resized to 128 × 128) into latent features defined with a variational distribution and a decoder projecting the features to the original inputs. The encoder and decoder each consisted of five convolutional layers, and the latent features were set to 512 dimensions. A regularizer was used to minimize discrepancies between the latent distribution and a Gaussian prior, which better supports data on low-dimensional manifolds. We also included an additional objective to align the latent features (after linear projection) with fc6 features from AlexNet, to ensure that the latent space described similar features to the general-object space. We trained the generative model on 1,900 faces and 1,292 objects, and validated the model on the 200 held-out images (100 faces and 100 objects).

To compare the reconstructed faces from neural responses at short and long latency, we used averaged population responses from several time windows. For ML cells, short-latency responses were averaged from 50–75 ms and 75–100 ms, and long-latency responses from 120–145 ms and 145–170 ms. We also used a combined window composed of windows from 62–87 ms and 132–157 ms. Thus, each of the three windows comprised two sub-windows. Note that the three windows each spanned exactly the same duration of time. The AM time windows were delayed by 20 ms. For each window, we treated responses from the two sub-windows as if they were from distinct cells, enabling the decoder to assign distinct axes to each sub-window (Fig. 5a).

For reconstruction, after training and freezing the generative model, we linearly decoded 512-dimension feature vectors from neural responses to each image using responses from the short, long and combined time windows (thus three separate linear decoders were trained). We then fed the latent features into the decoder of the probabilistic generative model to reconstruct faces. For learning linear decoders mapping neural activity to latent features, we trained on 1,425 face images (a subset of the 1,900 images used for generative model training) and tested on the remaining 100.

We also passed each original face image through the encoder and decoder of the generative model directly to obtain the best possible reconstruction of each face, allowing us to separate fine face-feature loss in the generative model itself from decoding inaccuracy due to noise in the neural signal.

### Visualizing new tuning directions
To visualize new tuning directions of cells ($\mathbf{v}_\perp$; Fig. 4h–k), for each cell we reconstructed a series of faces with features that gradually varied along $\mathbf{v}_\perp$, leveraging the probabilistic generative model described above ('Face reconstruction' section). Because $\mathbf{v}_\perp$ lies in our 60-dimension face space, we also trained a linear transformation matrix to transform features from this 60-dimension face space to the generative model's latent space (512 dimensions). We first normalized each cell's axis to unit length and then sampled seven locations along it, at projection lengths of $[-4, -2, -1, 0, 1, 2, 4]\,\sigma$, where $\sigma$ is the s.d. of the projections of our 1,425 original faces onto $\mathbf{v}_\perp$. This yielded seven 60-dimension feature vectors, which we then transformed to 512 dimensions. Finally, we fed these 512-dimension features into the generative model to generate the reconstructed faces.

### Using a DNN to quantify face identification performance
To quantify the goodness of reconstruction (beyond human visual evaluation; Fig. 5c,d), we leveraged a DNN pretrained to discriminate faces[38] (using the VGGFace2 dataset[36]), whose feature space provides an objective metric for evaluating similarity between faces. We embedded all our neural-reconstructed faces and best possible reconstructed faces into the feature space of this DNN and then calculated the Euclidean distance between each face's neural reconstruction and its best possible counterpart.

To compare the goodness of reconstruction among the three time windows, in Fig. 5c,d, we compared the distance of reconstructions from the three time windows to the best possible reconstruction for each face. The closest face of the three was considered to be 'correct'. We repeated this recognition task for all 1,525 faces. We further repeated this whole process using only subsets of the neurons to investigate how the number of cells affects the reconstruction quality for different time windows. For each cell count, we randomly resampled the cells 30 times to estimate the s.e.m.

### Simulating results of face reconstruction analyses
We simulated a neural population whose short-latency responses tuned to a smaller set of face dimensions and whose long-latency responses tuned to a larger set of face dimensions, to test our hypothesis that the results of Fig. 5d might be explained by a trade-off between redundancy and diversity.

Simulated neuron responses were created by linearly combining different dimensions of the latent space features of the DNN[38] mentioned above. We first performed PCA on this DNN's latent space to get a 60-dimension feature for each image. Then, for simulating short-latency responses of each artificial neuron, we linearly combined a random subset of the first five dimensions of the features and applied Gaussian noise. For long-latency responses, we linearly combined a

random subset of dimensions from the full 60 dimensions and applied Gaussian noise. These simulated neurons were then treated as real neurons and underwent the face reconstruction and recognition analyses described above.

## Face categorization and discrimination time course

For both face categorization (face versus object) and face identity discrimination, we summarized how well the two categories or single identities are separated in population space. In each time window, let $\mathbf{r}_i \in \mathbf{r}^N$ be the population response vector (across $N$ units) to face image $i$ (distinct identity). We computed all pairwise Euclidean distances $d_{ij} = \|\mathbf{r}_i - \mathbf{r}_j\|_2$ across categories or across identities and reported a population separation index (PSI):

$$\text{PSI} = \frac{\text{mean}_{i<j} d_{ij}}{\sigma_{\mathbf{r}}}$$

where $\sigma_{\mathbf{r}}$ is the pooled s.d. of responses across units and identities in the window. PSI is unitless and increases when category- or identity-specific responses are more dispersed relative to the overall response variability. We repeated this computation for each 20-ms window to obtain the categorization and discrimination time courses (Extended Data Fig. 9d,f).

## Recurrent neural network training

We trained a recurrent neural network with 100 units whose dynamics at each timestep were described by

$$\mathbf{h}_t = \tanh(\mathbf{x}_t + W\mathbf{h}_{t-1})$$

where $\mathbf{h}_t$ is the vector of unit activations at time $t$, $W$ is the weight matrix and $\mathbf{x}_t$ is the external input at time $t$. During training, the network was run for two timesteps with the same random linear gradient provided as input at both timesteps, and the mean squared error between the output at time 2 and the reversed input gradient was used as the loss. The network was trained for 10,000 iterations, each with a random linear gradient, using the Adam optimizer (learning rate 0.001).

## Reporting summary

Further information on research design is available in the Nature Portfolio Reporting Summary linked to this article.

## Data availability

The data supporting the findings of this study are available from the corresponding authors (Y.S. and D.Y.T.) upon reasonable request.

## Code availability

All the code is available from the corresponding authors (Y.S. and D.Y.T.) upon reasonable request.

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

**Acknowledgements** We thank C. Sohn for animal care support; members of the Tsao laboratory, P. Bao, X. Dai, J. Gallant, J. Gao, S. Kornblith, Y. Ma, F. Tong and A. Tsao for discussions and comments; and P. Bao and L. She for technical advice. Funding: this work was supported by the NSF (to D.B.), NIH (EY030650-01), the Office of Naval Research, and the Howard Hughes Medical Institute.

**Author contributions** Y.S. and D.Y.T. conceived the project and designed the experiments; Y.S., J.K.H. and F.F.L. collected the data; Y.S. and D.B. analysed the data, with help from S.C. Y.S., D.B. and D.Y.T. interpreted the data. Y.S., D.B., J.K.H. and D.Y.T. wrote the paper, with feedback from F.F.L. and S.C.

**Competing interests** The authors declare no competing interests.

**Additional information**
**Correspondence and requests for materials** should be addressed to Yuelin Shi or Doris Y. Tsao.

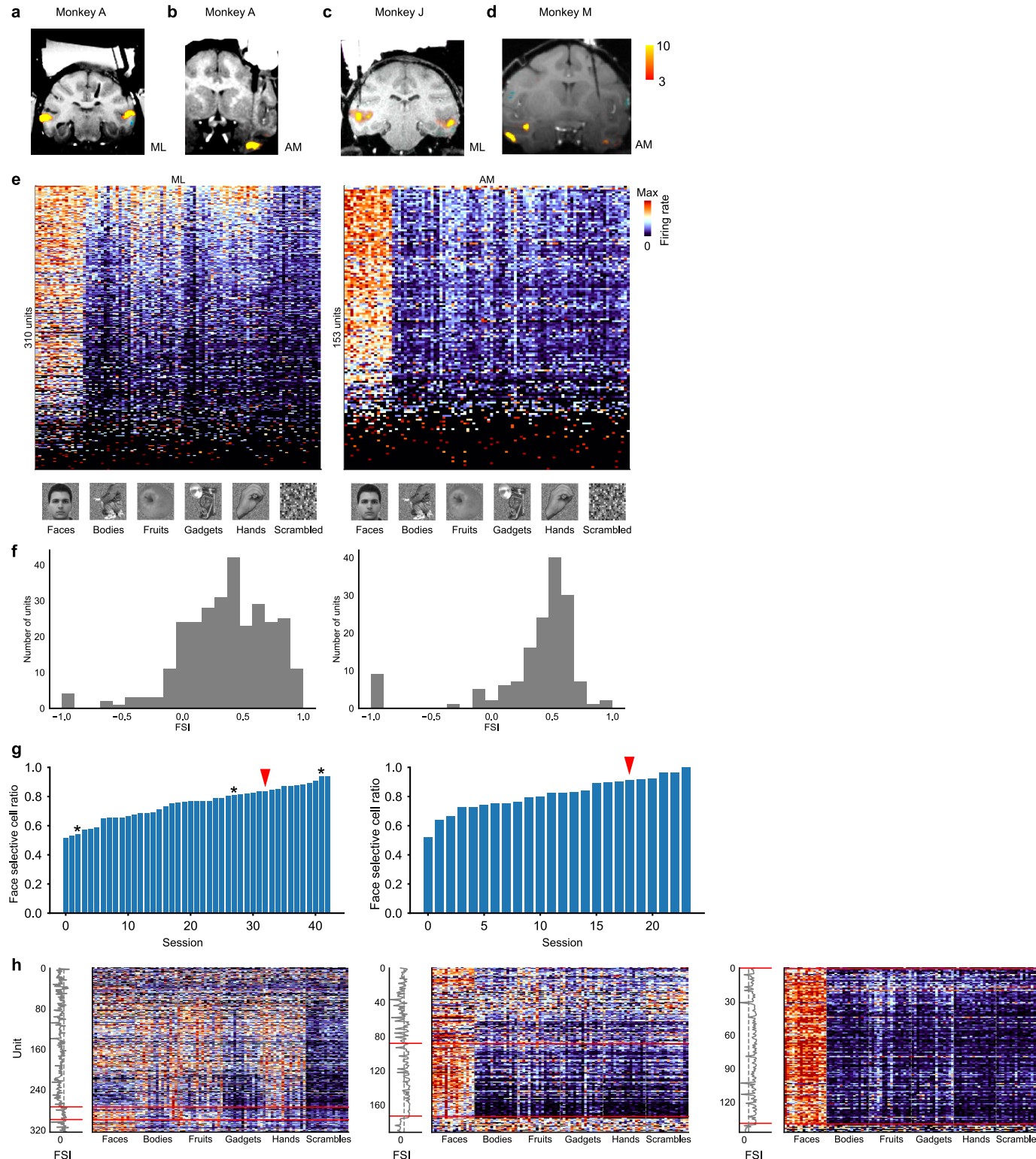

**Extended Data Fig. 1** | See next page for caption.

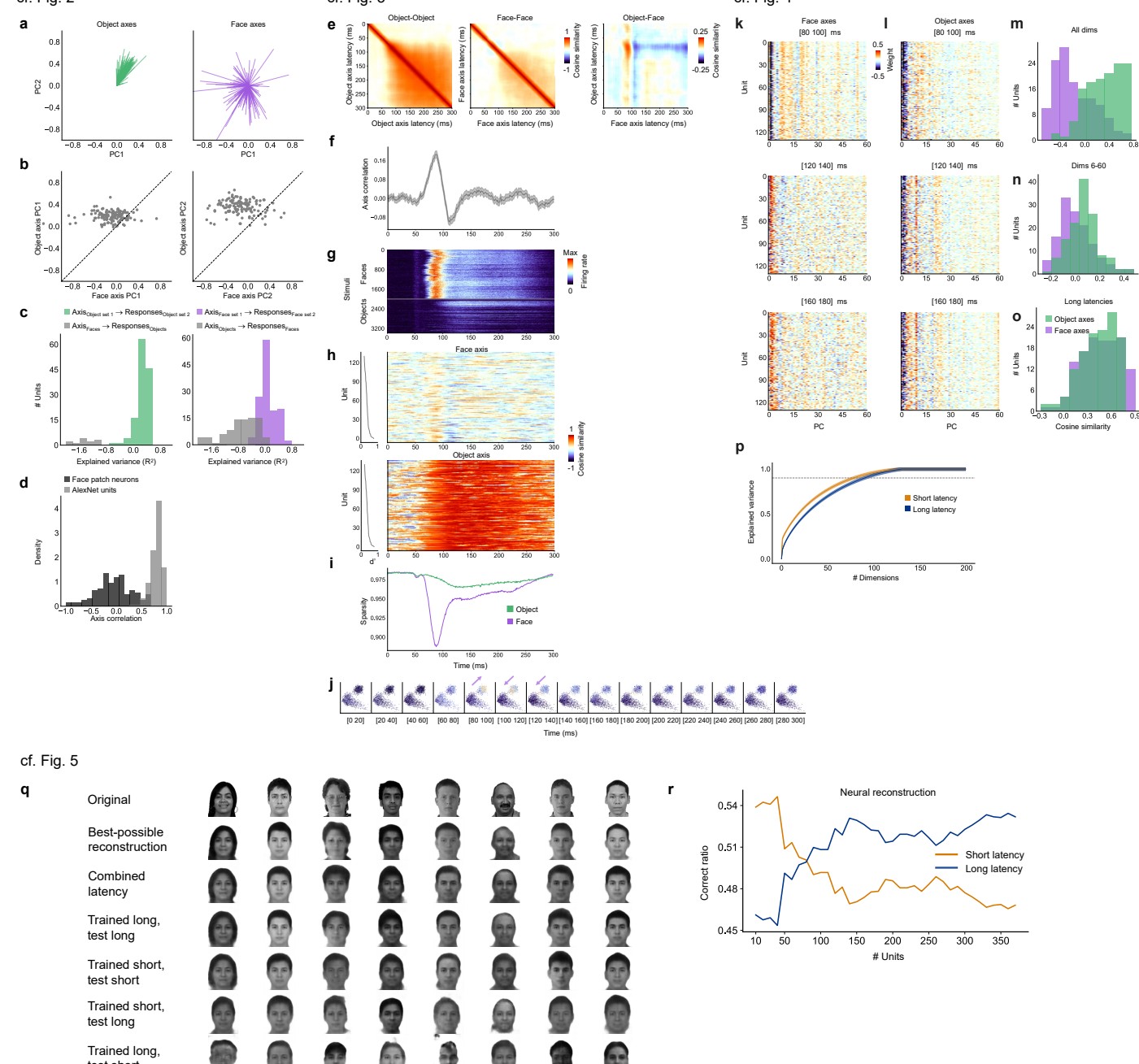

cf. Fig. 2

cf. Fig. 3

cf. Fig. 4

cf. Fig. 5

**Extended Data Fig. 2** | See next page for caption.

**Extended Data Fig. 2 | Main results computed for monkey J, ML. a**. Distribution of object (left, green) and face (right, purple) axes, projected onto the top two dimensions of the 60-dimensional (60-d) object space (N = 131 cells, only cells with $R^2 > 0$ on the test set for both object and face axes are included). Face and object stimuli in this experiment were presented with random interleaving. **b**. Scatter plots of object vs. face axis weights for PC1 (left) and PC2 (right). **c**. Cross prediction accuracies, same conventions as Fig. 2c. Units with $R^2 < -2$ are not shown. **d**. Histogram of correlations between the face and object axes for each cell in the 60-d object space for real (dark grey) and AlexNet (light grey) units, normalized by the correlation between axes fitted from two disjoint subsets of the same category. **e**. From left to right: matrices of mean cosine similarities across the population between (object, object), (face, face), and (face, object) axes for different pairs of latencies (N = 131 cells). **f**. Correlation between face and object axes as a function of time (diagonal values in the rightmost similarity matrix in (e)). **g**. Mean response time course to each face and object stimulus, averaged across cells and trials. **h**. Cosine similarity between the overall object axis for each cell (computed using a 50–220 ms window) and its time-varying face (top) and object (bottom) axes, sorted according to face selectivity $d'$ (left). **i**. Response sparseness as a function of time for faces and objects. **j**. Time-resolved scatter plots of face and object stimuli projected onto PC1 and PC2 of object space, colour coded by the mean response magnitude across the population (N = 131 cells) to each stimulus. Purple arrows indicate the approximate face-axis direction in each time window. **k**. Matrix of face axis weights in AlexNet face space for each cell, computed using a short (80–100 ms, top), long-early (120–140 ms, middle), and long-late (160–180 ms, bottom) latency window; same conventions as Fig. 4a. **l**. Matrix of object axis weights; conventions same as (k). **m**. Purple (green) distribution shows correlation coefficients across units between face (object) axis weights at short and long-early latency. Conventions as in Fig. 4c. Face axis weights were negatively correlated (two-sided one-sample Student's t-test, t(130) = −6.03, p = 1.62 × 10⁻⁸); object axis weights were positively correlated (two-sided one-sample Student's t-test, t(130) = 12.75, p = 1.20 × 10⁻²⁴). **n**. Same as (m), using higher face space dimensions 6–60 to compute axis correlation for each cell. Face axis weights were uncorrelated (two-sided one-sample Student's t-test, t(130) = 0.52, p = 0.60); object axis weights were positively correlated (two-sided one-sample Student's t-test, t(130) = 6.96, p = 1.47 × 10⁻¹⁰). **o**. Same as (m), using axis weights at long-early and long-late latencies (120–140 ms and 160–180 ms) and all 60 dimensions. Face and object axis weights were both positively correlated (face: two-sided one-sample Student's t-test, t(130) = 21.83, p = 2.56 × 10⁻⁴⁵, object: two-sided one-sample Student's t-test, t(130) = 22.39, p = 1.91 × 10⁻⁴⁶). **p**. Explained variance in z-scored population responses to faces at short (80–100 ms, orange) and long (120–140 ms, blue) latencies, as a function of the number of PCs. More dimensions were required to explain 90% of response variance at long compared to short latency (mean ± s.d.: 90.5 ± 4.53 for long vs. 78.8 ± 4.47 for short, two-sided paired Student's t-test, t(99) = −20.54, p = 1.81 × 10⁻³⁷). **q**. Reconstructions of faces using responses from different time windows. Same conventions as Fig. 5b. **r**. Face identification performance on faces reconstructed from neural activity in different time windows. Same conventions as Fig. 5d. Credits: face images in **q** are reproduced from the CVL Face Database[48] (http://lrv.fri.uni-lj.si/facedb.html), the Chicago Face Database[52] (chicagofaces.org) and MUCT[54] (www.milbo.org/muct).

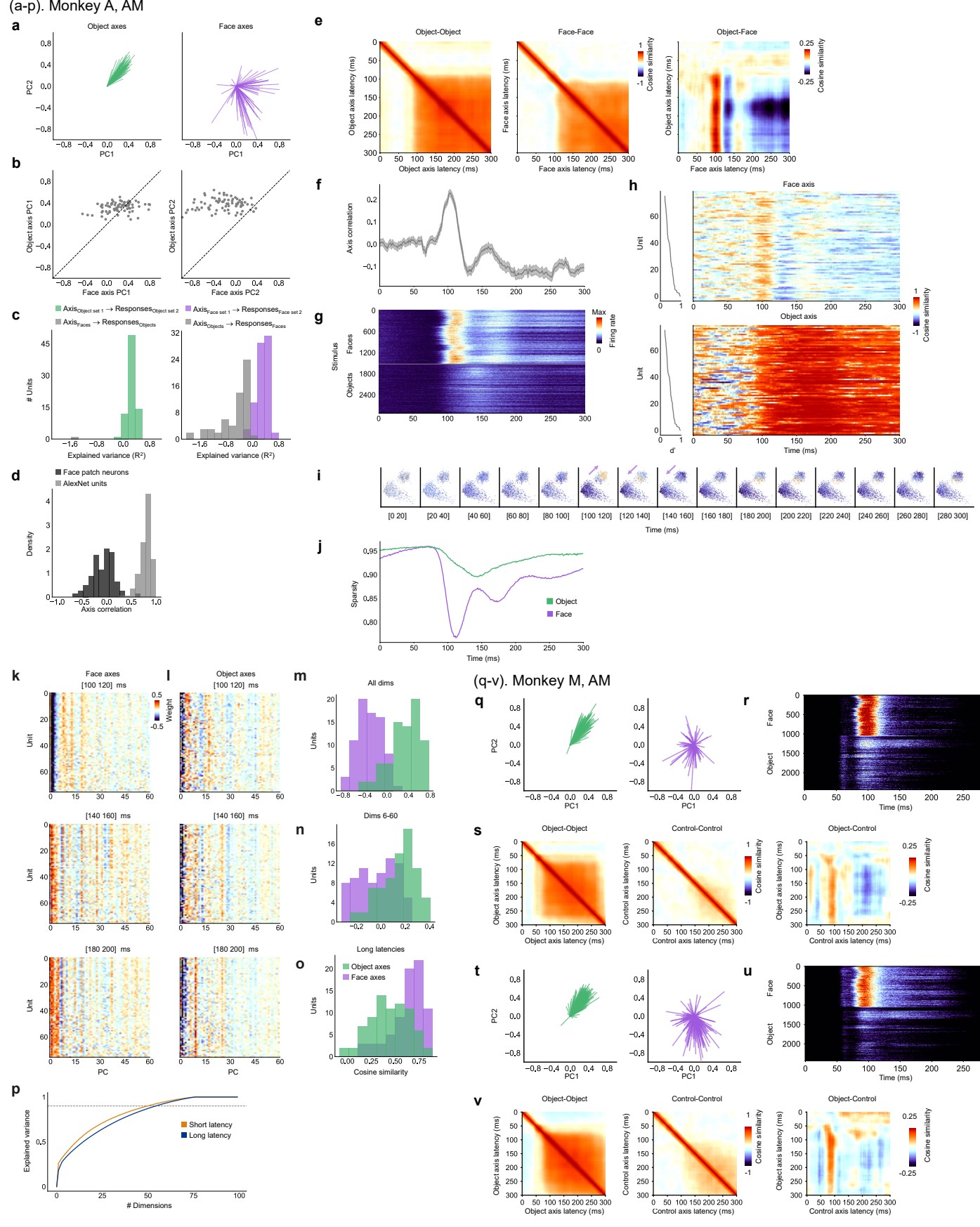

**(a-p). Monkey A, AM**

**(q-v). Monkey M, AM**

**Extended Data Fig. 3** | See next page for caption.

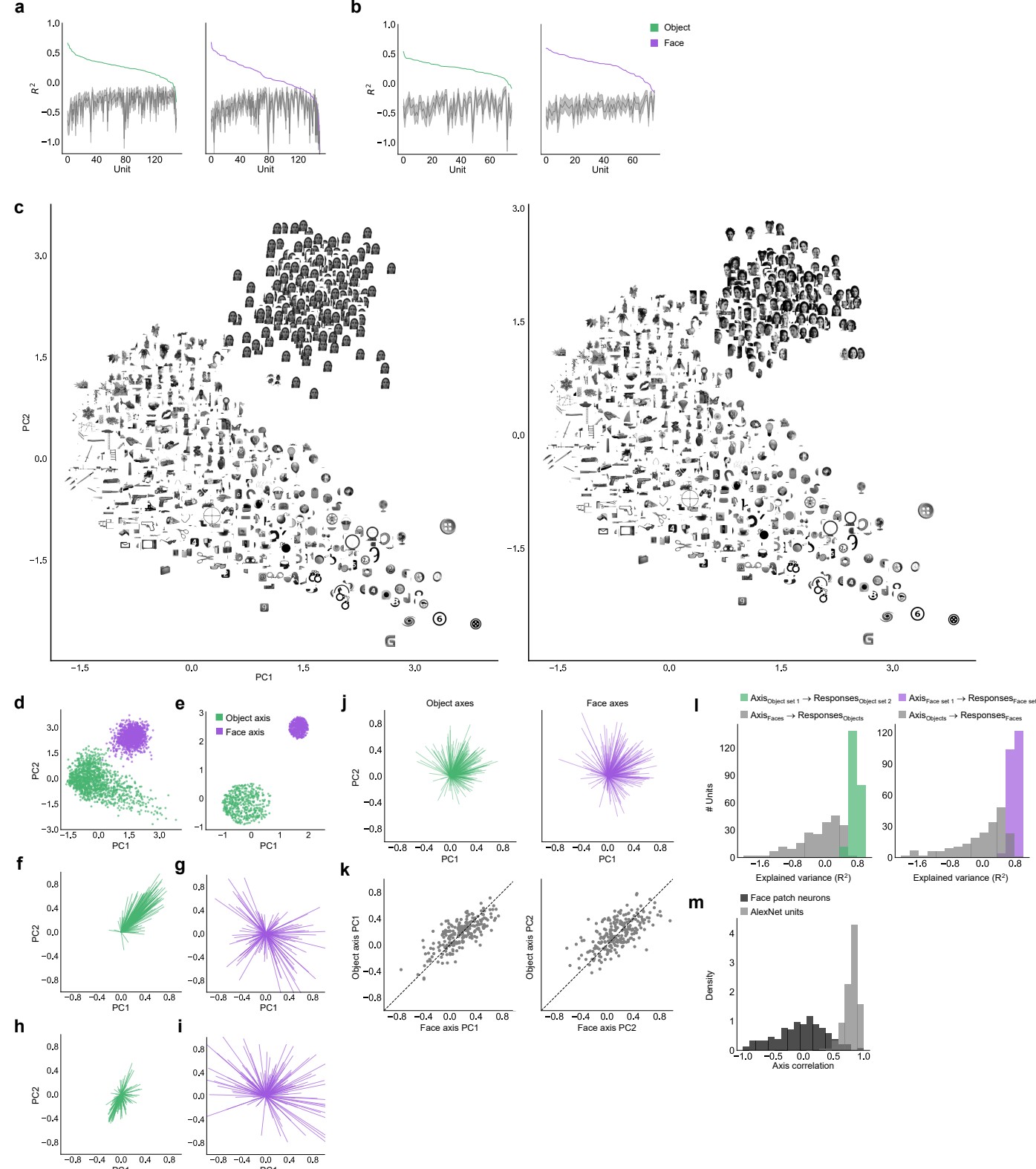

**Extended Data Fig. 4** | See next page for caption.

**Extended Data Fig. 4 | Quantification and controls of axis tuning. a**. Left: Explained variance of object responses for each ML cell using the object axis (green) together with explained variance for stimulus-shuffled data (grey; data are presented as mean ± s.d.). Right: Explained variance of face responses for each cell using the face axis (purple) and stimulus-shuffled control (grey). Across the population, 147/151 (151/151) units had significantly higher face (object) axis $R^2$ than a random shuffle. **b**. Same as (a) for AM; 58/76 (73/76) units had significantly higher face (object) axis $R^2$ than a random shuffle. **c**. Face and object images embedded in object space by projecting onto PC1 and PC2 subspace. Left: Face images used in the experiments are represented by a single template face placed at the corresponding embedding locations; we do not show the original face photographs because the dataset license does not permit reproduction of all images. Right: The same analysis repeated with a set of synthetic face images (Syn-Vis-v0; not shown to the monkey) to illustrate how face features distribute within this space. **d**. Distribution of the full training set of 1425 face (purple) and 1,292 object (green) stimuli projected onto the top two dimensions of the 60-d feature space. **e**. Distribution of a subset of face (purple) and object (green) stimuli chosen to make the face and object stimulus distributions maximally Gaussian. **f**. Distribution of object axes computed using responses to the subset of object images selected in (e), projected onto the top two dimensions of the 60-d feature space. **g**. Same as (f), for face axes. **h**. Same as (f), computed for stimulus-shuffled data. **i**. Same as (g), computed for stimulus-shuffled data. **j**–**l**. Same as Fig. 2a–c, computed using the 230 most face-selective units from AlexNet layer fc6 (FSI ranging from 0.2 to 0.46). **m**. Histogram of correlations between the face and object axes for each cell in the 60-d space for real (dark grey; N = 151 cells from the main ML session in monkey A) and AlexNet (light grey) units, same conventions as Extended Data Fig. 2d. Credits: object images in **c** are used with permission from ref. 6, Springer Nature Limited; face images in **c** are reproduced from the FEI database[47] (https://fei.edu.br/~cet/facedatabase.html) and Syn-Vis-v0[55] (https://huggingface.co/datasets/retowyss/Syn-Vis-v0).

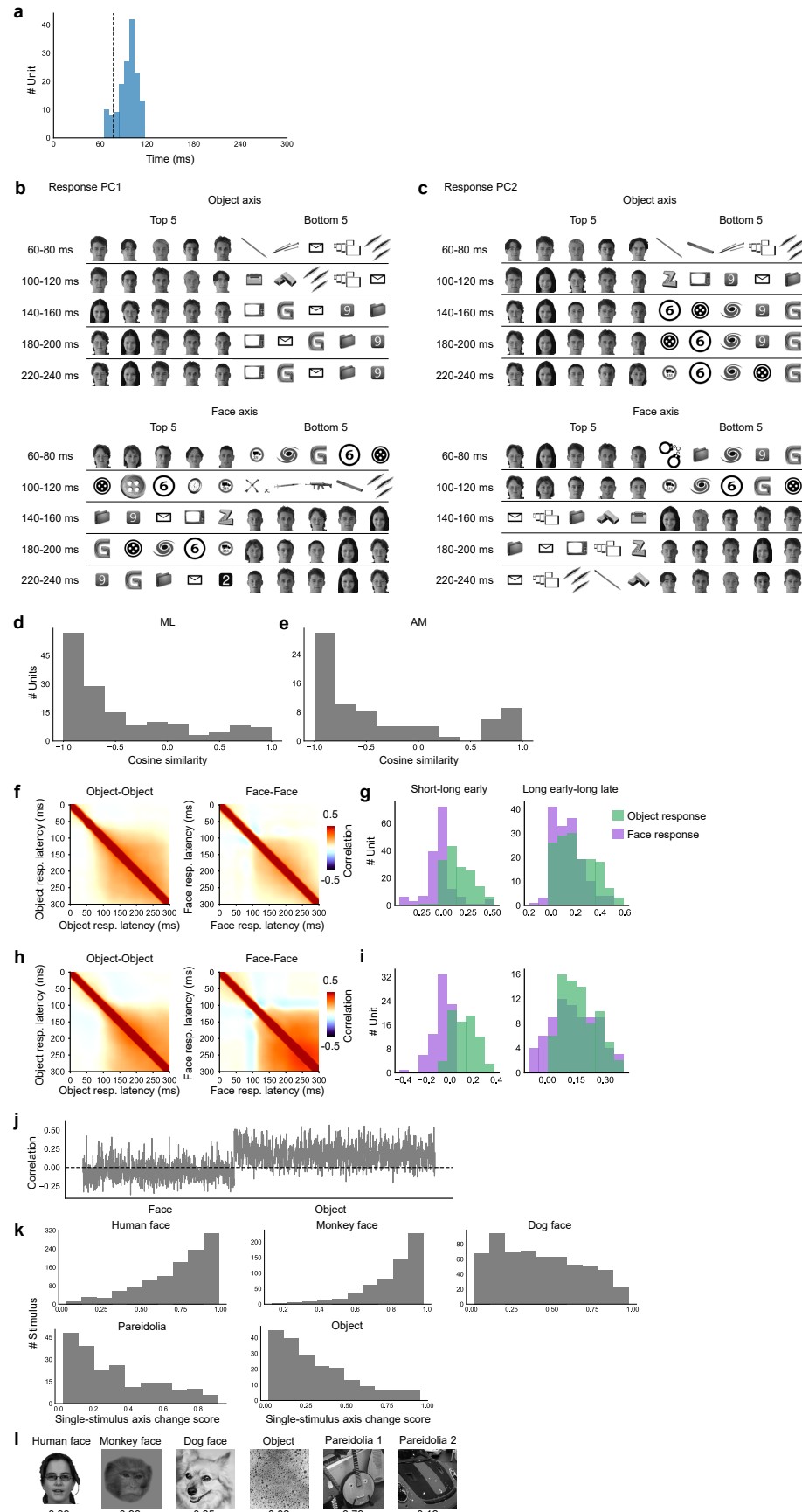

**Extended Data Fig. 5** | See next page for caption.

**Extended Data Fig. 5 | Figures related to axis dynamics. a**. Histogram showing the distribution of face axis switch times across cells in the ML face patch, defined as the first time point at which the face–object axis correlation drops below half of its peak value for each unit. The dashed line indicates the average response latency across the population (76.9 ms), computed by averaging the half-peak time of each unit's response. **b**. Images that project most strongly onto the extremes of the object (top) and face (bottom) axis of the first response component of ML at different time windows (Methods). For the face images, original identities are replaced with visually similar substitute images. **c**. Same as (b), but for the second response component of ML. **d**. Distribution of cosine similarities between face axis (140–160 ms) and the overall object axis (50–220 ms), computed using only their PC1 and PC2 components, for ML. Cells with cosine similarity less than −0.5 (corresponding to an angle of at least 120°) between the two axes were counted as reversing. **e**. Same as (d) for AM computed over 160–180 ms. **f**. From left to right: matrices of mean correlations across the population between (object, object) and (face, face) population responses for different pairs of latencies (N = 151 cells; only cells with $R^2 > 0$ on the test set for both object and face axes were used). **g**. Left: Purple (green) distribution shows correlations across units between face (object) responses at short and long-early latency. Right: Correlations across units between face (object) responses at long-early and long-late latencies. **h, i**. Same as (f), (g) for AM. **j**. Population response cross-time correlation of each individual stimulus calculated from the ML face cell responses. **k**. Histogram of single-stimulus axis-change score (Methods) to various categories of stimuli, including human faces, monkey faces, pareidolia, dog faces, and objects. **l**. Single-stimulus axis-change score for example human face, monkey face, dog face, object, and 2 pareidolia images. The scores are annotated below each image. The dog image is a substitute illustrative image. Credits: object images in **b** are used with permission from ref. 6, Springer Nature Limited; face images in **b** are reproduced from the CVL Face Database[48] (http://lrv.fri.uni-lj.si/facedb.html); face images in **l** are reproduced from MUCT[54] (www.milbo.org/muct).

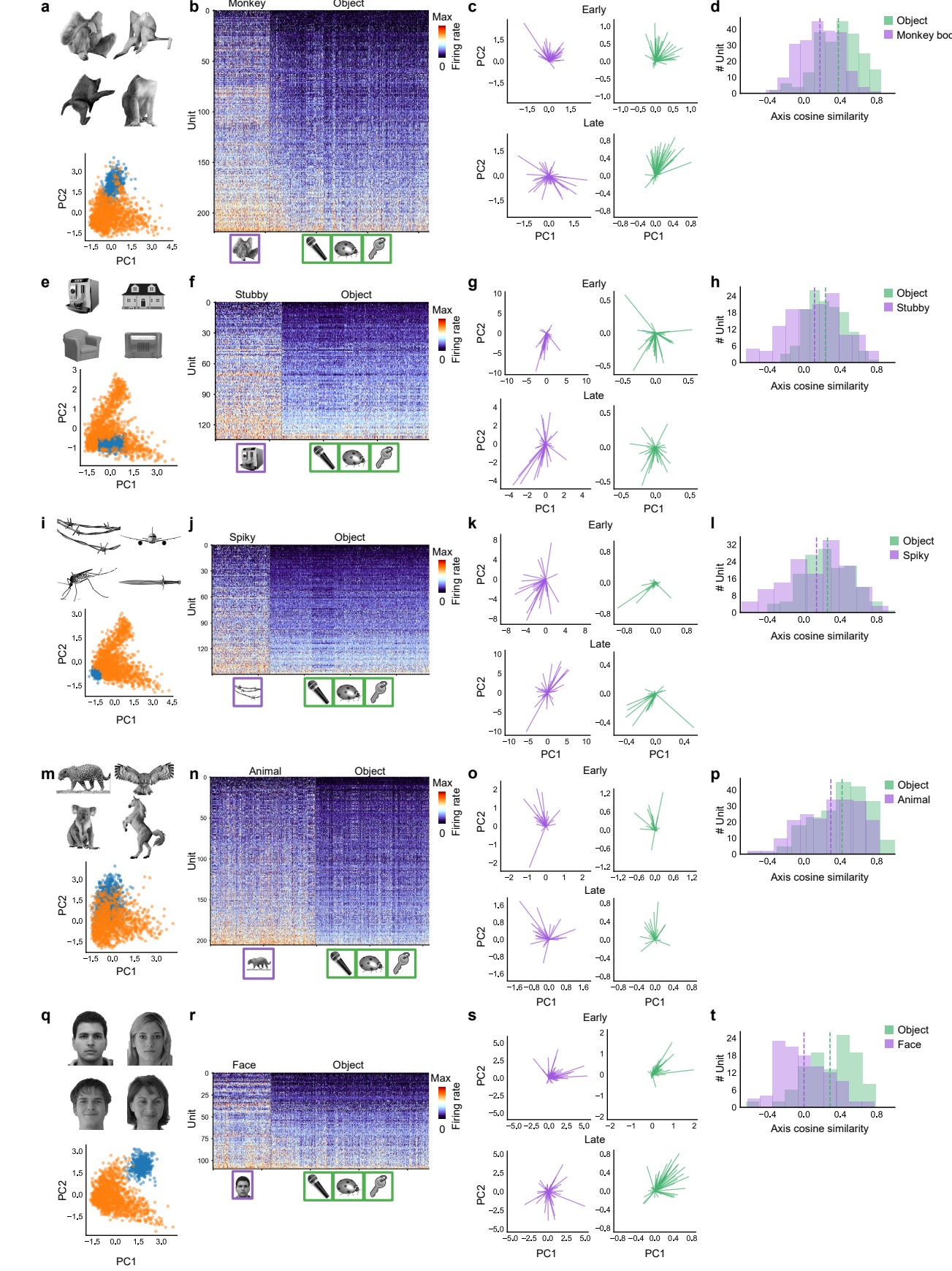

**Extended Data Fig. 6** | See next page for caption.

**Extended Data Fig. 6 | Axis dynamics of units outside face patches. a**. Top: Example images from the monkey-body image set (monkey-body images shown here are illustrative replacements; images from all other categories are the original stimuli shown to the monkey). Bottom: Distribution of features for monkey-body stimuli and other objects in PC1–PC2 of the object space. **b**. Time-averaged responses of all monkey-body-selective units (peak $d' > 0.1$, Methods) to monkey-body and non-monkey-body stimuli. **c**. Distribution of monkey-body (left, purple) and object (right, green) axes projected onto the top two dimensions of the 60-d object space, for early (top) and late (bottom) latency responses of monkey-body-selective units. (Only units with peak $d' > 0.2$ are used). **d**. Histogram of correlations of monkey-body (purple)/object (green) axes between early and late latencies for each unit. (The early and late time windows were defined individually for each unit: early window was centered on a unit's peak response time; late window began 20 ms after the early window.) **e**–**h**. Same as (a-d) for stubby-object-selective units. **i**–**l**. Same as (a-d) for spiky-object-selective units. **m**–**p**. Same as (a-d) for animal-selective units. For these units, we did not observe axis switch over time—their tuning axes remained stable. **q**–**t**. Same as (a-d) for face cells outside face patches. Credits: object images in **e, f, i, j, m, n** are used with permission from ref. 6, Springer Nature Limited; face images in **q** and **r** are reproduced from the CVL Face Database[48] (http://lrv.fri.uni-lj.si/facedb.html).

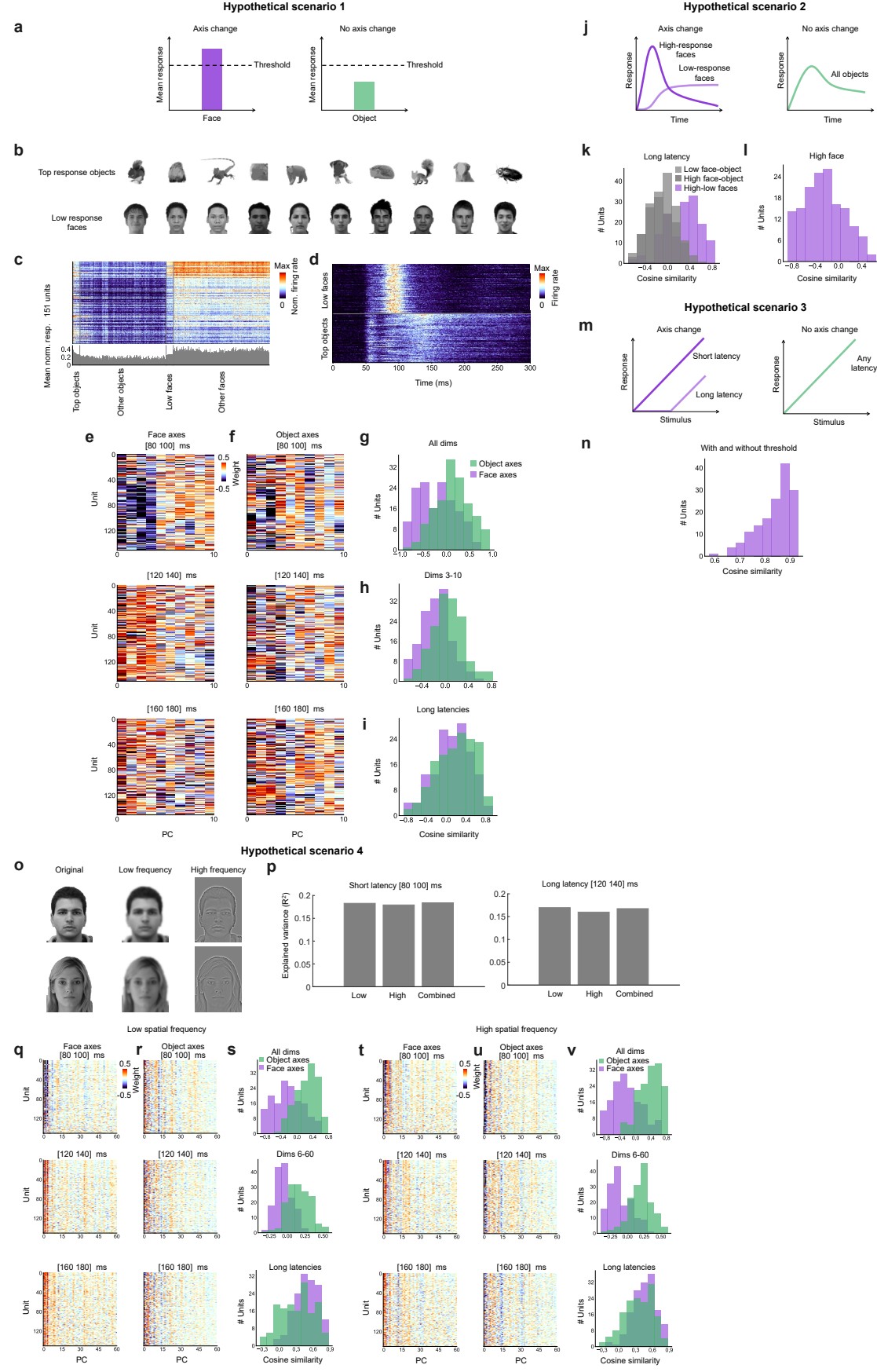

**Extended Data Fig. 7** | See next page for caption.

**Extended Data Fig. 7 | Testing three cell-intrinsic mechanisms and the contributions of low and high spatial frequency components to axis change.** **a**. Hypothetical scenario 1: a high mean response magnitude triggers axis change. **b**. The 10 most-effective object stimuli and 10 faces selected from the 75 least-effective faces; shown faces are limited to stimuli with reuse permission. **c**. Top: Mean responses averaged over 50–220 ms to the 100 most-effective non-face object stimuli, other objects, the 100 least-effective face stimuli, and other faces. Bottom: bar graph of the mean response to each stimulus across cells (N = 151 cells). **d**. Response time courses to the 100 least-effective faces (top) and 100 most-effective non-face objects (bottom), averaged across cells. **e**. Matrix of face axis weights in AlexNet face space for each cell, computed using responses to the least-effective faces in (c), same conventions as in Fig. 4a. Here, we only fit axes to the top 10 face space dimensions due to the smaller number of images used to fit the axes. **f**. Matrix of object axis weights in AlexNet face space for each cell, computed using responses to the most-effective objects in (c), same conventions as in (e). **g**. Purple (green) distribution shows cosine similarities across units between face (object) axis weights at short and long-early latency; conventions as in Fig. 4c. All dimensions (1–10) were used to compute cosine similarities shown here. Face axis weights were negatively correlated (two-sided one-sample Student's t-test, t(150) = −6.60, p = 6.72 × 10^{−10}); object axis weights were positively correlated (two-sided one-sample Student's t-test, t(150) = 4.85, p = 3.10 × 10^{−6}). **h**. Same as (g), using higher face space dimensions 3–10 to compute cosine similarities for each cell. Face axis weights were negatively correlated (two-sided one-sample Student's t-test, t(150) = −7.31, p = 1.50 × 10^{−11}); object axis weights were positively correlated (two-sided one-sample Student's t-test, t(150) = 2.55, p = 0.012). **i**. Same as (g), showing cosine similarities across units between face (object) axis weights at long-early and long-late latencies (120–140 ms and 160–180 ms), using all 10 dimensions. Face and object axis weights were both positively correlated (face: two-sided one-sample Student's t-test, t(150) = 4.13, p = 5.98 × 10^{−5}, object: two-sided one-sample Student's t-test, t(150) = 6.60, p = 6.58 × 10^{−10}). **j**. Hypothetical scenario 2: delayed responses to weaker stimuli lead to a change in tuning. To address this, we first identified the 50% most-effective and 50% least-effective face stimuli for each cell, determined by the mean response in the time window 80–100 ms. We then computed face axes separately using these two groups of stimuli, at both short (80–100 ms) and long (120–140 ms) latency. **k**. Purple distribution shows cosine similarities across units between face axis weights at long (120–140 ms) latency computed using the 50% most-effective and 50% least-effective faces for each cell; the two sets of face axes were positively correlated (two-sided one-sample Student's t-test, t(150) = 12.13, p = 4.89 × 10^{−24}). Grey distributions: cosine similarities between the two sets of face axes and the object axes at the same latency; the two sets of face axes were both negatively correlated with object axes (low face-object: two-sided one-sample Student's t-test, t(150) = −5.69, p = 6.57 × 10^{−8}; high face-object: two-sided one-sample Student's t-test, t(150) = −4.52, p = 1.27 × 10^{−5}). **l**. Distribution of cosine similarities across units between face axis weights at short (80–100 ms) and long (120–140 ms) latency computed using the 50% most-effective faces for

each cell; axes were negatively correlated (two-sided one-sample Student's t-test, t(150) = −8.70, p = 5.59 × 10^{−15}). The result rules out the possibility that axis reversal is driven solely by responses to non-optimal, low-contrast stimuli. **m**. Hypothetical scenario 3: cell-intrinsic adaptation leads to axis change−face cells increase firing threshold following a strong transient response. To investigate this possibility, we measured axis tuning of model cells encoding a single axis with and without application of a raised threshold. **n**. Distribution of cosine similarities across units between face axes mapped for the same units with and without application of a raised threshold (set to one s.d. above mean response); axes were significantly correlated (two-sided one-sample Student's t-test, t(150) = 146.67, p = 6.96 × 10^{−164}). **o−v**. Hypothetical scenario 4: a 'coarse to fine' progression in arrival of visual information to a face patch triggers the change in face axis. **o**. Two example faces (left) filtered to show low (middle) and high (right) spatial frequency components (Methods). **p**. Explained variance for short (left, 80–100 ms) and long (right, 120–140 ms) latency responses, using low spatial frequency features, high spatial frequency features, or a combined set of features. The number of features used to compute explained variance was the same in all three conditions (low: 120, high: 120, combined: 60 + 60). **q−s**. Axis weights and axis weight correlations computed using low spatial frequency features, same conventions as Fig. 4a−e. Face axis weights showed negative correlation between the two time windows (all dimensions: two-sided one-sample Student's t-test, t(150) = −7.47, p = 6.25 × 10^{−12}; higher dimensions: two-sided one-sample Student's t-test, t(150) = −4.56, p = 1.03 × 10^{−5}). Object axis weights were positively correlated between the two time windows (all dimensions: two-sided one-sample Student's t-test, t(150) = 11.27, p = 9.88 × 10^{−22}; higher dimensions: two-sided one-sample Student's t-test, t(150) = 12.69, p = 1.60 × 10^{−25}). Both face and object axis weights were positively correlated between the two long latency windows (face: two-sided one-sample Student's t-test, t(150) = 29.28, p = 6.66 × 10^{−64}; object: two-sided one-sample Student's t-test, t(150) = 13.70, p = 3.15 × 10^{−28}). **t−v**. Axis weights and axis weight correlations computed using high spatial frequency features, same conventions as Fig. 4a−e. Face axis weights showed negative correlation between the two time windows (all dimensions: two-sided one-sample Student's t-test, t(150) = −8.05, p = 2.33 × 10^{−13}; higher dimensions: two-sided one-sample Student's t-test, t(150) = −7.59, p = 3.11 × 10^{−12}). Object axis weights were positively correlated between the two time windows (all dimensions: two-sided one-sample Student's t-test, t(150) = 16.07, p = 2.02 × 10^{−34}; higher dimensions: two-sided one-sample Student's t-test, t(150) = 14.84, p = 3.09 × 10^{−31}). Both face and object axis weights were positively correlated between the two long latency windows (face: two-sided one-sample Student's t-test, t(150) = 26.80, p = 4.55 × 10^{−59}; object: two-sided one-sample Student's t-test, t(150) = 17.50, p = 4.51 × 10^{−38}). Credits: object images in **b** are used with permission from ref. 6, Springer Nature Limited; face images in **b** and **o** are reproduced from the CVL Face Database[48] (http://lrv.fri.uni-lj.si/facedb.html); face images in **b** are reproduced from the Chicago Face Database[52] (chicagofaces.org), the FEI database[47] (https://fei.edu.br/~cet/facedatabase.html) and MUCT[54] (www.milbo.org/muct).

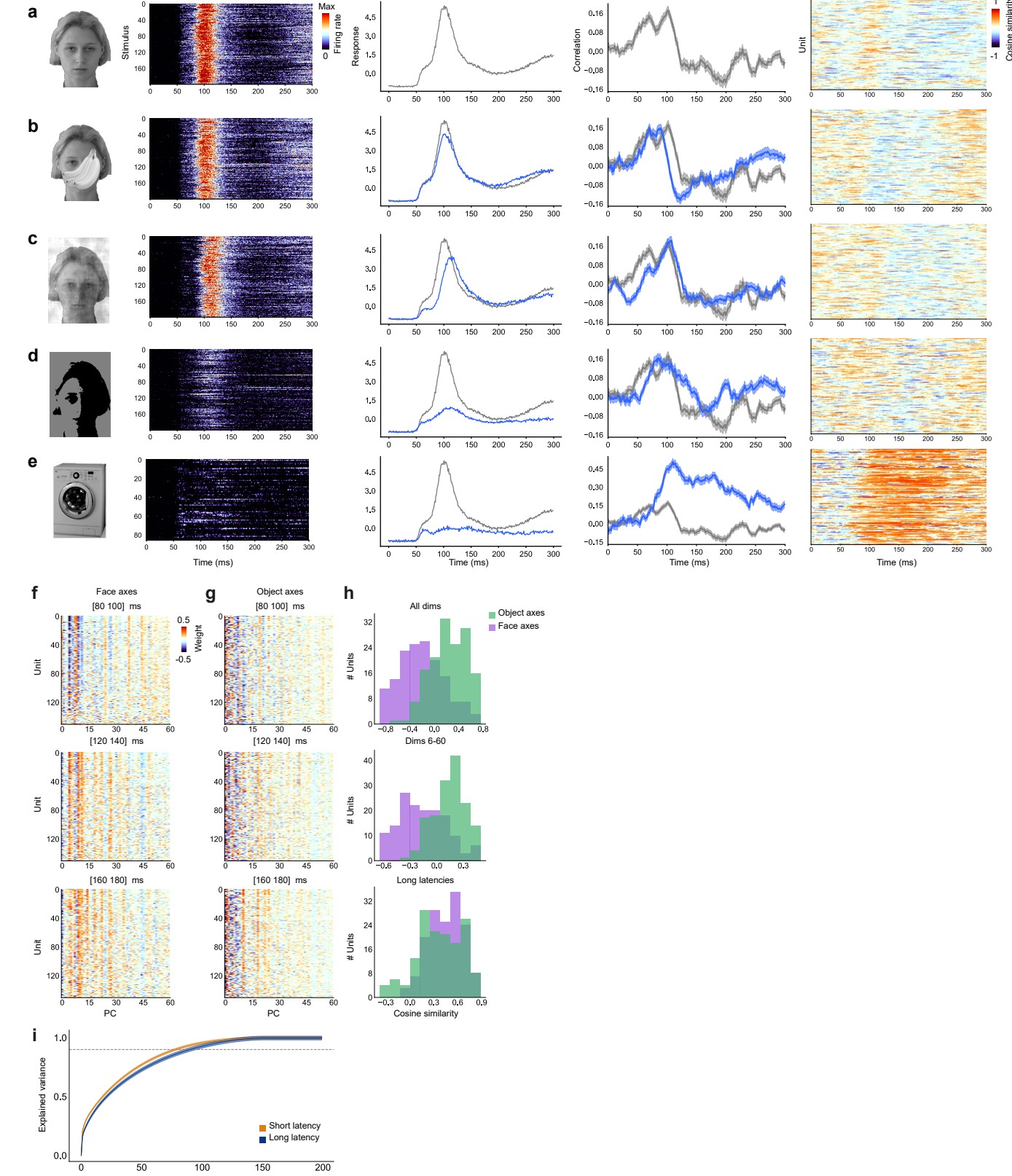

**Extended Data Fig. 8** | See next page for caption.

**Extended Data Fig. 8 | Axis dynamics for degraded faces and axis change using VGGFace2 face space. a**. Time course of responses and axis correlations for clear faces. First column, an example clear face image. Second column, PSTH of the cell-averaged response to each clear face. Third column, average response time course across all units and all clear face identities. Fourth column, time course of the correlation between the time-varying face axis and the cell's overall object axis, averaged across cells (same conventions as Fig. 3e). Fifth column, time courses of correlations between each unit's time-varying face axis and its overall object axis. **b–e**. Same as (a), but applied to degraded faces (occluded, noise-masked, and Mooney) and to non-face objects. In the third and fourth columns, the degraded-face or object time course is plotted in blue; the clear-face time course is plotted in grey as a control baseline (Methods). **f**. Matrix of face axis weights in VGGFace2 face space (Methods) for each cell, computed using short (80–100 ms, top), long-early (120–140 ms, middle), and long-late (160–180 ms, bottom) latency windows; same conventions as Fig. 4a.

**g**. Matrix of object axis weights; conventions same as (f). **h**. Top row: Purple (green) distribution shows correlation coefficients across units between face (object) axis weights at short and long-early latency. Conventions as in Fig. 4c. Face axis weights were negatively correlated; object axis weights were positively correlated. Middle row: Using higher face space dimensions 6–60 to compute correlation for each cell. Face axis weights were not significantly correlated; object axis weights were positively correlated. Bottom row: Using axis weights at long-early and long-late latencies and all 60 dimensions. Face and object axis weights were both positively correlated. **i**. Explained variance in population responses at short (80–100 ms, orange) and long (120–140 ms, blue) latencies, as a function of the number of PCs. More dimensions were required to explain 90% of response variance at long compared to short latency. Data are presented as mean ± s.d. Credits: object images in **e** are used with permission from ref. 6, Springer Nature Limited; face images are reproduced from the CVL Face Database[48] (http://lrv.fri.uni-lj.si/facedb.html).

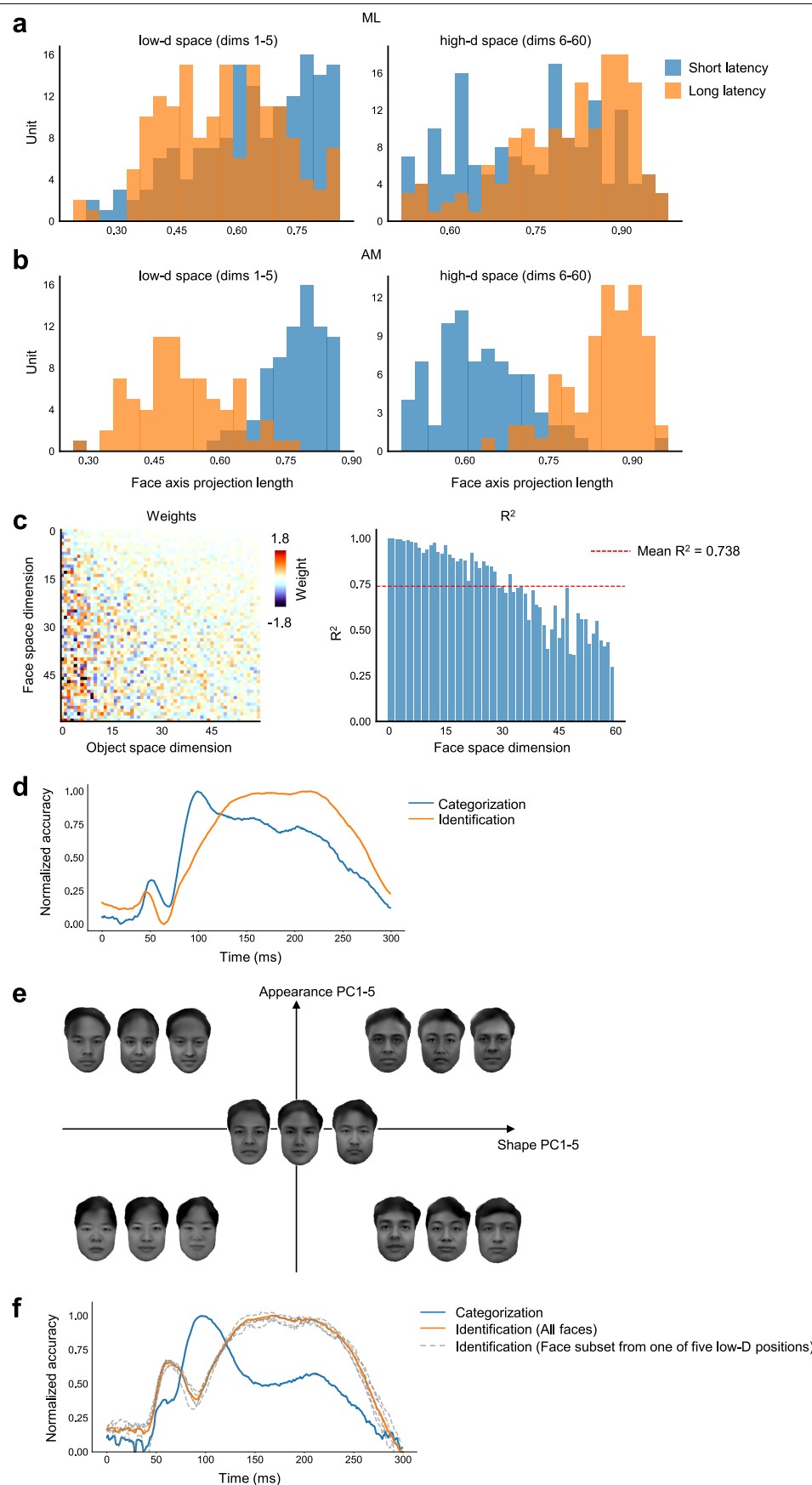

**Extended Data Fig. 9** | See next page for caption.

**Extended Data Fig. 9 | Decoding face category vs. identity over time.**
**a**. Projection lengths of short-latency (65–85 ms, blue) and long-latency (135–155 ms, orange) face axes onto face-space dimensions 1–5 (left) and 6–60 (right) in ML. **b**. Same as (a) in AM. **c**. Left: regression weights from linear models predicting each face-space dimension using all object-space dimensions. Lower face-space dimensions are predicted primarily by lower object-space dimensions, indicating that the low-dimensional structure of the two spaces is aligned. Right: explained variance ($R^2$) for predicting each face-space dimension. **d**. Time course of face categorization (face vs. object, blue) and face identity discrimination (orange), computed from population responses of all ML cells using a 20-ms sliding window (Methods). **e**. Schematic of the synthetic face generation. We constrained the first 5 shape PCs and the first 5 appearance PCs to take one of five discrete positions within the resulting 10-dimension subspace, while leaving all higher-dimensional features randomized (Methods). **f**. Time courses of face categorization (blue) and identity discrimination (orange) using the low-dimensionality-restricted synthetic face set. The five grey dashed lines show identity-discrimination time courses for each of the five face subsets in which the low-dimensional PCs were fixed (with higher-dimensional features randomized).

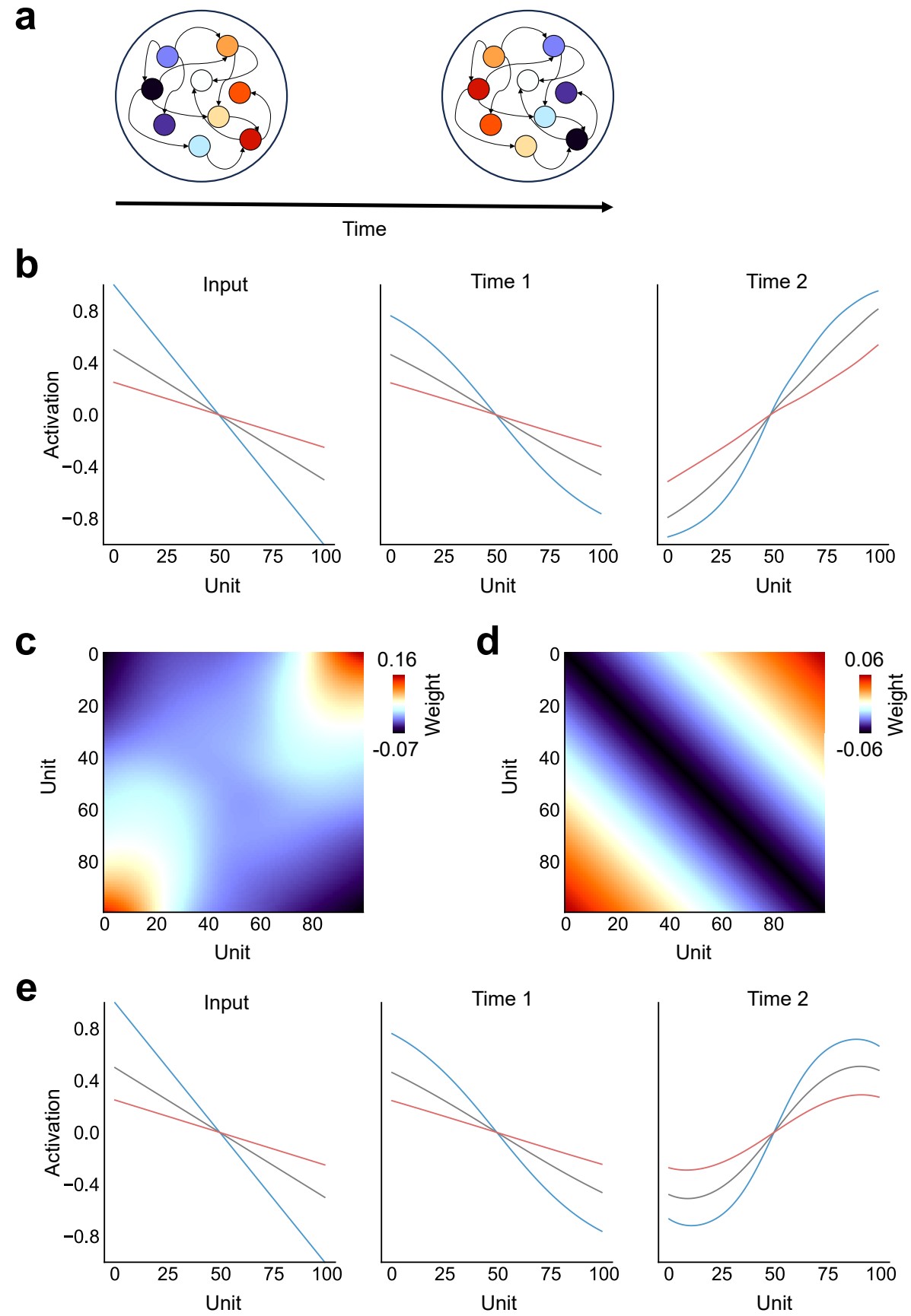

**Extended Data Fig. 10 | An RNN model of face axis reversal. a.** Architecture of a simple RNN trained to reverse a gradient in its input representation (Methods). **b.** Activity of each RNN unit over time for several input gradients (colours denote different input gradients), demonstrating stable gradient reversal (see Fig. 3e,f). **c.** Learned RNN weight matrix. The matrix is dominated by local inhibition and long-range excitation. **d.** Weight matrix of a second RNN explicitly incorporating only local inhibition and long-range excitation. **e.** Activity of each RNN unit over time for several different input gradients, for an RNN with weights as in (d).

# Reporting Summary

## Statistics

For all statistical analyses, confirm that the following items are present in the figure legend, table legend, main text, or Methods section.

| n/a | Confirmed | |
|---|---|---|
| ☐ | ☒ | The exact sample size (*n*) for each experimental group/condition, given as a discrete number and unit of measurement |
| ☐ | ☒ | A statement on whether measurements were taken from distinct samples or whether the same sample was measured repeatedly |
| ☐ | ☒ | The statistical test(s) used AND whether they are one- or two-sided *Only common tests should be described solely by name; describe more complex techniques in the Methods section.* |
| ☐ | ☒ | A description of all covariates tested |
| ☐ | ☒ | A description of any assumptions or corrections, such as tests of normality and adjustment for multiple comparisons |
| ☐ | ☒ | A full description of the statistical parameters including central tendency (e.g. means) or other basic estimates (e.g. regression coefficient) AND variation (e.g. standard deviation) or associated estimates of uncertainty (e.g. confidence intervals) |
| ☐ | ☒ | For null hypothesis testing, the test statistic (e.g. *F*, *t*, *r*) with confidence intervals, effect sizes, degrees of freedom and *P* value noted *Give P values as exact values whenever suitable.* |
| ☒ | ☐ | For Bayesian analysis, information on the choice of priors and Markov chain Monte Carlo settings |
| ☒ | ☐ | For hierarchical and complex designs, identification of the appropriate level for tests and full reporting of outcomes |
| ☐ | ☒ | Estimates of effect sizes (e.g. Cohen's *d*, Pearson's *r*), indicating how they were calculated |

*Our web collection on statistics for biologists contains articles on many of the points above.*

## Software and code

Policy information about availability of computer code

| Data collection | Electrophysiological signals were collected using NHP Neuropixels probes and recorded with the SpikeGLX system (Release_v20230815-phase30, HHMI/Janelia Research Campus, USA) and OpenEphys (version 0.6.4). |
|---|---|
| Data analysis | We listed all software used in the experiments and for anlaysis in the Methods section. Functional imaging data are prcoessed with Freesurfer (version 4.7.1) and FSL (version 5.0). Electrophysiological data was spike sorted using Kilosort (version 3 and 4). AlexNet (https://github.com/BVLC/caffe/tree/master/models/bvlc_alexnet original version from Krizhevsky A, Sutskever I, Hinton GE. "ImageNet Classification with Deep Convolutional Neural Networks." Advances in neural information processing systems. 2012.) Custom code written in Python 3.10, MATLAB R2023a was used for analysis, which is available from the lead corresponding author upon. reasonable request. |

For manuscripts utilizing custom algorithms or software that are central to the research but not yet described in published literature, software must be made available to editors and reviewers. We strongly encourage code deposition in a community repository (e.g. GitHub). See the Nature Portfolio guidelines for submitting code & software for further information.

## Data

Policy information about availability of data

All manuscripts must include a data availability statement. This statement should provide the following information, where applicable:
- Accession codes, unique identifiers, or web links for publicly available datasets
- A description of any restrictions on data availability
- For clinical datasets or third party data, please ensure that the statement adheres to our policy

> Datasets generated and/or analysed during the current study are available from the corresponding author upon request.

## Research involving human participants, their data, or biological material

Policy information about studies with human participants or human data. See also policy information about sex, gender (identity/presentation), and sexual orientation and race, ethnicity and racism.

| | |
|---|---|
| Reporting on sex and gender | N/A |
| Reporting on race, ethnicity, or other socially relevant groupings | N/A |
| Population characteristics | N/A |
| Recruitment | N/A |
| Ethics oversight | N/A |

Note that full information on the approval of the study protocol must also be provided in the manuscript.

# Field-specific reporting

Please select the one below that is the best fit for your research. If you are not sure, read the appropriate sections before making your selection.

☒ Life sciences  ☐ Behavioural & social sciences  ☐ Ecological, evolutionary & environmental sciences

For a reference copy of the document with all sections, see nature.com/documents/nr-reporting-summary-flat.pdf

# Life sciences study design

All studies must disclose on these points even when the disclosure is negative.

| | |
|---|---|
| Sample size | Sample sizes were maximized given the recording capacity of NHP Neuropixels probes (384 channels) and are consistent with previous studies using similar methods such as She, Liang, et al. "Temporal multiplexing of perception and memory codes in IT cortex" Nature (2024) and Bao, Pinglei, et al. "A map of object space in primate inferotemporal cortex" Nature (2020). |
| Data exclusions | All neurons encountered along the probe trajectory were recorded. For our main axis mapping analyses, we excluded non-visually responsive cells, non-face-selective cells, and cells that lacked significant axis tuning. |
| Replication | Results were replicated across at least 2-3 different animals and across 2 different face patches (ML and AM) in independent recording sessions. |
| Randomization | Stimuli were presented in a randomized order. |
| Blinding | Investigators were not blinded to experimental groups (face vs non-face object images) because the nature of this study requires separate models to be built for face and object images. |

# Reporting for specific materials, systems and methods

We require information from authors about some types of materials, experimental systems and methods used in many studies. Here, indicate whether each material, system or method listed is relevant to your study. If you are not sure if a list item applies to your research, read the appropriate section before selecting a response.

## Materials & experimental systems

| n/a | Involved in the study |
|-----|----------------------|
| ☒ | ☐ Antibodies |
| ☒ | ☐ Eukaryotic cell lines |
| ☒ | ☐ Palaeontology and archaeology |
| ☐ | ☒ Animals and other organisms |
| ☒ | ☐ Clinical data |
| ☒ | ☐ Dual use research of concern |
| ☒ | ☐ Plants |

## Methods

| n/a | Involved in the study |
|-----|----------------------|
| ☒ | ☐ ChIP-seq |
| ☒ | ☐ Flow cytometry |
| ☐ | ☒ MRI-based neuroimaging |

# Animals and other research organisms

Policy information about studies involving animals; ARRIVE guidelines recommended for reporting animal research, and Sex and Gender in Research

| | |
|---|---|
| Laboratory animals | Three male rhesus macaques (Macaca mulatta) of 8-12 years old were used in this study. |
| Wild animals | The study did not involve wild animals. |
| Reporting on sex | Only male animals were used. |
| Field-collected samples | The study did not involve field-collected samples. |
| Ethics oversight | All procedures conformed to local and US National Institutes of Health guidelines, including the US National Institutes of Health Guide for Care and Use of Laboratory Animals. All experiments were performed with the approval of the UC Berkeley Animal Care and Use Committee. |

Note that full information on the approval of the study protocol must also be provided in the manuscript.

# Plants

| | |
|---|---|
| Seed stocks | *Report on the source of all seed stocks or other plant material used. If applicable, state the seed stock centre and catalogue number. If plant specimens were collected from the field, describe the collection location, date and sampling procedures.* |
| Novel plant genotypes | *Describe the methods by which all novel plant genotypes were produced. This includes those generated by transgenic approaches, gene editing, chemical/radiation-based mutagenesis and hybridization. For transgenic lines, describe the transformation method, the number of independent lines analyzed and the generation upon which experiments were performed. For gene-edited lines, describe the editor used, the endogenous sequence targeted for editing, the targeting guide RNA sequence (if applicable) and how the editor was applied.* |
| Authentication | *Describe any authentication procedures for each seed stock used or novel genotype generated. Describe any experiments used to assess the effect of a mutation and, where applicable, how potential secondary effects (e.g. second site T-DNA insertions, mosiacism, off-target gene editing) were examined.* |

# Magnetic resonance imaging

## Experimental design

| | |
|---|---|
| Design type | Block design |
| Design specifications | During the fMRI experiment, stimuli were presented in 24 s blocks at an interstimulus interval of 500 ms |
| Behavioral performance measures | Monkey's eye position was monitored using an infrared eye tracking system (Eyelink). Juice reward was delivered every2–4 s if fixation was properly maintained (within a 3.4 degree square window). |

## Acquisition

| | |
|---|---|
| Imaging type(s) | Functional and anatomical imaging |
| Field strength | 3 Tesla |
| Sequence & imaging parameters | T1-weighted anatomical volumes were measured with MP-RAGE sequence( TR 2,300 ms; IR 1,100 ms; TE 3.37 ms; 0.5mm isotropic voxels) . EPI volumes were acquired in an AC88 gradient insert (Siemens) TR was 2000 ms,TE was 17 ms,voxels were $1 \times 1 \times 1$ mm with an no gap between slices. Matrix size was (96, 96, 64) (read [x], phase [y], slice [z]), thefield of view was $96 \times 96$ mm in-plane. Flip angle was 80°. |
| Area of acquisition | Whole brain |

Diffusion MRI ☐ Used ☒ Not used

## Preprocessing

| | |
|---|---|
| Preprocessing software | Analysis of functional volumes was performed using the FreeSurfer Functional Analysis Stream(Massachusetts General Hospital). Volumes were corrected for motion and undistorted based on acquired field map. |
| Normalization | To concatenate different scans, each voxel's responses were percentage transformed with 0 mean value |
| Normalization template | We did not normalize any imaging data into template. All the analysis were done in the single subject's original space. |
| Noise and artifact removal | We remove the linear or quadratic trends in the time series. |
| Volume censoring | Motion noises were removed by putting the motion parameters as the regressors in the GLM analysis. |

## Statistical modeling & inference

| | |
|---|---|
| Model type and settings | The analysis used only first-level analysis. |
| Effect(s) tested | We ran t-tests between different conditions within each single subject. |

Specify type of analysis: ☒ Whole brain ☐ ROI-based ☐ Both

| | |
|---|---|
| Statistic type for inference<br><br>(See Eklund et al. 2016) | All the analyses were done using voxel-wise inference. |
| Correction | We did not apply any multiple-comparison correction in the fMRI imaging analysis. We set p value at 0.001. |

## Models & analysis

| n/a | Involved in the study |
|---|---|
| ☒ | ☐ Functional and/or effective connectivity |
| ☒ | ☐ Graph analysis |
| ☒ | ☐ Multivariate modeling or predictive analysis |

