## [Peer Review File · Nature]

Rapid, concerted switching of the neural code in inferotemporal cortex

Corresponding Author: Professor Doris Tsao

Version 0:

Reviewer comments:

Referee #1

(Remarks to the Author)

In this paper the authors contrast two prevalent theories about face perception: the domain specific theory, with specialized mechanisms dedicated to faces, and the domain general theory, with computations not specific to faces. To address this issue, they recorded neurons in monkey face patches AM and ML, while the animals viewed pictures of faces and objects. The key, novel finding is that, rather than having different population of neurons being involved in different aspects of face processing, there is a dynamic and rapid switch of the neurons' code, going from detection, at first, to face discrimination later on. They support this claim by showing the reversal of the projection axes representing face stimuli at about 100 ms (with each axis determined by the projection of layer fc6 of an AlexNet network into the 2 largest principal components), something that was not present for the object stimuli. Overall, I find the paper very interesting not only for colleagues working on face perception but for those interested in neuronal coding in general, posing the intriguing hypothesis that, instead of having sets of neurons with a given and fixed function, the neurons' code may change dynamically to perform different functions. This is the type of papers I would personally like to see published in Nature, but I think the authors should consider the comments below to improve the presentation of the results, which is at times a bit cluttered and convoluted, deviating the attention from the main message.

1. It is surprising that the authors find qualitatively similar results in AM and ML. As far as I remember from previous works of the authors, responses in AM are much more selective and, given that the axes switch described in this paper comes together with an increase in sparseness (but see below), one would expect different findings in these two areas. I suggest the authors to at least briefly address this point in the discussion.

2. It makes sense that the proportion of face neurons is lower than in previous studies due to the use of neuropixels (given that not all recording sites are at the peak of the face activations). The percentage of face responsive neurons is actually quite close to the one recently reported in analogous face processing areas in humans (Quiari Quiroga et al. Nat Comm. 14: 5661; 2023) and in that study, as here, due to clinical-experimental constraints, the recordings were not necessarily at the peak of the face activations. To encourage comparisons across species (and with the authors previous studies), I think it would be useful to quantify the proportion of face responsive neurons as a function of their location with respect to the peak of the face activations. Ideally, this could be a plot showing such proportion as a function of distance to the peak activation, However, I guess there might not be enough responses to produce such a plot, but at least the authors could quantify the proportion of face selective responses at the peak of the face activations vs. the one found in more distant recording sites.

3. Related to the previous point, another interesting analogy with human studies is that in the abovementioned paper, with recordings in the human midfusiform gyrus, using a decoding approach it was also found that face discrimination was preceded by face detection (and also by face-familiarity detection), although in this case there was not a finding of a switch of representation axes, which is the novel finding of this study. I would encourage the authors to mention the human study to foster comparisons across species.

4. Do the authors expect what they show for face cells here to also apply to the object cells they have described in a previous study (Nature 2020)? That means, would they expect that object cells switch from object detection to object discrimination? I am not asking to add such an analysis, but this could be mentioned in the discussion, supporting their point of a general switching mechanism of single neuron responses.

5. Lines 215-216: Why does the object axis become stable at 60 ms, while the face access does it much later, at 118 ms? Is it because face neurons detect the picture is not a face (and won't be further processed as it is the case with the face pictures)?

6. It is very difficult to see anything in the inset of Figure 3f. This could be shown better in the supplementary information.

7. The claim that the axes reversal is accompanied by a sparsening of the face cell responses makes sense and is very important, but, in my view, is poorly supported by the presented data. In fact, the only result related to this is depicted in Figure 3h, which actually shows the opposite effect (a decrease of face selectivity at the time of the axes reversal – i.e. at about 100 ms). If the authors want to keep this claim, it should be better supported – explaining what's happening in Figure 3h, showing a few examples and/or quantifications of such sparsening, etc.

8. Section "Face axis reversal in low dimensions is accompanied by emergence of diverse new tuning to higher dimensions of face space". I find this section to be confusing. It could eventually be skipped and some of the information there incorporated into the other sections or moved to the Supplementary Information. First, the only evidence of new tuning I could find in this section is a relatively small change in the number of PCs needed to explain the variance of the responses (Fig. 4f, going from 82 to 96 dimensions to explain 90% of the variance). The section starts with some modelling results that in my view are presented a bit out of the blue and do not offer any direct evidence supporting the authors' claim. I have already mentioned my concerns about the claim of an increase of sparseness at the time of the axis switch, and I don't see any other good evidence of a new tuning, compared with what was presented in the previous sections. The last 3 pages of this section describe different controls, which are important (particularly the one showing that results are not due to larger responses to faces) but that can just be mentioned and described in detail in the Supplementary Information (or in a separate section about controls to rule out alternative hypothesis, instead of a section describing a new tuning, as it is currently done). Furthermore, the controls are presented as supporting lateral inhibition, but there's no direct evidence of lateral inhibition in this paper (just a modelling result). I see these as support to the claim of an axis reversal, ruling out alternative explanations, but not as supporting the presence of lateral inhibition. Finally, scenario two (lines 357 and below) is not clear to me. Would be good to explain more clearly why this scenario, not being an axis reversal, will look as such.

9. Related to the previous point, it is not clear what panels 4h-k show.

10. The decoding approach used by the authors, seems unnecessarily convoluted to me. First, they reconstruct faces from the neurons' responses, as in Chang and Tsao's Cell 2017 paper. However, in that paper the accuracy achieved with the face reconstructions is truly remarkable, which does not seem to be the case here, where the authors make relatively vague and subjective statements (e.g. talking about reasonable reconstructions) and the quantification is made by comparing to the best possible reconstruction of the faces (something that depends on the model) and not comparing to the original face pictures. Does this offer any advantage compared to just having two decoders with a sliding window (instead of taking two time intervals) comparing the decoding of face detection and face discrimination? If yes, this should be explained in methods and if not, the authors may consider using the more straightforward approach mentioned here.

11. Fig. d-e. Why does the reconstruction accuracy decrease with the number of neurons for the short-latency window. This is very counterintuitive to me (in general, the more neurons, the better accuracy) and it is not discussed in the paper.

12. Lines 490-494. I don't see why category information should necessarily be expected to rise before identity information. This is the case if neurons are tuned to large differences, but not if they encode more detailed and specific (face) features.

13. Lines 504-509. In this paragraph the authors challenge the feedforward processing view of invariant object (face) recognition, but I believe no much can be said in this respect, because this paper does not deal with invariance – i.e. the authors show a change of code from face detection to face discrimination but this does not necessarily apply to how invariant recognition is processed by the brain.

Minor points:

14. Line 545. "Indeed we found..." Why we? Was any of the authors involved in the study of reference 54?

15. Line 238-240. Please edit.

Summarizing, in my view this is a very interesting paper, proposing the novel idea of a dynamic change of the neurons' code to encode different aspects of the stimuli (face detection and face discrimination). This claim is well supported by the data and main results, but I believe the authors should improve the presentation to better convey this message and increase the impact of the paper.

Referee #2

(Remarks to the Author)

Shi et al. Rapid, concerted switching of the neural code in inferior temporal cortex

In this study, the authors record neuronal activity in two regions within the macaque monkey inferior temporal cortex. These regions are purported to represent “faces”, according to functional magnetic resonance imaging (fMRI) measurements. The authors use photographs of “faces” and “non-faces” and fit the neuronal responses using the activations of an old neural network (Alexnet). Based on the weights of these fits, the authors argue that the representation of faces is different from that of non-faces and that this representation rapidly changes over time after stimulus presentation.

(1) This study follows a traditional decades-old approach whereby investigators decide on an arbitrary set of stimuli and present those images to a monkey. Such studies have multiple intrinsic biases. In the context of the current work, some of the questions include:

What is a “face”? What is a “non-face”? Are human “faces”, monkey “faces”, dog “faces”, coin “faces” all the same? How controlled are the stimuli for possible confounding factors (the list is long: contrast, textures, intermediate features in neural networks, other features of bottom-up or even top-down saliency)? How many “faces” and “non-faces” are shown? Conclusions depend strongly on these arbitrary decisions but the current study lacks any type of even minimal controls, definition, let alone justification for any of these choices or evaluation of how conclusions depend on these arbitrary choices.

(2) To circumvent these challenges, several investigators have developed unbiased and systematic ways of studying tuning properties in visual cortex. Notable examples include Bashivan et al Science 2019, Walker et al Nature Neuroscience 2019, and Ponce et al Cell 2019. There are differences among these approaches, but they consistently show that the stimuli that trigger strong responses in neurons along the ventral visual cortex in macaques are not quite “faces”, or “chairs”, but rather complex feature combinations. Of note, Ponce et al 2019 record from similar areas to the ones examined in the current paper. Additionally, subsequent studies by Bardon et al PNAS 2022 showed that humans do not perceive the stimuli derived in such an unbiased manner as “faces”.

(3) There are multiple problems that pervade the literature claiming that “faces” are special. These problems have been extensively discussed in the literature. I will only focus here on problems that are particularly relevant for the current study.

3.1 First, there is no clear definition of what a “face” is. From caricatures, to cartoon movies depicting cars with faces, to pareidolia, to the face of an ant to the face in a coin, to clocks that have shapes like faces, to many other versions, the definition remains ambiguous and it is unclear whether any conclusions would extrapolate to such stimuli. There have been other works that study some of these images but none of them are included as controls in the current work, casting a doubt on whether the current study can really draw any conclusion about “faces”.

3.2 Given problem 3.1, to what extent are the results—two discrete neural codes and a binary, gate-like switching between the codes—generalizable to more ambiguous stimuli? The study worked hard to distinguish the ill-defined “face” category. (I could not help noticing the peculiar animal images with faces blocked out.) Besides the pareidolia images and synthetic stimuli mentioned above, there are also studies showing that single eyes can drive the activity of neurons in the PL face patch (Issa & Dicarlo, J Neurosci. 2012). Should “face” neurons respond categorically to all these ambiguous stimuli as either non-faces or faces? Or, does the neural code depend on the stimulus in a continuous, not binary fashion? Or, more likely, the neural code represents general features that should not be separated as “faces” and “non-faces”.

3.3 In addition to the “low-level” problems alluded to in question (1), face photographs are very homogeneous within image space. One can debate about how to define such homogeneity. For the sake of argument, one could define this homogeneity as the standard deviation between stimuli in pixel space, or even better, in a “high-level” feature space like conv5 or fc6 in the Alexnet layers that the authors study (any other more modern neural network would work as well for this definition). This homogeneity in stimulus space is likely to drive many of the conclusions in the current study. Consider another group of stimuli (e.g., chairs). Imagine that you match the degree of homogeneity in this group (chairs) to the one for “faces” in the current study. Imagine matching “low-level” image properties as well. Imagine that you also match the large bias in the number of stimuli, that is, use about 1,500 chairs and a heterogenous set of 1,500 non-chairs. Then select those neurons that have $d' > 0.2$ for chairs versus non-chairs. Under these experimental conditions, one would likely obtain the same results for chair neurons. One could go on to do this for tables, rhinoceros, clouds, or any group of interest.

(4) These issues are exacerbated by the poor block designs. Block designs are nefarious (see for example Li et al IEEE TPAMI 2021 for one of many example discussions on these issues), introducing confounds related to adaptation, baseline changes, attentional changes, potential learning effects, and many other issues that are hard to control. Stimulus randomization is critical when attempting to make any statements about different stimuli when there is no reason to block stimuli.

(5) Another difficulty in the interpretation of the current results is that most of the conclusions are seen through the lens of the weights in the fitting procedure. That is, the conclusions are not directly about the tuning properties of neurons in cortex, but rather about the weights in this fitting procedure. Consider a model that perfectly fits the neuronal responses across any possible stimuli. Then, it seems that one can derive interesting conclusions about the neuronal responses from studying the weights in the model. Now consider a model that only poorly explains neuronal responses and does not extrapolate across novel stimuli. Then conclusions derived from such a fitting procedure would make little sense. The current situation is neither one extreme nor the other. The model partially fits neuronal activity. Therefore, at best, the conclusions derived from these

weights at best can be interpreted as partially reflecting neuronal responses. The authors themselves acknowledge similar concerns: AlexNet is not an optimal model (lines 98, 566), AlexNet itself is a highly nonlinear “blackbox” model (lines 569–570), etc.

(6) To the extent that there is something special about the two areas picked by the authors based on fMRI, it would be important to document whether there are any differences between those neurons inside and outside those so-called “face patches”. Additionally, it would be important to repeat analysis for neurons that are considered “non-face” neurons within those patches. Thus, there would be at least four different groups. “Face patch face neuron”, “face patch non-face neuron”, “non-face patch face neuron”, “non-face patch non-face neuron”. After doing all of this, one might be able to conclude that there is some topography in the representation of information in IT cortex, consistent with extensive studies of the spatial organization of responses in IT cortex (for a review on this, see Tanaka *Ann. Rev. Neurosci.* 2006).

(7) The current study finds fewer “face” neurons than what was claimed in a previous study (roughly 50% versus more than 90% in previous work by Tsao et al (*Science* 2006)). What is remarkable is that the threshold to define a “face” neuron based on d' is much lower in this study compared to the previous study. Such a lower threshold should lead to a much higher proportion of neurons considered to be “face” neurons. The result is the opposite. The authors quickly dismiss this challenge by saying that the electrodes record from neurons outside so-called “face” patches. If this is the case, it would be important to document which neurons are considered to be inside versus outside (according to anatomical criteria, without looking at the neuronal responses to “faces” versus “non-faces” to avoid circular arguments), then show the neurons considered to be “face” neurons inside and outside and then perform the analyses in the paper for “face” neurons inside patches, “face” neurons outside patches, “non-face” neurons inside patches, “non-face” neurons outside patches as noted in the previous question. Related to this question, the work in Bardon et al *PNAS* 2022 showed that the differences between neurons inside and outside so-called “face” patches are minimal.

(6) It is difficult to evaluate the major conceptual advances claimed in the study. Some difficulties arise from the author’s phraseologies that are not clearly defined. First, all claims center on there being different neural codes. Thus, an explicit definition of what constitutes a (singular) neural code is in order. I don’t believe such an explicit definition is found anywhere in the paper. The abstract opens,

“[...] the concept of neural coding through tuning functions. According to this idea, neurons encode stimuli through fixed mappings of stimulus features to firing rates.”

This definition is not specific enough. A face and a non-face contain different stimulus features by definition, so a general, singular “neural code” can assign the two images different firing rates, and therefore cannot support a conclusion of different neural codes. The paper needs a clear definition of a neural code that can change before the paper can give meaning to any claims about different, switching, or dynamic neural codes.

(8) The paper implies that the operational definition of a neural code in this work is the weighting vector in (dimension-reduced) AlexNet fc6 space. Defined thus, a neural code can change. However, this definition is severely limited. For example, even a hypothetical, static (i.e., not varying in time) neural code that is piecewise linear (and globally nonlinear) in the fc6 space will comprise different neural codes for the different linear regions. This constrained definition of a neural code makes the conceptual conclusion much less widely applicable than it first seems. Within the visual domain, see concern (4) that current models are imperfect models of neurons. Since IT responses are not perfectly linear in AlexNet fc6, the model-based results can only be interpreted as, “an approximate neural code approximately flips.” For the broader implications, it is unclear how one should adjudicate the sameness, differentness, and change in a neural code for logic (line 584), for which an artificial model that linearly maps onto neurons has not been identified.

(9) It seems that the intervening AlexNet model is dispensable when comparing within an image category. In this case, tuning direction inversion may be tested directly using neuron responses across individual images, without using neural network fitting, avoiding concern (4). Thus, it is relevant for the authors to add versions of Figs. 3b (left two columns). 4c, e, and related supplementary figures analyzed directly using responses without neural network fitting.

(10) The notions of a neural code “use[d] to represent a [stimulus]” (lines 101, 115) and “stimulus-dependent switching” (lines 30, 100) are also ill-defined. Under the above operational definition, a neural code cannot be defined for a single stimulus, because linear fitting requires multiple stimuli. (Incidentally, it is for this reason that the neural code in Extended Data Fig. 12e has fewer dimensions than in the main figures.) Arguably, given a defined neural code (e.g., the “object axis”), one can evaluate its fit to a specific stimulus by comparing the predicted and observed firing rates. However, the paper has always evaluated neural code fit as R^2 , an across-stimulus metric, in this paper. If a neural code cannot be defined for one stimulus alone, the study’s conclusion becomes more like, “The response differences among faces correlate with opposite AlexNet-fc6 features than the response differences among objects.” This altered conclusion may seem subtle, but it calls into question whether each and every face engages a stimulus-gated, domain-specific processing mechanism. This question also implicates the third major conceptual advance claimed in the discussion (lines 539–548). In other words, given an ambiguous image X (e.g., a pareidolia image, an animal with a visible face), just how exactly do the authors propose to concretely answer the question of whether that image is a face?

(11) The authors should specify in what exact sense they are claiming to “suggest/reveal a novel/new mechanism/form of neural representation/computation” (lines 29, 99–100, 408, 475). The mechanism of lateral inhibition seems able to predict qualitatively the main results: Faces that strongly excite the local population cause the population to inhibit itself, such that “average” faces (i.e., those that abruptly and strongly drive many face neurons) are initially preferred and later anti-preferred, whereas diffusely (across the population and/or time) or weakly activating faces become preferred to average faces in the late period. (Of course, what seems obvious is subjective.) The authors train a minimal RNN model to invert the input

gradient, and the model learns a “simple structure consistent with lateral inhibition” (line 290). To me, this is an interesting albeit confirmatory result. As the authors themselves eruditely review, lateral inhibition is a well-established, widespread circuit motif (e.g., Carandini & Heeger, *Nat. Rev. Neurosci.* 2012) that has been specifically speculated to play an important role in IT representations (lines 572–576). Closer afield, Koyano et al. (*Curr. Biol.* 2021) studied face-neuron responses to faces and found a late-emerging (100–300+ ms) preference for caricatures that reverses an earlier (50–100 ms) preference for the average face against caricatures (their Fig. 4B). While the direct comparison of face- and object-tuning axes (which requires an intervening deep-net model) showing opposite axis directions is novel to my knowledge and contradicts a recent result (Vinken et al., 2023), and the point about axis inversion over time is perhaps more emphatically made than in prior publications, the overall concept of a dynamic (and even inverting) tuning axis is not new.

(12) It is unclear how the conclusions in this study connect to the essentially opposite conclusions reached by Vinken et al 2023.

(13) The main figures show results from only one monkey, which deviates from normal practice in the field. It is important to show that the results are qualitatively reproducible across monkeys. Across-monkey reproductions are conventionally shown in the main figures, not the supplementary materials. The results in Extended Data Fig. 2e arguably differ from the main conclusion that faces switch to and stay with a category-specific code, a conclusion that affects some of the interpretations (e.g., the “second major conceptual advance” regarding behavioral results on lines 525-537). Moreover, the example monkey uses a block design, which, as the authors point out, engages well-known adaptation effects. Incidentally, the interleaved experiment incompletely controls for adaptation effects: On average half of the stimuli would still be faces, or 1.5 faces per second at the study’s presentation rate (150/150 ms on/off), while adaptation effects last more than a few seconds. I am not sure adaptation can fully explain the study’s results, but I don’t believe the current experiments rule out contributions from adaptation.

Minor concerns:

(14) The terms “Rapid” and “concerted” are frequently used as part of the conclusions, but neither is really shown except qualitatively. Going just by examples, one could point to cell 8 in Fig. 3g, which does not switch in the same 20-ms window as the other cells. It will be useful to more precisely define and quantify what rapid and concerted mean.

(15) How does prosopagnosia indicate a face detection gate (lines 46–50)? The same evidence is consistent with face-specific regions always (i.e., in an ungated way) doing detailed feature processing.

(16) On lines 61–64, it is relevant to mention that general-purpose deep nets are better models of “face” neuron responses than face-only deep nets (e.g., Vinken et al 2023).

Referee #3

(Remarks to the Author)

In this manuscript, the authors investigate the important question of whether face-selective cells use general (not specific to faces) versus face-specific mechanisms.

The research question is highly relevant and of significant interest to the community, especially given the renewed interest in whether face cells use a domain-general or domain-specific code (Vinken et al., 2023). The results support the conclusion that face cells exhibit a dynamic code that changes from early to late components. Overall, these findings are highly relevant and supported by a large number of data analyses. Moreover, several control analyses were performed to rule out possible alternative explanations. Methodologically, the manuscript employs state-of-the-art approaches. While the writing is clear, some conclusions appear insufficiently supported by the analyses performed. I concur with the novelty of revealing a dynamic coding for face cells. However, as a reader, I often found myself with lingering questions that, if addressed, could significantly enhance its potential impact. Below, I discuss these points.

1) Feature dimensions underlying early and late components:

The manuscript raises a crucial question regarding the underlying feature dimensions for the initial domain-general and subsequent domain-specific face code proposed by the authors. The absence of specific analyses addressing this question leaves a gap in understanding.

I give you some examples.

-You propose that face-selective cells are initially agnostic to the category of face. This conclusion suggests that the type of coding allowing discrimination between faces and objects is not based on the category of an object. However, this conclusion is not supported by specific analyses. What is the feature dimension underlying the single-axis encoding? In Extended Data Fig. 8b, examples of face and object images projecting maximally onto the extremes of the first two components show an interesting animacy distinction: high-projection objects are all animate, and low-projection objects are all inanimate. If animacy (in line with Bao et al., 2020) is a relevant axis in the object space for initial discrimination between faces and objects, can a cell encoding animacy be considered agnostic to the category of faces? First, it needs to understand that something is a face before concluding that it is animate. In other words, the examples shown in Extended Fig 8b do not seem to fully support the idea of agnostic cells. It would be interesting to see examples of face and object images projecting maximally onto the extremes of the first two components taken at different time windows as shown in Figure 3f. Such visualization could offer valuable insights into the dimensions underlying the domain-general code of face

cells as proposed by the authors.

-You conclude that 'Newly emergent face encoding axes improve face discrimination'. [line 455: 'Overall, these results show that the drastic changes in feature tuning occurring at ~100 ms have a functional consequence, markedly improving the ability of the neural population to perform fine face discrimination']. To conclude this, one needs to show that late components [120-140ms] better discriminate face identities relative to early components [80-100ms]. The analyses performed (Fig 5) to support this statement do not directly test this question. For instance, Figure 5b demonstrates that while long latencies appear to yield slightly better reconstructions, short latencies also produce very good results. Further, combining short and long latencies gives the best possible reconstruction. How does this fit with the idea that at short latencies face cells are agnostic to faces and encode feature dimensions common to faces and objects? How do these results support the complete change of neural code proposed by the authors? Wouldn't you expect a substantial difference between early and late latencies in their face reconstruction ability? Also, here you use very short latencies [50-75ms] but from your previous analyses, it seems that the object and face axes align at 80-100ms.

Related to this you write: [line 333: 'Each of the four cells showed new tuning to different features, e.g., $v \perp u$ for Cell 1 varied from small inter-eye distance and sharp chin to large inter-eye distance and round chin']. This statement lacks support from specific analyses. Upon reviewing these images, the primary distinction I observe is the variation in luminance, transitioning from bright to dark from left to right. How does this emphasis on a low-level feature align with the suggestion that the face code during late latencies is specific to face identity?

Overall, throughout the manuscript, I was expecting analyses to try to understand what the underlying dimensions of the domain-general and domain-specific code are. In my view, such an investigation would increase the potential impact of this work.

2) Comparisons between AM and ML.

The manuscript's observation regarding the largely similar pattern of dynamics in face patch AM and ML, albeit with longer latency in AM, raises intriguing questions about potential differences between these regions. Does it mean that AM represents faces in the same way as ML but just slightly later? According to these results, AM appears to be a sort of copy of ML. This is rather unexpected and in contrast with previous findings from the same lab (Tsao et al., 2010). It would be extremely interesting to further investigate potential differences between AM and ML representations. By inspecting Extended Data Fig. 9, as the authors point out, there seem to be already some interesting differences between ML and AM (line 262: 'There was also a weak, brief second period of re-alignment at ~160 ms not observed in ML'). I was expecting more analyses in this direction given that data has been acquired from both areas.

3) AlexNet trained on ImageNet

The authors used AlexNet trained on ImageNet to generate a 60-dimensional space capturing features of both faces and objects (or only faces for the second part investigating domain-specific coding) to compute the object and the face axes. Thus, what comes after is constrained by the assumption that AlexNet is a good model to capture the object and face space in primates.

While I acknowledge that CNNs are currently considered the best models for primate perception, there is also ample evidence highlighting their limitations. One of the most striking differences between biological vision and convolutional neural networks (CNNs) is their bias toward image texture over object shape (Geirhos et al., 2018). Moreover, a CNN trained on ImageNet does not inherently learn face-specific features since faces are not included in the training set. This aspect becomes particularly relevant in the second part of the analyses where the authors test the domain-specific code for faces.

Thus, overall, the use of AlexNet trained on ImageNet raises concerns regarding its ability to capture face-specific features. Given these limitations, especially concerning face-specific features, utilizing a face identity-trained network such as VGG-face could offer more relevant insights, particularly in the context of the domain-specific code for faces.

4) Some conclusions are not supported by sufficient evidence.

In my opinion, it might be beneficial to reconsider some of the conclusions that seem to surpass the support provided by the analyses. For instance, the sentence in the abstract, 'new tuning developed to multiple higher feature space dimensions supporting fine face discrimination', appears to lack sufficient evidence from my perspective. To strengthen this conclusion, it may be advisable to conduct a direct test of face identity discrimination at both early and late latencies, and possibly include a comparison between ML and AM.

Additional comments:

[line 322: 'Fig. 4f shows that indeed, more dimensions were required to explain 90% of response variance at long compared to short latency (96 dim for long, 82 dim for short)']. Is this difference significant?

[line 116: 'Our findings challenge the notion that object recognition can be explained by largely feedforward processes, instead revealing an essential role for recurrent dynamics in core visual processing']. How does this statement relate to the fact that you created the object and face axes based on a feedforward model?

Extended Data Fig. 5 (d and f) shows the distribution of face axes computed for stimulus-shuffled data. These results seem

to suggest that shuffling the data does not affect the face axes' length. This is not what I would expect if these face axes project toward dimensions relevant to face identity discrimination.

Version 1:

Reviewer comments:

Referee #1

(Remarks to the Author)

The authors have addressed all my previous comments in the revised version.

My only final comment is that it would be nice if the authors could briefly discuss their findings reported in Nature last year (She et al Nature 2024), where they describe a rotation of the feature axis at long latencies for familiar compared to non-familiar faces, which seems quite relevant within the context of the current paper (showing a switch of axis for individual discrimination vs categorization).

Referee #3

(Remarks to the Author)

The authors have thoroughly addressed all the reviewers' comments, and I am satisfied with the revisions; I therefore recommend the manuscript for publication.

Response to Referees' comments:

We thank all three reviewers for their careful reading of our paper and helpful suggestions. By addressing the concerns raised and incorporating our responses into the manuscript, we believe we have significantly improved the paper.

General summary of the revision (relevant to all reviews):

In response to the reviewers' comments, we have performed a series of new experiments with new stimulus sets as well as additional monkeys. We have also restructured parts of the paper to clarify our key message. Below, we summarize the new data and major revisions:

1. New experiment to characterize axis dynamics in cells selective for non-face categories and face cells outside face patches

To assess the generality of the axis change phenomenon (a question raised by Reviewers 1 and 2), we recorded from neurons selective for a range of non-face object categories (e.g., spiky objects, stubby objects, animals, monkey bodies) as well as from face-selective neurons located outside canonical face patches. We found that axis change is not exclusive to face cells—some non-face category-selective neurons also exhibited axis change, and when this occurred, it was also stronger for the preferred category. However, axis change in non-face cells was generally weaker and less temporally consistent across cells than that in face cells within face patches. In contrast, face-selective neurons outside face patches consistently showed axis change, albeit with a modest temporal delay compared to those within face patches. These results suggest that **axis change is a general phenomenon in IT cortex but is most robust and coordinated in face-selective circuits.**

2. New experiment to test axis change in response to ambiguous faces

Reviewer 2 raised the question: if face-selective neurons exhibit axis change in response to clearly face-like stimuli but not to clearly non-face stimuli, what happens with ambiguous stimuli, such as degraded faces or Pareidolia? To investigate this, we conducted new experiments in which monkeys were shown a range of animal faces (e.g., monkey, dog), degraded face images (e.g., occluded, blurry, Mooney), and Pareidolia stimuli. We also developed a novel metric—the single-stimulus axis change score—to quantify whether an individual image elicits axis change based on the dynamics of the population response. Using this metric, we found that human and monkey faces consistently triggered axis change, while axis change in response to Pareidolia images was more variable. For degraded human faces, milder degradations still triggered axis change, although in certain cases the change occurred later in time, while more severe degradations failed to elicit axis change. This analysis **extends the axis change framework from clear faces to a much larger set of stimuli whose classification as faces is more ambiguous.** Moreover, the single-stimulus axis change score provides a new metric to assess whether a stimulus is a face or not.

3. New experiment to directly show that high-dimensional features improve face discrimination at later latencies

Reviewers 1 and 3 suggested using a more direct analysis to demonstrate that emergent tuning in higher feature dimensions contributes to improved face discrimination, beyond the evidence provided by reconstruction. In response, we performed a time-resolved analysis using 20-ms sliding windows. For face categorization (face vs. non-face), we used a linear decoder. For face identity discrimination, we measured the normalized pairwise distance among population responses to different face stimuli. This analysis revealed that: (1) Discrimination performance increased significantly later than categorization, and (2) Discrimination continued to rise over time, peaking after 100 ms—consistent with the timing of axis change and the emergence of new tuning dimensions.

However, one might still argue that the late improvement in discrimination could result from a refinement of tuning in low-dimensional features rather than the emergence of new tuning in higher-dimensional features. To directly test this, we designed a new experiment that effectively “knocked out” the contribution of low-dimensional features. We generated synthetic faces in which variance was tightly constrained in the lower dimensions, while higher-dimensional features were allowed to vary freely. If face discrimination relied primarily on low-dimensional features, this manipulation should have greatly reduced neural discrimination accuracy. Instead, we observed that discrimination performance still improved at later latencies. This result provides strong evidence that **higher-dimensional features play a critical role in supporting fine face discrimination during later stages of processing.**

4. New experiment to rule out alternative explanations for axis change based on feature statistics

Reviewer 2 asked whether the axis change phenomenon could be explained by distinct low-level visual features of face images. To test whether axis change might be driven by low-level visual features, we identified a subset of face and object images with maximal overlap across various low-level features. We then trained decoders to predict each image’s axis change score using these low-level features, along with the categorical face/object labels. Among all features tested, the categorical distinction between faces and objects was the strongest predictor of axis change scores, suggesting that the high-level concept of category, rather than low-level visual features, drives axis change.

5. Direct comparison between ML and AM dynamics

Reviewers 1 and 3 asked for a more detailed characterization of differences between axis change dynamics in ML and AM, two patches which our lab previously showed have distinct view-invariance properties. To ensure data from two monkeys per patch, we performed AM recordings in an additional monkey. Confirming previous findings from our lab, we found that AM neurons have stronger tuning to appearance versus shape features than ML. But importantly, both patches show axis change, suggesting each patch independently applies its own face detection gate to ensure that only valid faces are further processed.

6. Characterizing face patch geometry with large-scale Neuropixels recordings

Reviewers 1 and 2 asked about the yield of faces cells using the newly developed NHP Neuropixels probes, how this compares to previous report from the lab using single unit recordings, and how this compares to human studies. To address this, we collated all Neuropixels sessions across two monkeys in which we attempted to target face patch ML and AM (67 sessions total, including many sessions not part of the current study) and analyzed the range (i.e., vertical extent along the probe) and concentration of face-selective cells in each session. We found that although the range of face-selective cells varied hugely across sessions, due to the different penetration trajectory, the proportion of face-selective cells was consistently high within face patches and substantially lower outside. **These results are consistent with prior single-unit findings, and provide a large-scale, high-resolution map of face cell distribution in IT cortex.**

7. Testing generality across different feature spaces

Reviewers 2 and 3 raised concerns about the robustness of the axis change phenomenon to choice of feature space. To ensure the results were not restricted to AlexNet features, we repeated key analyses using features extracted from a state-of-the-art DNN (ResNet-50) trained on VGG-Face2. We replicated all our main findings, demonstrating that **the axis change phenomenon is a general property of face-selective neurons, not an artifact of the chosen feature space.**

Referee #1 (Remarks to the Author):

In this paper the authors contrast two prevalent theories about face perception: the domain specific theory, with specialized mechanisms dedicated to faces, and the domain general theory, with computations not specific to faces. To address this issue, they recorded neurons in monkey face patches AM and ML, while the animals viewed pictures of faces and objects. The key, novel finding is that, rather than having different population of neurons being involved in different aspects of face processing, there is a dynamic and rapid switch of the neurons' code, going from detection, at first, to face discrimination later on. They support this claim by showing the reversal of the projection axes representing face stimuli at about 100 ms (with each axis determined by the projection of layer fc6 of an AlexNet network into the 2 largest principal components), something that was not present for the object stimuli. Overall, I find the paper very interesting not only for colleagues working on face perception but for those interested in neuronal coding in general, posing the intriguing hypothesis that, instead of having sets of neurons with a given and fixed function, the neurons' code may change dynamically to perform different functions. This is the type of papers I would personally like to see published in Nature, but I think the authors should consider the comments below to improve the presentation of the results, which is at times a bit cluttered and convoluted, deviating the attention from the main message.

We are grateful to the reviewer for their appreciation of our work and valuable suggestions.

1. It is surprising that the authors find qualitatively similar results in AM and ML. As far as I remember from previous works of the authors, responses in AM are much more selective

and, given that the axes switch described in this paper comes together with an increase in sparseness (but see below), one would expect different findings in these two areas. I suggest the authors to at least briefly address this point in the discussion.

We thank the reviewer for highlighting the surprising similarity between ML and AM. Initially, we were also surprised to find that AM exhibits axis change behavior, implying that AM goes through a face detection stage similar to ML. However, this redundancy could be advantageous — each patch may independently apply its own face detection gate to ensure that only valid faces are further processed. Such redundancy would enhance robustness. Moreover, anatomically, the face patch system is not a strictly feedforward hierarchy: AM receives substantial direct input from PL, bypassing ML entirely (Grimaldi & Tsao, Neuron 2016).

Beyond the similar axis change dynamics, we identified several notable differences between ML and AM:

(1) Shape vs. appearance tuning: Prior work from our lab has suggested that ML neurons are more tuned to shape, whereas AM neurons are more tuned to appearance (Chang et al., 2017). To test this in our data, we transformed the AlexNet-based feature space into the shape-appearance space defined in Chang et al., 2017 and performed axis mapping by regressing neural responses onto these new features. We then decomposed each cell's face axis into shape and appearance components and defined a shape preference index as the relative magnitude of the shape vs. appearance axis. Using this index, we confirmed our lab's previous finding that ML neurons show stronger shape tuning than AM neurons when using time-averaged responses (**Reviewer Fig. 1a**).

It is important to note that in the present study, we only presented frontal-view faces, limiting the extent to which view-dependent differences between ML and AM can be assessed.

(2) There is a weak re-alignment period observed in AM's face axis dynamics (**Reviewer Fig. 1e**), which is not present in ML (**Reviewer Fig. 1c**). And this realignment coincides with a second peak of raw responses in AM (**Reviewer Fig. 1d**).

We have added these to the Discussion:

"Face patches ML and AM exhibited strikingly similar dynamics (Extended Data Figs. 7, 14, 23). By ensuring that a stimulus is categorized as a face only after it successfully passes the face detection gate in each patch, the robustness of face categorization may potentially be enhanced."

Regarding sparsity, we re-examined response sparsity in ML and AM (see also Reviewer 1, Point 7). Contrary to earlier findings (Freiwald & Tsao, Science 2010), responses in our current dataset were actually *sparser* in ML than in AM (**Reviewer Fig. 2a**). To better understand this discrepancy, we re-analyzed our data using the same single-unit sparsity metric as in Freiwald & Tsao 2010. Even with this approach, AM was not significantly sparser than ML. We observed the same trend in monkey J (**Reviewer Fig. 2c, d**).

What could contribute to the discrepancy with our previous paper? One possibility is that in Freiwald & Tsao 2010, the stimulus set used to probe sparsity consisted of 8 different views of 25 identities, whereas here we used thousands of frontal faces without any view variations. To test if the discrepancy in relative sparsity between AM and ML might be due to a difference in stimulus sets, we showed the same stimulus set used in Freiwald & Tsao 2010, consisting of 8 different views of 25 identities, to monkey A while recording with Neuropixels probes. With this stimulus set, indeed AM population responses were now sparser than those in ML (Independent t-test (Welch's), $t(202) = 19.68$, $p = < 0.05$), using both population (**Reviewer Fig. 2e**) and single-unit (**Reviewer Fig. 2f**) sparsity metrics. This suggests that sparsity measurements could be highly subject to stimulus set structure. Since this analysis of sparsity does not directly impinge on the central conclusions of our paper regarding axis change, we do not include it in the manuscript.

Reviewer Figure 1. Comparison between ML and AM. **(a)** Shape preference indices for ML and AM of time-averaged responses. **(b)** Mean response time course to each face and object stimulus in ML, averaged across cells. **(c)** Cosine similarity between the overall object axis for each cell (computed using a time window of 50-220 ms) and its time-varying face axes in ML, sorted from top to bottom according to face selectivity d' (left). **(d)** Same as (b) for AM. **(e)** Same as (c) for AM.

Reviewer Figure 2. Relative sparsity of face patches ML and AM is dependent on stimulus set structure. **(a)** Population sparsity time course of ML and AM face cells in response to our thousand face stimuli in monkey A. A higher number indicates greater sparsity. **(b)** Single-unit sparsity distribution of ML and AM face cells in response to the thousand face stimuli in monkey A. Here, a lower number indicates greater sparsity. **(c, d)** Same plots as **(a)** and **(b)** from monkey J. **(e, f)** Same plots as **(a)** and **(b)** computed from responses to a face view stimulus set in monkey A.

2. It makes sense that the proportion of face neurons is lower than in previous studies due to the use of Neuropixels (given that not all recording sites are at the peak of the face activations). The percentage of face responsive neurons is actually quite close to the one recently reported in analogous face processing areas in humans (Quiari Quiroga et al. Nat Comm. 14: 5661; 2023) and in that study, as here, due to clinical-experimental constraints, the recordings were not necessarily at the peak of the face activations. To encourage comparisons across species (and with the authors previous studies), I think it would be useful to quantify the proportion of face responsive neurons as a function of their location with respect to the peak of the face activations. Ideally, this could be a plot showing such proportion as a function of distance to the peak activation, However, I guess there might not be enough responses to produce such a plot, but at least the authors could quantify the proportion of face selective responses at the peak of the face activations vs. the one found in more distant recording sites.

We fully agree that Neuropixels recordings provide a unique opportunity to map neurons' face-selectivity across large spatial extents, and that enabling comparisons with human studies is valuable. Since the probe insertion trajectory varied across sessions, the portion of the face patch sampled in each experiment also fluctuated significantly.

To gain a comprehensive picture of how face selectivity varies with distance from face patch center, we compiled all high-quality Neuropixels recording sessions (67 sessions in total) in which we successfully targeted either face patch ML or AM—including many sessions that were not part of the current study, but for which we had obtained responses to our standard set of face and object screening stimuli. For each session, we first identified the probe segment that traversed the face patch (by finding the maximum continuous stretch of units with FSI ≥ 0.33 , allowing for small gaps of three units or less below threshold), and then computed the proportion of face-selective cells (defined as FSI ≥ 0.33) within and outside of it.

The proportion of face-selective cells within the mapped face patch region remained relatively consistent, averaging 0.75 across all ML sessions (**Reviewer Fig. 3a**). As examples, we show responses of all recorded cells to the screening stimuli for three representative sessions—those with low, median, and high ratio of mapped face cells (**Reviewer Fig. 3c**). In contrast, the proportion of face-selective cells recorded *outside* the face patch was much lower, averaging 0.28 for ML (**Reviewer Fig. 3b**). AM recordings showed a similar pattern, with the ratio of face cells being 0.81 for inside face patches and 0.24 outside (**Reviewer Fig. 3d, e**). The proportions of cells in our Neuropixels sessions are consistent with prior reports from our lab (Tsao et al., *Science*, 2006) and notably higher than those reported in recent human studies (Quian Quiroga et al., *Nat. Comm.*, 2023). Neuropixels recordings pick up every single cell including ones that are not very large and easily isolated as well as ones that are not clearly visual; therefore it is not surprising that the percentage of face-selective selective assessed with Neuropixels is slightly lower than that assessed with single tungsten electrode recordings. Together, these findings provide a detailed, large-scale map of face-selective neuron distribution in the IT cortex.

Reviewer Figure 3. Face selective unit distribution across different Neuropixels recording sessions. (a) Ratio of face-selective cells ($FSI \geq 0.33$) within the face patch range for each ML session, sorted. (b) Same as (a) but for face-selective cells outside face patch range. (c) Response profile of three example ML sessions with low, middle and high face cell ratios. (d, e) Same as (a, b) but for AM-targeting sessions.

We have added relevant clarification to the manuscript:

“These percentages are lower than what has been reported using single tungsten recordings^{15,28}; likely because (1) Neuropixels probes capture activity from all nearby neurons, including smaller or less well-isolated units, as well as neurons that are not strongly visually driven, and (2) portions of the probe often extend beyond face patch boundaries (Extended Data Fig. 3).”

3. Related to the previous point, another interesting analogy with human studies is that in the abovementioned paper, with recordings in the human fusiform gyrus, using a decoding approach it was also found that face discrimination was preceded by face detection (and also by face-familiarity detection), although in this case there was not a finding of a switch of representation axes, which is the novel finding of this study. I would encourage the authors to mention the human study to foster comparisons across species.

We appreciate the reviewer’s suggestion and now cite and discuss the relevant work by Quian Quiroga et al. (2023), which reported similar temporal dynamics in human FFA showing that face categorization precedes face identity decoding:

“A similar phenomenon has also been reported in a human FFA electrophysiology study recently⁴⁷.”

4. Do the authors expect what they show for face cells here to also apply to the object

cells they have described in a previous study (Nature 2020)? That means, would they expect that object cells switch from object detection to object discrimination? I am not asking to add such an analysis, but this could be mentioned in the discussion, supporting their point of a general switching mechanism of single neuron responses.

We thank the reviewer for raising this fundamental question, does the axis change phenomenon generalize beyond face patches? To address this, we conducted a new series of experiments using a large set of novel object categories—including spiky objects, stubby objects, animals, and monkey bodies—and recorded from regions in IT cortex that showed selectivity for these categories based on fMRI localizers (Bao et al., Nature 2020).

We first examined cells selective for monkey bodies (defined as peak $d' > 0.1$, **Reviewer Fig. 4a**). These cells reliably preferred monkey bodies over general objects (**Reviewer Fig. 4b**). For each 20-ms response window, we mapped each cell's object axis and body axis. Similar to face cells, these cells' body axes initially aligned with object axes and rotated away at later latencies (**Reviewer Fig. 4c**, top vs. bottom). Consistently, many cells showed negative correlations between their early and late body axes (**Reviewer Fig. 4d**, and their object axes less so. We further plotted examples of single-unit pairwise correlation matrices that clearly demonstrated axis reversal (**Reviewer Fig. 5**). Our overall conclusion is that monkey body cells show axis reversal for monkey bodies, but the degree of axis reversal is weaker and less consistent compared to face cells (compare **Reviewer Fig. 4d** with **4t**).

Similar patterns were observed in cells selective for stubby and spiky (**Reviewer Fig. 4e–h, i–l**). In contrast, animal body-selective cells largely maintained stable animal body axes throughout the trial, showing no clear evidence of reversal (**Reviewer Fig. 4m–p**).

Interestingly, we also encountered face-selective cells outside canonical face patches. These outside face-patch face cells showed clear axis reversal, albeit weaker and with a delayed onset relative to face-patch neurons (by ~ 20 ms; **Reviewer Fig. 6a** vs. **6b**), suggesting that this phenomenon might be inherited from face patch dynamics via recurrent interactions.

Finally, we note that variability in the magnitude and consistency of axis change across object-selective populations could also reflect differences in recording site. Face-selective neurons are well-characterized and spatially clustered, whereas for other categories the density of optimally selective neurons may have been lower at the recording sites we sampled.

In summary, our new experiments demonstrate that axis change behavior is not exclusive to face-selective cells—it can also be observed, to varying degrees, in neurons selective for other object categories in IT. However, the effect is strongest, most consistent, and most concerted in face-selective populations. These results suggest that dynamic changes in tuning axes may be a more general coding strategy in IT cortex, with faces serving as a particularly salient example.

Reviewer Figure 4 (Replicated as Extended Fig. 17). Category-selective units outside face patches show various degrees of axis change at long latency. (a) Top: Example images from monkey body image set. Bottom: Distribution of monkey body stimuli and general non-monkey

body object stimuli features in PC1-2 of the object space. **(b)** Time averaged responses of all monkey body units to monkey body and non-monkey body stimuli. **(c)** Distribution of monkey body (left, purple) and object (right, green) axes projected onto the top two dimensions of the 60-d object space, for early (top) and late (bottom) latency responses of monkey body cells. (Only units with peak $d' > 0.2$ are used in this panel.) **(d)** Histogram of correlation of monkey body (purple) / object (green) axis between early and long latency for each unit. (The early and late time windows were defined individually for each unit: the early window was centered on the unit's peak response time, and the late window was offset by 20 ms following the early window.) **(e-h)** Same as a-d for stubby cells. **(i-l)** Same as a-d for spiky cells. **(m-p)** Same as a-d for animal cells. **(q-t)** Same as a-d for face cells outside face patches.

Reviewer Figure 5. Axis change pattern of example single monkey body-selective cells. Matrices of cosine similarities for three example monkey body-selective cells between (object, object), (monkey body, monkey body), and (monkey body, object) axes (computed using all 60 dimensions) for different pairs of latencies.

Reviewer Figure 6. Axis dynamics of face-selective units inside and outside face patches. **(a)** Matrices of mean cosine similarity across the face-selective cells outside face patches, comparing (object, object), (face, face), and (face, object) axes at different time points. Red arrow on top of the rightmost matrix indicates the face axis switch time (~110 ms). **(b)** Same as (a) but for face-selective units inside face patch ML. Red arrow on top of the rightmost matrix indicates the face axis switch time (~90 ms).

We have incorporated the above findings into the manuscript:

“Finally, we asked whether the observed axis change behavior is unique to face-selective cells or reflects a more general coding strategy among category-selective populations across IT cortex. To explore this, we recorded from neurons outside face patches and presented a diverse set of object categories including spiky objects, stubby objects, animals, and monkey bodies (see Methods; Extended Data Fig. 17). We found that axis change is not exclusive to face cells. Some category-selective neurons outside face patches also exhibited axis change, though this was typically weaker and less consistent than that observed in face-selective populations. This suggests that dynamic tuning is a broader computational feature of IT cortex, potentially supporting different levels of processing demands across domains. Notably, face-selective neurons showed the strongest and most temporally concerted axis change behavior. Furthermore, we found that even face-selective neurons located outside canonical face patches exhibited axis change, albeit with slightly delayed timing compared to neurons inside face patches (Extended Data Fig. 18), raising the possibility that their dynamics may be inherited through recurrent interactions with face patch circuits.”

5. Lines 215-216: Why does the object axis become stable at 60 ms, while the face axis does it much later, at 118 ms? Is it because face neurons detect the picture is not a face (and won't be further processed as it is the case with the face pictures)?

Indeed, we believe that during the initial alignment period—when a face cell's face axis aligns with its object axis—the neuron is engaged in face detection. The reason the face axis stabilizes later at the population level (**Reviewer Fig. 7a**, middle panel; note the absence of a sharp square in the early face–face similarity matrix) lies in the timing variability across neurons. Specifically, the onset of face axis alignment—defined as the half-peak time of the correlation between each unit's face and object axis (indicated by

black tick marks in **Reviewer Fig. 7b**, top)—does not occur in a synchronized manner across the population. In contrast, the face axis reversal occurs more concertedly (black tick marks in **Reviewer Fig. 7b**, top), as does the object axis onset, which is also more consistent across units. This temporal coherence results in earlier and more sharply defined stabilization of the object axis at the population level.

The distributions of face axis onset, face axis reversal, and object axis onset times are shown in **Reviewer Fig. 7c**. As expected, the onset of face axis alignment is more variable than either of the other two. This variability likely accounts for the delayed and less coherent stabilization of the face axis across the population, even though individual face cells are already actively engaged in face detection during early stages of the response.

Reviewer Figure 7. Cells' face axis initiation times are more variable than object axis across the population. **(a)** Matrices of mean cosine similarities for face cells between (object, object), (face, face), and (face, object) axes (computed using all 60 dimensions) for different pairs of latencies. **(b)** Cosine similarity between the overall object axis for each cell (computed using a time window of 50-220 ms) and its time-varying face (top) and object (bottom) axes, sorted from top to bottom according to face selectivity d' (left). Black tick marks indicate axis onset time, red tick mark in top panel indicate face axis reversal time. **(c)** Histograms of face axis initiation time (std = 18.14 ms), object axis initiation time (std = 10.30 ms), and face axis reversal time (std = 12.21 ms) across the cell population.

6. It is very difficult to see anything in the inset of Figure 3f. This could be shown better in the supplementary information.

We apologize for the difficulty in viewing the inset. We have added a zoomed-in version of the inset to the supplementary figures (**Extended Data Fig. 10**). Please see below.

Reviewer Figure 8 (Replicated as Extended Data Fig. 10). Zoomed-in version of Figure 3f.

7. The claim that the axes reversal is accompanied by a sparsening of the face cell responses makes sense and is very important, but, in my view, is poorly supported by the presented data. In fact, the only result related to this is depicted in Figure 3h, which actually shows the opposite effect (a decrease of face selectivity at the time of the axes reversal – i.e. at about 100 ms). If the authors want to keep this claim, it should be better supported – explaining what’s happening in Figure 3h, showing a few examples and/or quantifications of such sparsening, etc.

We apologize for the confusion. The Treves-Rolls population sparsity metric increases its value as the population responses becomes sparser, so **Fig. 3h** shows that the population is the least sparse slightly before 100 ms and does becomes sparser during axis change (at ~100 ms). This also matches the raw response time course (**Fig. 3d**), where the response becomes sparser for all faces at ~100 ms.

8. Section “Face axis reversal in low dimensions is accompanied by emergence of diverse new tuning to higher dimensions of face space”. I find this section to be confusing. It could eventually be skipped and some of the information there incorporated into the other sections or moved to the Supplementary Information. First, the only evidence of new

tuning I could find in this section is a relatively small change in the number of PCs needed to explain the variance of the responses (Fig. 4f, going from 82 to 96 dimensions to explain 90% of the variance). The section starts with some modelling results that in my view are presented a bit out of the blue and do not offer any direct evidence supporting the authors' claim. I have already mentioned my concerns about the claim of an increase of sparseness at the time of the axis switch, and I don't see any other good evidence of a new tuning, compared with what was presented in the previous sections. The last 3 pages of this section describe different controls, which are important (particularly the one showing that results are not due to larger responses to faces) but that can just be mentioned and described in detail in the Supplementary Information (or in a separate section about controls to rule out alternative hypothesis, instead of a section describing a new tuning, as it is currently done). Furthermore, the controls are presented as supporting lateral inhibition, but there's no direct evidence of lateral inhibition in this paper (just a modelling result). I see these as support to the claim of an axis reversal, ruling out alternative explanations, but not as supporting the presence of lateral inhibition. Finally, scenario two (lines 357 and below) is not clear to me. Would be good to explain more clearly why this scenario, not being an axis reversal, will look as such.

We thank the reviewer for the helpful suggestions. We have modified this section to address all the concerns raised. Specifically:

1. *Refining the logical flow of the section:*

We created a new section summarizing these controls, and restructured the section to present a more direct and coherent narrative. We begin by asking what mechanisms could explain the observed axis reversal. There are two major possibilities: (1) A cell-intrinsic mechanism, which we briefly discuss along with controls that rule it out. (2) Lateral inhibition within the population, which would naturally lead to the emergence of new tuning dimensions. To demonstrate that this mechanism is plausible, we present the recurrent neural network (RNN) simulation. We then describe our empirical findings showing that new tuning dimensions emerge in higher-dimensional face space.

2. *Condensing and moving control analyses to supplementary material:*

We condensed the introduction for the three controls and moved the detailed descriptions of the control analyses to the supplementary materials.

3. *Clarifying the role of lateral inhibition:*

We explicitly state that while lateral inhibition provides a plausible explanation for the observed effects, we do not have direct experimental evidence of lateral inhibition in this study.

4. *Strengthening evidence for new tuning in high dimensions:*

To rigorously evaluate whether tuning to new higher-dimensional features emerged, and whether this contributed to improved discrimination performance at long latency, we conducted a new experiment that effectively "knocked out" the contribution of low-dimensional features (see **Reviewer Fig. 28** below). We generated synthetic faces in

which variance was tightly constrained in the lower dimensions, while higher-dimensional features were allowed to vary freely. If face discrimination relied primarily on low-dimensional features, this manipulation should have greatly reduced neural discrimination accuracy. Instead, we observed that discrimination performance still improved at later latencies. This result provides strong evidence that higher-dimensional features play a critical role in supporting fine face discrimination during later stages of processing.

These revisions improve the logical structure of the section, make the manuscript more concise and reader-friendly, and ensure that all claims are properly contextualized. We appreciate the reviewer's insightful suggestions, which have helped strengthen this section significantly. Due to the extensive changes we made to this section, we refer the reviewer to the revised manuscript lines 306-406 to see the changes in detail.

9. Related to the previous point, it is not clear what panels 4h-k show.

We now clarify this in the manuscript (new additions are bolded):

"To directly visualize the new tuning dimensions emerging at long latency, for each cell, we first computed its preferred axis at 80-100 ms (\vec{v}_1) and at 120-140 ms (\vec{v}_2). We then **derived** the component of \vec{v}_2 orthogonal to \vec{v}_1 (\vec{v}_\perp) (Fig. 4g). This component represents a new tuning direction that is orthogonal to the cell's previous tuning direction. **We projected the features of each face onto \vec{v}_\perp (x-axis) and the principal orthogonal direction to \vec{v}_\perp (y-axis) and plotted neural responses at different time windows (80–100 ms, 120–140 ms, 160–180 ms) for four example cells.** This analysis **readily** revealed tuning to new dimensions emerging around the time of reversal in single cells (Fig. 4h-k, top). In each of the four example cells shown, new tuning is apparent along \vec{v}_\perp at 120-140 ms. To further illustrate what these new dimensions encode, we generated faces varied along \vec{v}_\perp (Fig. 4h-k, bottom). Each cell showed new tuning to different features, e.g., \vec{v}_\perp for Cell 1 varied from small inter-eye distance and sharp chin to large inter-eye distance and round chin."

10. The decoding approach used by the authors, seems unnecessarily convoluted to me. First, they reconstruct faces from the neurons' responses, as in Chang and Tsao's Cell 2017 paper. However, in that paper the accuracy achieved with the face reconstructions is truly remarkable, which does not seem to be the case here, where the authors make relatively vague and subjective statements (e.g. talking about reasonable reconstructions) and the quantification is made by comparing to the best possible reconstruction of the faces (something that depends on the model) and not comparing to the original face pictures. Does this offer any advantage compared to just having two decoders with a sliding window (instead of taking two time intervals) comparing the decoding of face detection and face discrimination? If yes, this should be explained in methods and if not, the authors may consider using the more straightforward approach mentioned here.

We thank the reviewer for suggesting a more direct analysis to examine the temporal evolution of face detection and discrimination. Following this suggestion, we computed face categorization accuracy (face vs. object) using a linear decoder on 20-ms sliding windows. For face discrimination, we measured the separability of responses to different face stimuli by calculating the mean pairwise Euclidean distance normalized by the pooled standard deviation, also using 20-ms sliding windows. Consistent with the central message of our manuscript, this analysis revealed that face identity decoding peaks later than face categorization decoding (**Reviewer Fig. 9a**). Additionally, we observed a clear

increase in face discrimination accuracy at later time windows compared to earlier ones, aligning with the results shown in **Fig. 5**. While this decoding-based approach reinforces our findings, we chose to retain the reconstruction-based analysis in the main text because it provides a more vivid and intuitive visualization of the underlying population coding. Nonetheless, we agree that this more direct, quantifiable analysis is a valuable complement and strengthens the overall conclusion. We have added **Reviewer Fig. 9a** as **Extended Data Fig. 24c** and added this note to the manuscript:

"We also analyzed face category and identity discrimination performance directly from the population responses using a 20-ms sliding window (see Methods), without relying on reconstruction. This analysis confirmed that face discrimination accuracy increased at later time points, peaking after the rise in categorization accuracy, consistent with a temporal shift from coarse to fine coding^{3,15} (Extended Data Fig. 24c)."

We also mention it in the Discussion:

"Tsao et al. came to a similar finding, comparing time courses for decoding face/object category versus individual face identity in face patch ML¹⁵. We have also confirmed that face categorization peaks earlier than face discrimination using the present dataset (Extended Fig. 24c)."

The reviewer also raised concerns about the quality of our reconstructions compared to those reported in Chang et al. (2017). To address this, we replicated the decoding analysis from that study by using distractor faces in feature space to evaluate decoding accuracy (**Reviewer Fig. 9b**). The decoding accuracy in our main session was only slightly lower than that reported in Chang et al. (2017) (compare **Reviewer Fig. 9b** and **c**), suggesting that the apparent drop in reconstruction quality likely stems from the image rendering step, rather than from a failure in decoding. Specifically, while both studies involve regression from neural responses to feature space, the final step in our pipeline—generating images from predicted features—relies on a realistic face autoencoder, which is less optimized for generative fidelity than the shape-appearance model used in Chang et al. (2017) for synthetic faces. This likely accounts for the perceived difference in reconstruction quality, despite comparable decoding performance.

Additionally, we conducted a new experiment in which the same set of cells was presented with both the synthetic faces from Chang et al. (2017) and the realistic faces used in our study. Interestingly, the decoding accuracy for realistic faces was even higher than for the synthetic faces in this session (**Reviewer Fig. 9d**). This further supports the conclusion that the decoding process itself is robust. Thus overall, we think that the drop in reconstruction quality is due to differences in the rendering models, not the decoding accuracy.

Reviewer Figure 9. Decoding analysis controls. (a) (Replicated as **Extended Fig. 24c**) Time course of face categorization (face vs. object) and face identity discrimination, computed using 20-ms sliding population responses from all ML cells. **(b)** Accuracy of decoding realistic face identities using time averaged response. Accuracy is measured as a function of number of distracting faces. **(c)** Result in Chang et al. Cell 2017 of the same analysis as in (b), performed on responses to synthetic faces (black curve, 50-d features). **(d)** Face identity decoding accuracy of synthetic faces (used in Chang et al. Cell 2017) versus realistic faces (used in the current manuscript) for a new experiment.

11. Fig. d-e. Why does the reconstruction accuracy decrease with the number of neurons for the short-latency window. This is very counterintuitive to me (in general, the more neurons, the better accuracy) and it is not discussed in the paper.

We appreciate the reviewer's concern and agree that, at first glance, it may seem counterintuitive that reconstruction accuracy decreases with the number of neurons for the short-latency window in **Fig. 5d**. To clarify, **Fig. 5d** compares reconstructions generated using short, long, and combined time windows and measures which is closest to the best possible reconstruction. The plot shows that with a small number of neurons, reconstructions based on short-latency responses *outperform* those based on long-

latency responses. However, as the number of neurons increases, long-latency responses lead to better reconstructions.

We believe this reflects an interaction between feature dimensionality and cell population size. At short latencies, face coding appears to be low-dimensional: the population uses a shared axis or small set of axes optimized for face detection. In such a regime, even a small number of neurons can effectively span the relevant feature space. In contrast, long-latency coding involves higher-dimensional, more detailed tuning required for identity discrimination. This higher-dimensional space cannot be adequately spanned with only a few neurons, but performance improves as more neurons are added.

To confirm this interpretation, we conducted a simulation of the face decoding task that systematically varied the number of neurons and the dimensionality of the feature space. As shown in **Fig. 5e**, the tradeoff between feature dimensionality and population size accurately reproduces the pattern observed in the neural data, supporting the idea that the interaction between these factors underlies the effect. We explain this finding in the manuscript:

"The pattern of recognition performance in Fig. 5d exactly matches what one would expect if the short latency response were specialized for face detection and the long latency response for face discrimination. For face detection, cells only need to represent a set of dimensions diagnostic of all faces. Tuning to a smaller number of dimensions leads to increased robustness due to redundancy in tuning. In contrast, for face discrimination, cells need to represent a larger set of dimensions enabling fine differentiation. Simulation of this tradeoff (redundancy vs. diversity) produced the same pattern of responses as we observed in the actual cell population (Fig. 5e)."

12. Lines 490-494. I don't see why category information should necessarily be expected to rise before identity information. This is the case if neurons are tuned to large differences, but not if they encode more detailed and specific (face) features.

We thank the reviewer for pointing this out. We agree that the original phrasing is too absolute, we have thus deleted this sentence.

13. Lines 504-509. In this paragraph the authors challenge the feedforward processing view of invariant object (face) recognition, but I believe no much can be said in this respect, because this paper does not deal with invariance – i.e. the authors show a change of code from face detection to face discrimination but this does not necessarily apply to how invariant recognition is processed by the brain.

The reviewer notes that our paper does not explicitly address the emergence of invariance and questions how our findings challenge the feedforward processing view of invariant object (face) recognition. We appreciate this feedback and apologize for the confusion, which stems from our imprecise use of terminology.

Our aim is not to challenge any ideas concerning generation of invariance per se, but rather to question the general framework of core object recognition as a feedforward process. Core object recognition refers to the recognition of objects under basic, simple conditions—i.e., without occlusion or ambiguity, involving a single object that is easily differentiable from the background (DiCarlo et al. Neuron 2012). Importantly, in this

framework, "invariance" refers to robust recognition under varying background or clutter, not necessarily the invariant representation of faces themselves.

In our study, stimuli were presented in the simplest possible setting—a single object on a uniform background—thus clearly falling within the scope of “core object recognition.” Despite this, we observed unexpected dynamics that suggest the involvement of recurrent processing. This finding directly challenges the prevalent belief that *core object recognition in IT cortex is purely feedforward* (cf. Kar et al. Nature neuroscience 2019). Even under these minimalistic conditions, our results demonstrate that additional processing dynamics occur, suggesting that core object recognition may not be as straightforwardly feedforward as traditionally thought. We now clarify this in the manuscript:

“Our results challenge a prevailing view that “core object recognition”—the ability to rapidly recognize objects within 200 ms despite substantial variation in appearance—can be explained primarily by feedforward processes³⁷. **They also call into question the dominant view that deep networks, widely considered the best models of IT cortex, are sufficient to capture core object recognition process in IT. The recognition of clear, isolated faces in the absence of any background clutter is a textbook example of core object recognition.** Yet, we find that even this process is consistently accompanied by a rapid switch in neural code within the key brain structures mediating face recognition³⁰.”

Minor points:

14. Line 545. “Indeed we found...” Why we? Was any of the authors involved in the study of reference 54?

We apologize for the confusion in our original phrasing. The sentence was intended to highlight evidence from our present findings: “We found that animal bodies with grayed-out heads evoked strong responses across the face cell population” (Fig. 3). Reference 54 was cited to support this observation, as it also demonstrated that face cells respond strongly to bodies with grayed-out heads. To address the ambiguity, we have revised the sentence as follows:

“Indeed, we found that animal bodies with grayed-out heads evoked strong responses across the face cell population (Fig. 3), consistent with earlier report⁵⁴, but these stimuli did not trigger the state change associated with actual faces.”

15. Line 238-240. Please edit.

We thank the reviewer for their careful reading and for identifying the grammar mistake. The sentence has been corrected to:

“Across the population, 62% of cells ~~that had~~ clearly flipped tuning in PC1-PC2 space (Extended Data Fig. 9a; angle $\geq 120^\circ$).”

Summarizing, in my view this is a very interesting paper, proposing the novel idea of a dynamic change of the neurons’ code to encode different aspects of the stimuli (face detection and face discrimination). This claim is well supported by the data and main results, but I believe the authors should improve the presentation to better convey this message and increase the impact of the paper.

We thank the reviewer again for their appreciation of our findings and extremely helpful comments!

Referee #2 (Remarks to the Author):

Shi et al. Rapid, concerted switching of the neural code in inferior temporal cortex

In this study, the authors record neuronal activity in two regions within the macaque monkey inferior temporal cortex. These regions are purported to represent “faces”, according to functional magnetic resonance imaging (fMRI) measurements. The authors use photographs of “faces” and “non-faces” and fit the neuronal responses using the activations of an old neural network (Alexnet). Based on the weights of these fits, the authors argue that the representation of faces is different from that of non-faces and that this representation rapidly changes over time after stimulus presentation.

(1) This study follows a traditional decades-old approach whereby investigators decide on an arbitrary set of stimuli and present those images to a monkey. Such studies have multiple intrinsic biases. In the context of the current work, some of the questions include: What is a “face”? What is a “non-face”? Are human “faces”, monkey “faces”, dog “faces”, coin “faces” all the same? How controlled are the stimuli for possible confounding factors (the list is long: contrast, textures, intermediate features in neural networks, other features of bottom-up or even top-down saliency)? How many “faces” and “non-faces” are shown? Conclusions depend strongly on these arbitrary decisions but the current study lacks any type of even minimal controls, definition, let alone justification for any of these choices or evaluation of how conclusions depend on these arbitrary choices.

We agree with the reviewer that prior studies on “faces” often rely on human-curated face photographs based on human semantics. This raises the question of (1) whether our findings depend on arbitrary human classification of faces, and furthermore (2) whether the findings are truly tied to the concept of faces or are merely driven by lower-level statistical features that happen to be common in face images.

Independence of results on human classification. While our original axis change metric was computed using *groups* of images, axis change should also be viewed as a response characteristic of cells to *specific images*. Thus, inspired by the reviewer’s comment above as well as a later comment, we now define a *single-stimulus axis change score* (see our response to comment #3 below for details). This metric assesses whether a single image triggers axis change in the population by calculating the rank correlation between early and late time windows of the population’s responses to that image. This metric is *completely independent of any human-defined classification of images*. Furthermore, this single-stimulus axis change score offers a novel and objective approach to address the fundamental question: What is a “face” to face cells? Specifically, for any IT neuron, we can infer its most preferred category by determining whether a particular type of object triggers axis change.

Inability of low-level features to explain results. The reviewer also points out that face and object images have distinct low-level statistical features. To determine whether axis change is driven by the high-level categorical aspect of face images or merely by unique lower-level features of face images, we first plotted the distribution of various low-level

features (including contrast, texture contrast, texture correlation, texture energy, texture homogeneity, saliency mean, saliency std) of the face and object images that we showed to the monkey, and identified 115 face and 115 object images that had maximum overlap (**Reviewer Fig. 10a**). We then trained decoders to predict single-stimulus axis change scores using each of these low-level features, plus the categorical labels (**Reviewer Fig. 10b**). The categorical distinction between faces and objects stands out as the strongest predictor of axis change scores, supporting the idea that axis change is not driven by low-level features.

Robustness of results to stimulus set size. Finally, we tested whether the number of stimuli affects the observed axis change patterns. In **Reviewer Fig. 11**, we repeated the analyses using randomly subsampled halves of the face or object stimuli and observed consistent results, confirming the robustness of the findings.

Overall, our findings suggest that axis change is a robust and high-level phenomenon that reflects the neural coding of face versus object categories. Moreover, our analyses demonstrate that these results are not driven by arbitrary stimulus choices or low-level image features but instead reflect genuine categorical distinctions.

Reviewer Figure 10. Semantic categorization of face vs object images best predicts the single-stimulus axis change score compared to other low-level features. **(a)** Distribution of various low-level features (contrast, texture contrast, texture correlation, texture energy, texture homogeneity, saliency mean, saliency std) of face (purple) and object (green) images. **(b)** Accuracy of using the low-level features (black bars) and the category label (red bar) of each overlapping face or object image to predict single-stimulus axis change scores.

Reviewer Figure 11. *Subsampling stimuli doesn't affect axis changing patterns. (a-c) Axis change results after subsampling half of object stimuli. (d-f) Axis change results after subsampling half of face stimuli.*

(2) To circumvent these challenges, several investigators have developed unbiased and systematic ways of studying tuning properties in visual cortex. Notable examples include Bashivan et al Science 2019, Walker et al Nature Neuroscience 2019, and Ponce et al Cell 2019. There are differences among these approaches, but they consistently show that the stimuli that trigger strong responses in neurons along the ventral visual cortex in macaques are not quite “faces”, or “chairs”, but rather complex feature combinations. Of note, Ponce et al 2019 record from similar areas to the ones examined in the current paper. Additionally, subsequent studies by Bardon et al PNAS 2022 showed that humans do not perceive the stimuli derived in such an unbiased manner as “faces”.

We thank the reviewer for suggesting these relevant studies. We are familiar with this literature and have, in fact, applied similar optimization methods within the face patch. However, during these investigations, we identified a significant limitation with this type of approach: If a face cell has two distinct encoding axes at short and long latencies, time-averaged response maximization will be dominated by the short-latency axis due to the larger response amplitude at early time points. As a result, this method fails to identify the stimuli that maximize the late-latency axis. More importantly, *a key point of our study is that strong responses do not necessarily translate to better discrimination.* In fact, the opposite is often true. Better discrimination occurs when face cells switch to face axes,

even though this occurs after the period of strongest firing in response to faces. Optimization-based techniques can potentially generate stimuli that elicit strong responses without triggering axis change, thereby missing the essential neural mechanism we describe.

Furthermore, we are not claiming that only realistic faces can trigger axis change. As we demonstrate throughout our manuscript and rebuttal, the axis change metric can be applied to any set of stimuli. This flexibility ensures that our findings are not constrained by the specific choice of image set.

Lastly, we think the study of Ponce et al. (Cell, 2019) powerfully confirms that the optimal images of cells presumed to be face selective based on classic criteria are indeed face-like when mapped using unsupervised optimization techniques (**Reviewer Fig. 12**).

Bardon et al. (PNAS, 2022) conducted a series of faceness rating experiments in human subjects and reported that, while images optimized for face-selective neurons not perceived as realistic faces, they were nevertheless rated as more face-like than those optimized for non-face neurons. We do not claim that axis change only occurs for realistic faces—our expectation is that any stimulus that projects strongly onto the face quadrant of object space should elicit axis change.

[REDACTION]

[REDACTION]

(3) There are multiple problems that pervade the literature claiming that “faces” are special. These problems have been extensively discussed in the literature. I will only focus here on problems that are particularly relevant for the current study. 3.1 First, there is no clear definition of what a “face” is. From caricatures, to cartoon movies depicting cars with faces, to pareidolia, to the face of an ant to the face in a coin, to clocks that have shapes like faces, to many other versions, the definition remains ambiguous and it is unclear whether any conclusions would extrapolate to such stimuli. There have been other works that study some of these images but none of them are included as controls in the current work, casting a doubt on whether the current study can really draw any conclusion about “faces”.

We thank the reviewer for bringing up this important question. We first confirmed that axis change is not limited to human faces. We constructed a monkey face image set and presented these stimuli to the monkey. We observed the axis change phenomenon with monkey faces as well (**Reviewer Fig. 13**), supporting the generalizability of this metric to at least one other category of faces.

But we understand that the reviewer’s core concern is not that we didn’t test a sufficient diversity of image sets, but rather that it is inherently difficult to define exactly what a “face”

is. To address this foundational question, we introduce the *single-stimulus axis change score*—an objective, response-based metric to evaluate whether an image is treated as a "face" by a population of face-selective neurons. To compute this score, we measured the Spearman rank correlation between population responses in early (60–80 ms for ML responses) and late (100–120 ms for ML responses) time windows for each image. To aid interpretability, we then trained a simple 1D classifier to distinguish face from object categories based on this correlation value. Note this classifier preserves the relative ranking of the raw correlation values across images. The single-stimulus axis change score is defined as the classifier's predicted probability that a given image is a face. Using this score, we can systematically assess the "faceness" of *any* image—even highly ambiguous or unconventional cases.

To first validate this score, we evaluated the faceness of clear face or non-face images shown to the monkey. In the dataset used in our manuscript, object images and face images exhibited distinct differences: nearly all face images had negative population response correlation, while object images had positive population response correlation (**Reviewer Fig. 14a**).

We then used the score to evaluate the faceness of more ambiguous stimuli including human faces, monkey faces, dog faces, and pareidolia images. The results are summarized in **Reviewer Fig. 14b**. Human and monkey faces were found to be the most face-like, whereas dog faces, pareidolia images, and objects were less so. For pareidolia images (**Reviewer Fig. 14b, Pareidolia**), there was a wide range of variance in the scores, consistent with the diverse nature of pareidolia images, which ranged widely from more face-like to more object-like. Some example Pareidolia images and their single-stimulus axis change score are shown in **Reviewer Fig. 14c**.

Overall, these findings show that (1) quantifying axis change does not depend on any pre-defined notion of faceness (or selectivity for any other category, for that matter), and (2) axis change provides a powerful new tool for evaluating faceness across a wide range of stimuli.

We have added these lines to the manuscript:

"Given that face axis reversal is evident in the raw neural responses and occurs consistently across face-selective cells, we hypothesized that the response amplitude ranking among face cells should also reverse when the monkey sees a face. If true, this would allow us to assess tuning changes for individual images by tracking shifts in response ranking across the face cell population. To test this, we define the single-stimulus axis change score (see Methods), which measures the rank correlation between population responses in early and late time windows. We confirmed that this score was strongly category-dependent: faces consistently triggered rank changes, whereas objects did not (Extended Data Fig. 16a). Applying this metric to other image types, including monkey faces, dog faces, and pareidolia images, we found that face cells showed significant tuning change for most monkey faces, a weaker effect for dog faces, and high variability across pareidolia images (Extended Data Fig. 16b)."

[REDACTION]

Reviewer Figure 13. Face cells also show axis change to monkey faces. (a) Example monkey face images. (b) Object and monkey face axis direction in PC1-2 space. (c) Matrices of mean cosine similarities across the population between (object, object), (face, face), and (face, object) axes (computed using all 60 dimensions) for different pairs of latencies.

Reviewer Figure 14 (replicated as Extended Data Fig. 16). Single-stimulus axis change score of various stimuli. **(a)** Cross-time correlation of population response to every stimulus, calculated from the ML responses in the main session. **(b)** Histogram of single-stimulus axis change score to various categories of stimuli from another session which we presented human faces, monkey faces, pareidolia, dog faces and objects. **(c)** Single-stimulus axis change scores of example human face, monkey face, dog face, object and Pareidolia images. The scores are annotated below each image.

3.2 Given problem 3.1, to what extent are the results—two discrete neural codes and a binary, gate-like switching between the codes—generalizable to more ambiguous stimuli? The study worked hard to distinguish the ill-defined “face” category. (I could not help noticing the peculiar animal images with faces blocked out.) Besides the pareidolia images and synthetic stimuli mentioned above, there are also studies showing that single eyes can drive the activity of neurons in the PL face patch (Issa & Dicarlo, J Neurosci. 2012). Should “face” neurons respond categorically to all these ambiguous stimuli as either non-faces or faces? Or, does the neural code depend on the stimulus in a

continuous, not binary fashion? Or, more likely, the neural code represents general features that should not be separated as “faces” and “non-faces”.

We thank the reviewer for raising this fascinating and important question. If faces trigger a switch in axis from object axis to face axis, and objects do not, the natural follow-up question is: what happens with ambiguous stimuli? Does the axis change occur categorically or continuously in response to degraded or partially ambiguous faces? To address this, we conducted additional experiments to degrade face images and assess how these degradations influence axis change behavior.

We rendered 200 distinct face identities, each with three types of degradation (noise, occlusion, and Mooney faces) (**Reviewer Fig. 15b-d, first column**). These degraded faces were presented to monkeys while recording from face patch ML using Neuropixels probes. To determine whether the face axes for these degraded faces behave differently than those for clear faces, we calculated the degraded face axis for face-selective cells and plotted the correlation between the time-resolved face axis and each cell’s averaged object axis. By comparing the time course of these correlations with those observed for clear faces, we evaluated the effects of degradation on axis change dynamics.

(1) Degradations with minimal impact (noise and occlusion)

For noisy and occluded faces, the raw response time course closely resembled that of clear faces (**Reviewer Fig. 15b, c, second and third columns**). The face-object axis correlation time course also mirrored that of clear faces (**Reviewer Fig. 15b, c, fourth and fifth columns**), indicating that these types of degradations did not significantly disrupt the axis change behavior or face responses.

(2) Degradations with delayed axis change (Mooney faces)

For Mooney faces, the response amplitudes were much smaller than those for clear faces (**Reviewer Fig. 15d, second and third columns**), suggesting that the face signal was weaker. However, the face axis still switched direction, albeit more slowly, and at more variable time across units (**Reviewer Fig. 15d, fourth and fifth columns**), indicating that the population required more time to resolve the identity of Mooney faces. This delay in axis switching suggests that the neurons could still process the Mooney stimuli as faces but required additional time and processing to do so.

These results demonstrate that the timing of axis change may depend on the amount of identity-related information available in the stimulus. The face cell population may require more time to switch axes when the face is degraded where ambiguity takes longer to resolve. This suggests that axis change reflects a recognition process that spans both clear and degraded face recognition—where even clear face recognition involves rich dynamics, but the process is faster and more reliable due to the higher information content.

We have added the following sentence to the manuscript:

“Conversely, some degraded face stimuli elicited relatively weak responses but still triggered axis change (Extended Data Fig. 26).”

Reviewer Figure 15 (Replicated as Extended Data Fig. 26). Response and axis correlation time course of degraded faces. (a) Time course of responses and axis correlations for clear faces. First column, an example clear face image. Second column, PSTH of the cell-averaged response to each clear face. Third column, average response time course across all units and all clear face identities. Fourth column, time course of the correlation between sliding-window face axis and the cell's average object axis, averaged across cells. Fifth column, the time-varying face – averaged object axis correlation time course for each unit. (b-e) Same as in (a) but for the other categories: occlusion, noisy, Mooney faces, non-face objects. For the third and fourth columns, target face category is shown in blue, clear face in gray.

3.3 In addition to the “low-level” problems alluded to in question (1), face photographs are very homogeneous within image space. One can debate about how to define such homogeneity. For the sake of argument, one could define this homogeneity as the standard deviation between stimuli in pixel space, or even better, in a “high-level” feature space like conv5 or fc6 in the Alexnet layers that the authors study (any other more modern neural network would work as well for this definition). This homogeneity in stimulus space is likely to drive many of the conclusions in the current study. Consider another group of stimuli (e.g., chairs). Imagine that you match the degree of homogeneity in this group (chairs) to the one for “faces” in the current study. Imagine matching “low-level” image properties as well. Imagine that you also match the large bias in the number of stimuli, that is, use about 1,500 chairs and a heterogeneous set of 1,500 non-chairs. Then select those neurons that have $d' > 0.2$ for chairs versus non-chairs. Under these experimental conditions, one would likely obtain the same results for chair neurons. One could go on to do this for tables, rhinoceros, clouds, or any group of interest.

We understand the reviewer is asking whether the phenomenon we describe is truly specific to faces—or whether similar results might be observed in neurons selective for

other object categories (e.g., chairs), particularly if those categories were matched in terms of within-category homogeneity and stimulus sampling.

To directly address this, we conducted new experiments using a broad set of novel object categories—including spiky objects, stubby objects, animals, and monkey bodies—and recorded from IT regions that showed selectivity for these categories based on fMRI localizers (Bao et al., Nature 2020). We found that neurons selective for monkey bodies, stubby objects, and spiky objects did exhibit axis reversal for their preferred categories. However, compared to face-selective neurons, the reversal was generally weaker and less consistent (compare **Reviewer Fig. 16d, h, l** with **16t**). In contrast, neurons selective for animal bodies showed no evidence of axis reversal; their tuning remained stable throughout the trial (**Reviewer Fig. 16m–p**). For further detail, please refer to our response to Reviewer 1, point #4.

Additionally, because we can measure axis change on a per-stimulus basis (**Reviewer Fig. 14** above), we believe it is unlikely that the observed dynamics are driven solely by homogeneity within stimulus groups. If axis change were simply an artifact of within-group feature similarity, we would not expect to see robust reversal for certain object categories and not others. Together, these findings support the conclusion that axis change is a category-dependent phenomenon—not just a consequence of stimulus statistics—and that faces, while not unique in exhibiting this effect, elicit it in the most robust and concerted manner.

Reviewer Figure 16 (Replicated as Extended Fig. 17). Category-selective units outside face patches show various degrees of axis change at long latency. **(a)** Top: Example images from monkey body image set. Bottom: Distribution of monkey body stimuli and general non-monkey body object stimuli features in PC1-2 of the object space. **(b)** Time averaged responses of all monkey body units (peak $d' > 0.1$) to monkey body and non-monkey body stimuli. **(c)** Distribution of monkey body (left, purple) and object (right, green) axes projected onto the top two dimensions of the 60-d object space, for early (top) and late (bottom) latency responses of monkey body cells. (Only units with peak $d' > 0.2$ are used in this panel.) **(d)** Histogram of correlation of monkey body (purple) / object (green) axis between early and long latency for each unit. (The early and late time windows were defined individually for each unit: the early window was centered on the unit's peak response time, and the late window was offset by 20 ms following the early window.) **(e-h)** Same as a-d for stubby cells. **(i-l)** Same as a-e for spiky cells. **(m-p)** Same as a-d for animal cells. **(q-t)** Same as a-d for face cells outside face patches.

(4) These issues are exacerbated by the poor block designs. Block designs are nefarious (see for example Li et al IEEE TPAMI 2021 for one of many example discussions on these issues), introducing confounds related to adaptation, baseline changes, attentional changes, potential learning effects, and many other issues that are hard to control. Stimulus randomization is critical when attempting to make any statements about different stimuli when there is no reason to block stimuli.

We thank the reviewer for highlighting the potential confounds introduced by block designs. We agree these are significant concerns. In our original manuscript, we demonstrated that the axis change phenomenon is robust across two monkeys, one presented with stimuli in a block design and the other with stimuli in an interleaved manner.

To further address this concern, we conducted a new recording session in which the same stimulus set was presented in both block design and interleaved modes within the same session. The axis correlation time courses for the two presentation modes were nearly identical (**Reviewer Fig. 17**), suggesting that the observed axis change pattern is not influenced by the presentation mode (block vs. interleaved). These findings provide strong evidence that the axis change phenomenon is a robust feature of face-selective cell activity and is not confounded by the use of a block design.

We have added the clarification to the manuscript:

"However, we observed the same pattern of results when the face and object stimuli were randomly interleaved (Extended Data Fig. 2a-d, Extended Data Fig. 6). This suggests that the axis difference is not related to prior expectations/adaptation and is instead purely driven by the current stimulus."

Reviewer Figure 17 (replicated as Extended Data Fig. 6). Axis change dynamics is not affected by presentation mode. (a-c) Axis mapping results with block design stimulus presentation. (d-f) Axis mapping results of the same neurons with interleaved stimulus presentation.

(5) Another difficulty in the interpretation of the current results is that most of the conclusions are seen through the lens of the weights in the fitting procedure. That is, the conclusions are not directly about the tuning properties of neurons in cortex, but rather about the weights in this fitting procedure. Consider a model that perfectly fits the neuronal responses across any possible stimuli. Then, it seems that one can derive interesting conclusions about the neuronal responses from studying the weights in the model. Now consider a model that only poorly explains neuronal responses and does not extrapolate across novel stimuli. Then conclusions derived from such a fitting procedure would make little sense. The current situation is neither one extreme nor the other. The model partially fits neuronal activity. Therefore, at best, the conclusions derived from these weights at best can be interpreted as partially reflecting neuronal responses. The authors themselves acknowledge similar concerns: AlexNet is not an optimal model (lines 98, 566), AlexNet itself is a highly nonlinear “blackbox” model (lines 569–570), etc.

We appreciate the reviewer’s thoughtful concerns regarding the limitations of model-based interpretations and acknowledge that no model is perfect. However, as long as a model demonstrates explanatory or predictive power, it can serve as a useful proxy to simplify complex problems and reveal patterns that may be challenging to discern directly from raw data. Of course, the conclusions drawn from such models must be carefully evaluated on a case-by-case basis, and we thank the reviewer for raising this important point.

The core conclusion of our study is that the encoding axis changes over time for faces, but remains stable for objects. It is hard to imagine why we would observe axis change for faces if the *raw neural tuning* for faces were fixed.

In fact, we observed tuning reversal directly in the raw neural responses, even without applying axis fitting. These dynamics closely match those revealed through axis-based analyses and further support our main conclusion (**Reviewer Fig. 18**). (For additional details, please see our response to Reviewer Question #9.)

Reviewer Figure 18 (Replicated as Extended Data Fig. 15). Tuning change computed directly from raw population responses. (a) Matrices of mean correlations across the population responses between (object, object) (left) and (face, face) (right) for different pairs of latencies in monkey A, ML. (b) Left: Purple (green) distribution shows correlations between face (object) responses at short (80-100 ms) and long-early latency (120-140 ms) across units. Right:

Correlations between face (object) responses at long-early and long-late latencies (160-180 ms). (c-d) Same as (a-b) for Monkey A, AM. (e-f) Same as (a-b) for Monkey J, ML.

We have added the following to manuscript:

"Direct assessment of tuning from the raw neural responses to faces and objects also yielded results consistent with the axis-based analyses. We computed mean similarity matrices of (object, object) and (face, face) responses across pairs of latencies and found that object responses stabilized early, whereas face responses became anticorrelated with their early-latency responses before stabilizing at a later latency—mirroring the axis change dynamics (Extended Data Fig. 15)."

(6) To the extent that there is something special about the two areas picked by the authors based on fMRI, it would be important to document whether there are any differences between those neurons inside and outside those so-called "face patches". Additionally, it would be important to repeat analysis for neurons that are considered "non-face" neurons within those patches. Thus, there would be at least four different groups. "Face patch face neuron", "face patch non-face neuron", "non-face patch face neuron", "non-face patch non-face neuron". After doing all of this, one might be able to conclude that there is some topography in the representation of information in IT cortex, consistent with extensive studies of the spatial organization of responses in IT cortex (for a review on this, see Tanaka Ann. Rev. Neurosci. 2006).

We thank the reviewer for raising this important question. Documenting the behavior of cells outside face patches, as well as distinguishing between face and non-face cells, is indeed crucial for understanding the spatial organization of information representation in IT cortex.

Following the reviewer's suggestion, we conducted additional experiments and recorded responses from both face-selective and non-face-selective cells outside face patches. We found that the face selectivity of face cells outside face patches was generally weaker than that of face cells within face patches (**Reviewer Fig. 19a vs e**). Despite this, the axis change phenomenon was observed in face cells both inside and outside face patches. However, the axis change in cells outside face patches was slightly delayed, about 20 ms later than face cells inside the face patch (~90 ms for face cells in face patch, and ~110 ms for face cells outside face patch, compare 3rd column of **Reviewer Fig. 19b** and **Reviewer Fig. 19f**), potentially because these cells inherit their tuning from face cells within face patches via horizontal connections.

Non-face-selective cells outside face patches did not exhibit axis change in response to faces, and their face axes remained unstable throughout the trial—unlike the consistent axis dynamics observed in face-selective cells (**Reviewer Fig. 19d**).

Although non-face-selective cells within face patches are relatively rare, their behavior is intriguing. While they do not show axis reversal like face-selective neurons, they do exhibit a strong initial alignment of their face axes to their object axes (**Reviewer Fig. 19h**, right panel). This suggests that they may share the same face detection mechanism as nearby face-selective cells.

Together, these results demonstrate that axis change in response to faces is a robust phenomenon confined to face-selective neurons, regardless of whether they are located

inside or outside face patches. However, the timing and strength of the effect may be modulated by the degree of face selectivity and spatial organization within IT cortex. These findings complement our main results and align well with previous studies on the spatial structure of categorical responses in IT.

We have added the following to the manuscript:

“Notably, face-selective neurons showed the strongest and most temporally concerted axis change behavior. Furthermore, we found that even face-selective neurons located outside canonical face patches exhibited axis change, albeit with slightly delayed timing compared to neurons inside face patches (Extended Data Fig. 18), raising the possibility that their dynamics may be inherited through recurrent interactions with face patch circuits.”

Reviewer Figure 19 (Replicated as Extended Data Fig. 18). Axis dynamics of face-selective and non-face-selective units inside and outside face patches. (a) Histogram of peak d' for face-selective cells recorded outside face patches in IT cortex. (b) Matrices of mean cosine similarity across the population, comparing (object, object), (face, face), and (face, object) axes at different time points. Red arrow on top of the rightmost matrix indicates the face axis switch time (~110 ms). (c, d) Same as (a–b) but for non-face-selective cells outside face patches. (e, f) Same as (a–b) but for face-selective cells inside the ML face patch. Red arrow on top of the rightmost

matrix indicates the face axis switch time (~90 ms). (g, h) Same as (a–b) but for non-face-selective cells inside the ML face patch.

(7) The current study finds fewer “face” neurons than what was claimed in a previous study (roughly 50% versus more than 90% in previous work by Tsao et al (Science 2006). What is remarkable is that the threshold to define a “face” neuron based on d' is much lower in this study compared to the previous study. Such a lower threshold should lead to a much higher proportion of neurons considered to be “face” neurons. The result is the opposite. The authors quickly dismiss this challenge by saying that the electrodes record from neurons outside so-called “face” patches. If this is the case, it would be important to document which neurons are considered to be inside versus outside (according to anatomical criteria, without looking at the neuronal responses to “faces” versus “non-faces” to avoid circular arguments), then show the neurons considered to be “face” neurons inside and outside and then perform the analyses in the paper for “face” neurons inside patches, “face” neurons outside patches, “non-face” neurons inside patches, “non-face” neurons outside patches as noted in the previous question. Related to this question, the work in Bardon et al PNAS 2022 showed that the differences between neurons inside and outside so-called “face” patches are minimal.

We thank the reviewer for highlighting this issue and would like to clarify a key point regarding the threshold used to define “face” neurons. The peak $d' \geq 0.2$ criterion employed in our study is, in fact, stricter than the previously used $FSI \geq 0.33$ criterion in our case. For the main session, only 39.63% of units were selected as face neurons using the d' criterion, compared to 47.47% when using the older FSI criterion. Thus, it is not accurate to say that we have loosened our selection criteria.

(For additional context, see **Reviewer Fig. 3** for a summary of face cell yields across 67 Neuropixels sessions including many sessions that were not part of the current study. In brief, the proportion of face-selective cells within face patches was relatively stable across sessions, averaging ~75% in ML and 81% in AM, whereas the proportion outside face patches was considerably lower, averaging ~28% in ML and 35% in AM.)

The reviewer raises an important point regarding potential differences in neuronal behavior inside versus outside face patches. To address this, we conducted additional analyses as suggested, examining the axis change phenomenon across four groups of neurons: (1) face-selective neurons inside face patches, (2) face-selective neurons outside face patches, and (3) non-face-selective neurons outside face patches. Note there are no non-face-selective cells inside the face cell boundary. This analysis differs from the analysis for the reviewer’s previous comment, as here cells from all four categories were confined to those recorded on trajectories targeting face patches.

We carefully defined the “face patch” range as the region along the probe with a high concentration of neurons exhibiting strong face selectivity (**Reviewer Fig. 20a**, between red boundary lines), identified as the longest continuous stretch of units with $FSI \geq 0.33$, allowing for small gaps of three units or less below threshold. For neurons outside

this range, we classified cells as face-selective if peak $d' \geq 0.2$, and as non-face-selective if peak $d' \leq 0$. Notably, no non-face-selective neurons were identified within the face patch region in the main session reported in the paper.

We found that face-selective neurons both inside and outside face patches exhibited axis change (Reviewer Fig. 20b-g). However, we note that classification of neurons as outside versus inside a face patch from the same penetration is very tricky and depends a lot on the exact criteria. We think a much more meaningful comparison is between face cells inside face patches and face cells recording in trajectories targeting other parts of IT nowhere in the vicinity of a face patch, shown in answer to the reviewer's previous question (Reviewer Fig. 19a, b).

In contrast, non-face-selective neurons outside face patches show no axis change behavior in response to faces (Reviewer Fig. 20h-j). Together, these results reinforce that axis change to faces occurs specifically in face-selective neurons, regardless of anatomical location.

Reviewer Figure 20. Axis dynamics of face-selective and non-face-selective units inside and outside face cell boundary. (a) Peak d' of all visually responsive units along the probe in ML face patch, red dash lines indicate the boundary of face-cell enriched area. (b) Orange plot of the face cells' time averaged responses to our screening set. (c) Matrices of mean cosine similarity across the population, comparing (object, object), (face, face), and (face, object) axes at different time points for face cells within the face cell boundary depicted in (a). (d) Mean response time course to each face and object stimulus, averaged across cells and trials for face cells within the face cell boundary. (e-g) Same as (b-d) but for face-selective cells outside the face cell boundary. (h-j) Same as (b-d) but for non-face-selective cells outside the face cell boundary.

(6) It is difficult to evaluate the major conceptual advances claimed in the study. Some difficulties arise from the author's phraseologies that are not clearly defined. First, all claims center on there being different neural codes. Thus, an explicit definition of what constitutes a (singular) neural code is in order. I don't believe such an explicit definition is found anywhere in the paper. The abstract opens, "[...] the concept of neural coding through tuning functions. According to this idea, neurons encode stimuli through fixed mappings of stimulus features to firing rates." This definition is not specific enough. A face and a non-face contain different stimulus features by definition, so a general, singular "neural code" can assign the two images different firing rates, and therefore cannot support a conclusion of different neural codes. The paper needs a clear definition of a neural code that can change before the paper can give meaning to any claims about different, switching, or dynamic neural codes.

We thank the reviewer for pointing out this potential confusion. We have clarified the concepts and definitions and the changes are shown below.

In the introduction, we now mathematically define what is commonly considered the best model of IT neurons, specifically the linear axis model:

"Specifically, IT cells can be modeled as computing linear combinations of features represented in late layers of these networks $r = \vec{c} \cdot \vec{f}$, where r is the response of the IT cell, \vec{f} is a vector of object features given by the DNN representation, and \vec{c} defines the 'preferred axis' of the cell."

In the results section, we further elaborate on this approach:

"For each face cell, we used responses to the non-face object stimuli (averaged over the time window 50–220 ms) to compute a preferred axis, the *object axis*, by linearly regressing responses of cells to the 60-d feature vectors corresponding to different objects (Fig. 1b). Similarly, we used responses to the face stimuli to compute a preferred *face axis*."

These definitions establish what we consider a singular, fixed neural model for a face cell, a fixed set of coefficients that linearly maps the object features to neural responses. The preferred axis (\vec{c}) characterizes the tuning of the cell and allows us to compare responses to faces versus objects within this framework.

Regarding the reviewer's question on how to define and interpret different, switching, or dynamic neural codes, we believe our clarification resolves any ambiguity. As defined above, if the preferred axis (\vec{c}) changes over time while the image features (\vec{f}) remain fixed, this necessarily indicates a change in the neural code. Since we observe that the face axis changes over time whereas the object axis remains stable, we believe it is justified to conclude that the face model and object model used by face-selective neurons are distinct. Additional rationale supporting this interpretation is provided in our response to the next question.

In summary, we believe the definitions we provide—particularly the use of the preferred axis to characterize tuning—clearly define what we mean by a neural code, as well as how we interpret dynamic and distinct neural codes in the context of our findings.

(8) The paper implies that the operational definition of a neural code in this work is the weighting vector in (dimension-reduced) AlexNet fc6 space. Defined thus, a neural code can change. However, this definition is severely limited. For example, even a hypothetical, static (i.e., not varying in time) neural code that is piecewise linear (and globally nonlinear)

in the fc6 space will comprise different neural codes for the different linear regions. This constrained definition of a neural code makes the conceptual conclusion much less widely applicable than it first seems. Within the visual domain, see concern (4) that current models are imperfect models of neurons. Since IT responses are not perfectly linear in AlexNet fc6, the model-based results can only be interpreted as, “an approximate neural code approximately flips.” For the broader implications, it is unclear how one should adjudicate the sameness, differentness, and change in a neural code for logic (line 584), for which an artificial model that linearly maps onto neurons has not been identified.

We appreciate the opportunity to clarify and refine our definition of a neural code. We understand the reviewer’s concern that a nonlinear or piecewise linear model could describe a cell’s responses to both faces and objects. However, such a model would offer little advantage in understanding the cell’s function. For example, if we modeled cells with a piecewise model, the conclusion might become something like: “The face portion of the model changes its tuning in response to face images, while the object portion remains unchanged for object images.” This framing would make the result unnecessarily convoluted and less insightful. While one can always absorb image features into a highly nonlinear model, we believe that two linear models more parsimoniously describe the cell’s behavior. This approach provides a clearer interpretation, showing that the neural code changes in a meaningful and interpretable way.

Operationally, we define the neural code in this study using AlexNet PCA axis weights. This choice is supported by prior research demonstrating the utility of convolutional neural networks (CNNs), such as AlexNet, for modeling inferotemporal (IT) cortex responses. For example, Yamins et al. (2014) demonstrated that CNN-derived features can robustly predict IT neural responses, establishing these models as among the “best models of IT” currently available.

While we acknowledge the limitations of these models, we believe our approach emphasizes simplicity and interpretability while leveraging well-supported tools to capture the functional specificity of IT responses.

(9) It seems that the intervening AlexNet model is dispensable when comparing within an image category. In this case, tuning direction inversion may be tested directly using neuron responses across individual images, without using neural network fitting, avoiding concern (4). Thus, it is relevant for the authors to add versions of Figs. 3b (left two columns), 4c, e, and related supplementary figures analyzed directly using responses without neural network fitting.

We thank the reviewer for this insightful suggestion. While fitting axes to responses allows us to decompose the response into separate feature dimensions and track changes within these dimensions, it is equally important to analyze the raw responses directly to validate our findings. To address this, we redid all our main analyses using raw responses.

First, we conducted an analysis where we calculated the correlation of population responses (rather than axes) across pairs of time points. Overall, the similarity matrix for

raw responses closely resembles the axis-based results: For the ML face patch in both monkeys A and J (**Reviewer Fig. 18a, e**), face responses show clear changes early on, stabilizing after approximately 100 ms. In contrast, object responses stabilize earlier, around 80 ms. These response stabilization times align precisely with the axis change times detected using axis fitting. For the AM face patch in monkey A, the dynamics are delayed by approximately 20 ms, again consistent with the axis dynamics. (**Reviewer Fig. 18c**). Next, we quantified these response dynamics across cells by plotting histograms of correlation values between each cell's responses at different time windows. Specifically, we compared correlations between the early and late time windows and between the late and late-late time windows. Across both monkeys and both face patches, correlations for face responses are more negative between the early and late time windows compared to the late-late time windows (**Reviewer Fig. 18b, d, f, purple**), reflecting dynamic changes in face responses. In contrast, object responses show no significant difference between these time windows (**Reviewer Fig. 18b, d, f, green**), indicating stable representations. These findings confirm that the dynamic changes we observed using axis fitting are also present in the raw responses, further validating our conclusions. This analysis demonstrates that the underlying dynamics do not rely on the intervening AlexNet model and are intrinsic to the population responses.

Reviewer Figure 18 (Replicated as Extended Data Fig. 15). Tuning change computed directly from raw population responses. (a) Matrices of mean correlations across the population

responses between (object, object) (left) and (face, face) (right) for different pairs of latencies in monkey A, ML. (b) Left: Purple (green) distribution shows correlations between face (object) responses at short (80-100 ms) and long-early latency (120-140 ms) across units. Right: Correlations between face (object) responses at long-early and long-late latencies (160-180 ms). (c-d) Same as (a-b) for Monkey A, AM. (e-f) Same as (a-b) for Monkey J, ML.

(10) The notions of a neural code “use[d] to represent a [stimulus]” (lines 101, 115) and “stimulus-dependent switching” (lines 30, 100) are also ill-defined. Under the above operational definition, a neural code cannot be defined for a single stimulus, because linear fitting requires multiple stimuli. (Incidentally, it is for this reason that the neural code in Extended Data Fig. 19e has fewer dimensions than in the main figures.) Arguably, given a defined neural code (e.g., the “object axis”), one can evaluate its fit to a specific stimulus by comparing the predicted and observed firing rates. However, the paper has always evaluated neural code fit as R², an across-stimulus metric, in this paper. If a neural code cannot be defined for one stimulus alone, the study’s conclusion becomes more like, “The response differences among faces correlate with opposite AlexNet-fc6 features than the response differences among objects.” This altered conclusion may seem subtle, but it calls into question whether each and every face engages a stimulus-gated, domain-specific processing mechanism. This question also implicates the third major conceptual advance claimed in the discussion (lines 539--548). In other words, given an ambiguous image X (e.g., a pareidolia image, an animal with a visible face), just how exactly do the authors propose to concretely answer the question of whether that image is a face?

We appreciate the reviewer’s insightful comment and agree that only by being able to assess the axis change phenomenon to any single stimulus, can we conclude that the reported phenomenon is happening to each and every face encountered. As introduced in response to question #3, we addressed this by developing the *single-stimulus axis change score*. This metric, computed directly from population neural responses, quantifies the degree to which a single image triggers a shift in the population’s response ranking. In doing so, it provides a concrete, interpretable measure of whether a stimulus engages the face-selective processing mechanism—even in ambiguous cases such as Pareidolia images or animals with visible faces.

To validate this score, we first applied it to human face and object images. Nearly all face images received high scores, while object images received low scores (**Reviewer Fig. 14a** above), confirming that the single-stimulus axis change score is a robust and reliable measure of “faceness.” We then extended the analysis to other categories, including monkey faces, dog faces, and Pareidolia images (**Reviewer Fig. 14b** above). Human and monkey faces scored highest, while dog faces, Pareidolia images, and non-face objects showed lower faceness scores overall. Importantly, Pareidolia images exhibited high variability, consistent with the broad range of visual similarity to true faces among these stimuli.

While we acknowledge the inherent biases in stimulus selection, the introduction of this metric offers a scalable and objective tool for determining what qualifies as a “face” from the perspective of the face-selective neural population.

We have incorporated the results on single-stimulus axis change score into the manuscript:

In main text:

“Given that face axis reversal is evident in the raw neural responses and occurs consistently across face-selective cells, we hypothesized that the response amplitude ranking among face cells should also reverse when the monkey sees a face. If true, this would allow us to assess tuning changes for individual images by tracking shifts in response ranking across the face cell population. To test this, we define the single-stimulus axis change score (see Methods), which measures the rank correlation between population responses in early and late time windows. We confirmed that this score was strongly category-dependent: faces consistently triggered rank changes, whereas objects did not (Extended Data Fig. 16a). Applying this metric to other image types, including monkey faces, dog faces, and pareidolia images, we found that face cells showed significant tuning change for most monkey faces, a weaker effect for dog faces, and high variability across pareidolia images (Extended Data Fig. 16b).”

In methods:

“Single-stimulus axis change score

This metric quantifies whether a single image triggers an axis change in the neuron population. We first computed the Spearman rank correlation between the early (60–80 ms) and late (100–120 ms) population responses of all face-selective cells to each image. To improve interpretability, we then normalized the axis change score by training a simple classifier to predict face or object category labels based on the score. The classifier’s resulting probability of predicting a face served as the single-stimulus axis change score. This normalization fully preserves the ranking of the rank correlation among the images while allowing for a more intuitive interpretation. A higher score reflects greater “faceness” of an image.”

(11) The authors should specify in what exact sense they are claiming to “suggest/reveal a novel/new mechanism/form of neural representation/computation” (lines 29, 99–100, 408, 475). The mechanism of lateral inhibition seems able to predict qualitatively the main results: Faces that strongly excite the local population cause the population to inhibit itself, such that “average” faces (i.e., those that abruptly and strongly drive many face neurons) are initially preferred and later anti-preferred, whereas diffusely (across the population and/or time) or weakly activating faces become preferred to average faces in the late period. (Of course, what seems obvious is subjective.) The authors train a minimal RNN model to invert the input gradient, and the model learns a “simple structure consistent with lateral inhibition” (line 290). To me, this is an interesting albeit confirmatory result. As the authors themselves eruditely review, lateral inhibition is a well-established, widespread circuit motif (e.g., Carandini & Heeger, Nat. Rev. Neurosci. 2012) that has been specifically speculated to play an important role in IT representations (lines 572–576). Closer afield, Koyano et al. (Curr. Biol. 2021) studied face-neuron responses to faces and found a late-emerging (100–300+ ms) preference for caricatures that reverses an earlier (50–100 ms) preference for the average face against caricatures (their Fig. 4B). While the direct comparison of face- and object-tuning axes (which requires an intervening deep-net model) showing opposite axis directions is novel to my knowledge and contradicts a recent result (Vinken et al., 2023), and the point about axis inversion over time is perhaps more emphatically made than in prior publications, the overall concept of a dynamic (and even inverting) tuning axis is not new.

We thank the reviewer for their detailed comments and thoughtful discussion of related literature. As the reviewer noted, there have been prior studies on the temporal dynamics of face cell representations (e.g., Sugase et al., Tsao et al., Brincat and Connor), as well as late-emerging preferences for caricatures versus average faces (e.g., Koyano et al.,

Curr. Biol. 2021, although more on this in the next paragraph). However, our findings go well beyond these previous studies in several critical ways.

First, while prior work has documented the time course of information decoding from face cells, none has examined in detail how tuning functions themselves change across time. Most studies focus on decoding performance or category-level responses without addressing the underlying tuning dynamics. The results reported by Koyano et al. can be explained by trial adaptation rather than active feature tuning changes. Their experimental design exposes monkeys to average faces more frequently than caricatures, which likely introduces adaptation effects rather than revealing genuine changes in tuning function.

Third, our findings reveal a novel and stimulus-dependent axis change that has not been described before. Specifically, we demonstrate that face cell tuning aligns with a general object axis at early time points and rotates to optimize face discrimination at later time points. This adaptive axis rotation is a structured and sophisticated phenomenon, suggesting a dynamic mechanism that aligns with higher-level functional goals. This form of computation, where tuning axes dynamically shift in response to stimulus features, has not been alluded to in prior studies and requires a more complex underlying circuit motif than simple lateral inhibition.

Finally, our work challenges the prevailing view that feedforward deep networks—widely regarded as the best models of IT cortex—are sufficient to explain core object recognition in IT. Until now, the relationship between observed IT dynamics and deep network-based models of IT responses has remained unclear. We bridge this gap by analyzing tuning changes using tools derived from deep networks. Our findings show that even simple, core object recognition of clear, unambiguous faces invokes recurrent dynamics, suggesting a novel computational framework for IT representation.

While our RNN model can capture part of the observed dynamics through lateral inhibition, this mechanism alone cannot fully explain our findings. For example, lateral inhibition does not account for the axis inversion in low-dimensional space while simultaneously supporting the emergence of new tuning dimensions in higher-dimensional space. Moreover, the observed stimulus dependency suggests the presence of additional normative principles governing these computations. The mechanisms underlying these dynamics remain an open question and point to exciting future directions in modeling and experimentation.

On a personal note, while our lab has worked in this field for over two decades, we were genuinely surprised by the findings of this study. The structured and dynamic nature of axis changes represents a fundamental departure from the static representations predicted by feedforward models and suggests that IT cortex employs a sophisticated and dynamic form of computation that remains to be fully understood.

Finally, for comparison we note recent exciting work by Aronov and colleagues on discovery of a “bar code” in chickadees for representing cache locations in a sparse way (Chettih et al. Cell 2024, Fang et al. bioRxiv 2024). The fundamental computational motif

for these bar codes is also simple lateral inhibition. Thus even though lateral inhibition is a well-known phenomenon, it can serve computational functions that were previously unknown.

We have added the following sentences to the discussion:

"We believe these findings reveal a new form of neural computation, demonstrating that the tuning of cells in macaque face patches is not fixed but can change over an extremely short timespan (within just 20 ms, cf. Fig. 3b-h). Crucially, we are not merely observing different latencies for different feature representations, which would be expected given the multiple synaptic pathways through which visual input can reach IT cortex. Rather, we have identified a wholesale switch in neural code from object axes mediating face detection to face axes mediating face discrimination. This switch is stimulus-gated and hence dynamically controllable. When the stimulus is a non-face object, this transformation never occurs—cells throughout the face patch population persist in using object axes for the entire stimulus duration."

"Our results challenge a prevailing view that "core object recognition"—the ability to rapidly recognize objects within 200 ms despite substantial variation in appearance—can be explained primarily by feedforward processes³⁷. They also call into question the dominant view that deep networks, widely considered the best models of IT cortex, are sufficient to capture core object recognition process in IT. The recognition of clear, isolated faces in the absence of any background clutter is a textbook example of core object recognition. Yet, we find that even this process is consistently accompanied by a rapid switch in neural code within the key brain structures mediating face recognition³⁰. We speculate that the dynamic switch in encoding axis is mediated by the rich local and long-range recurrent connections within IT cortex^{5,51}. Supporting this idea, our RNN model showed that a simple form of lateral inhibition can lead to axis reversal (Extended Data Fig. 21). While lateral inhibition was one of the earliest circuit motifs to be identified^{39,52}, we are only beginning to understand its full range of computational functions⁵³."

(12) It is unclear how the conclusions in this study connect to the essentially opposite conclusions reached by Vinken et al 2023.

Vinken et al. (2023) relied exclusively on time-averaged neuronal responses, which inherently precludes their ability to uncover detailed temporal dynamics. While their conclusion (namely, that face cells used the same axis for encoding both faces and objects) differs from ours, they appear to lack sufficient supporting evidence, even within their time-averaged framework, as explained below.

First, we note that the core observation in their study aligns with ours: encoding models trained on faces or objects fail to cross-predict effectively. They propose two potential explanations for this: (1) overfitting or out-of-distribution performance drops (i.e., ground truth = one model for both faces and objects), and (2) the presence of unique, face-specific features (i.e., ground truth = distinct models for faces and objects, as we find). As they note: *"Does this mean that face-to-face response variability is partially determined by face-specific features? This could be the case. However, an alternative explanation is that the DNN encoding models overfitted on the stimulus domain used for mapping the model to neural data."*

However, Vinken et al. dismiss the face-specific feature hypothesis based on a single experiment, where they examined cross-predictions between human and monkey faces rather than faces and objects. They reported lower cross-prediction performance between human and monkey faces compared to within-category predictions. From this, they argue that since human and monkey faces share many features, the cross-prediction gap must result from overfitting rather than face-specific neural coding.

We find this reasoning flawed. First, human and monkey faces have numerous distinctive features, making it entirely plausible that these differences contribute to the observed

cross-prediction gap. Second, even if the observed gap between human and monkey faces were due to out-of-distribution effects, this conclusion does not logically extend to faces and objects.

In lieu of such indirect and inconclusive reasoning, we directly examine face and object tuning in a time-resolved manner, and provide concrete, evidence-based support for the existence of face-specific neural coding. Our approach leverages the temporal dynamics of neural responses, offering a more direct and robust investigation into the mechanisms underlying face selectivity.

We have rephrased our explanations in the manuscript:

“A recent study examining face and object tuning within face cells²² came to a very different conclusion—that these computations are domain general—but crucially, this study did not analyze face and object encoding in a time-resolved manner.”

(13) The main figures show results from only one monkey, which deviates from normal practice in the field. It is important to show that the results are qualitatively reproducible across monkeys. Across-monkey reproductions are conventionally shown in the main figures, not the supplementary materials. The results in Extended Data Fig. 2e arguably differ from the main conclusion that faces switch to and stay with a category-specific code, a conclusion that affects some of the interpretations (e.g., the “second major conceptual advance” regarding behavioral results on lines 525-537). Moreover, the example monkey uses a block design, which, as the authors point out, engages well-known adaptation effects. Incidentally, the interleaved experiment incompletely controls for adaptation effects: On average half of the stimuli would still be faces, or 1.5 faces per second at the study’s presentation rate (150/150 ms on/off), while adaptation effects last more than a few seconds. I am not sure adaptation can fully explain the study’s results, but I don’t believe the current experiments rule out contributions from adaptation.

We thank the reviewer for raising this important point. The reason we initially placed the data from the second monkey in the supplementary figures is that most of our figure panels are not easily combinable like line plots (which can accommodate results from multiple monkeys as overlaid traces), but rather full matrix-style visualizations and standalone population summaries. Including both animals in each main figure would lead to redundancy and clutter, potentially obscuring the core message.

As an alternative, we considered pooling units across animals into a single analysis. We have taken this approach in a previous study (She et al., Nature 2024), which faced similar concerns. However, for the current work, which focuses on local population dynamics within individual face patches, we opted against pooling across monkeys out of concern that combining units from different animals might blur differences in intrinsic recurrent dynamics in each monkey and face patch. That said, we have generated pooled-unit versions of the key figures (**Reviewer Fig. 21–23**) and would be happy to include them in the main manuscript if the reviewer feels this alternative presentation would be more appropriate. We also performed new experiments in face patch AM with an additional monkey (Monkey M), and the results are now presented in the supplementary materials

(**Extended Data Fig. 7, 14**). Regarding the reviewer's observation that the face–object axis correlation matrix for monkey J (**Extended Data Fig. 2e**) shows weaker negative correlation compared to monkey A, this is indeed correct. However, we emphasize that this matrix reflects a population average and can vary depending on the specific subpopulation of neurons recorded in each session. To further clarify, we assessed axis reversal at the single-cell level by quantifying the extent to which each neuron's face axis diverged from its object axis during the late response period. We found that 83% of cells in monkey A and 56% of cells in monkey J exhibited clear axis reversal, confirming that this is a robust, population-level phenomenon observed in the majority of face-selective neurons. Example single-cell axis similarity matrices from both animals are shown in **Reviewer Fig. 24a–f** (top two cells per monkey show axis reversal; bottom cell shows non-reversing cells). These findings support the conclusion that, despite noisier average patterns in monkey J, the core effect of axis reversal remains strong and widespread. This conclusion is also supported by **Extended Data Fig. 2f–j** of the manuscript. Finally, as noted in the main text (**Fig. 3e**), the degree of axis reversal correlates with face selectivity: neurons with higher face-selective d' values tend to exhibit stronger divergence between their face and object axes. We confirm and quantify this relationship in **Reviewer Fig. 24g–h**. This link helps explain the observed variability across sessions and further reinforces the interpretation that axis change is a defining feature of face-selective neurons. We have now included **Reviewer Fig. 24** in the manuscript.

The reviewer also raises the possibility that the axis change phenomenon specific to faces could be influenced by the higher presentation frequency of faces compared to other object subcategories. To address this concern, we conducted a new experiment with a variant stimulus set that included 2,500 objects and 500 faces, with all images randomly interleaved during the stimulus display. Under this design, the average interval between two face presentations was extended to 1.8 seconds (150/150 ms on/off, compared to 0.6 seconds with the main stimulus set). We presented this stimulus set while recording from monkey M in face patch AM. The results replicated all key findings from the main session (**Reviewer Fig. 25f–i**): we observed the axis change phenomenon for faces, consistent with our original conclusions. As a control, we also recorded from the same face patch in the same animal using our standard stimulus set (1,392 objects and 1,050 faces) and observed no qualitative differences between the two conditions (**Reviewer Fig. 25a–d**). These results confirm that the face-specific axis change is not driven by face presentation frequency.

Reviewer Figure 21. Fig. 2 with units pooled across both monkeys.

Reviewer Figure 22. Fig. 3 with units pooled across both monkeys.

Reviewer Figure 23. Fig. 4 with units pooled across both monkeys.

Reviewer Figure 24 (Replicated as Extended Fig. 9). Axis switch in single face cells. (a–c) Cosine similarity matrices for three example cells from Monkey A (ML) comparing (object, object), (face, face), and (face, object) axes across pairs of latencies (computed using all 60 dimensions). Top two cells exhibit axis reversal; the bottom cell does not. (d–f) Same as (a–c) for three example cells from Monkey J (ML). (g) Histogram of peak d' values for cells from the two sessions. Gray: Monkey A; Blue: Monkey J. (h) Scatter plot showing the relationship between each cell's peak d' and its axis switch strength in Monkey A (ML).

Reviewer Figure 25 (Replicated as Extended Fig. 19). Adaptation to face images is not the reason for axis change. **(a)** For original face-object ratio session, monkey M, AM: Distribution of object (left, green) and face (right, purple) axes, projected onto the top two dimensions of the 60-d object space. **(b)** Positions of faces and objects in the first to PCs of the feature space. **(c)** From left to right: matrices of mean cosine similarities across the population between (object, object), (face, face), and (face, object) axes (computed using all 60 dimensions) for different pairs of latencies. **(d)** Cosine similarity between the overall object axis for each cell (computed using a time window of 50–220 ms) and its time-varying face axes, sorted from top to bottom according to face selectivity d' (left). **(e)** Mean response time course to each face and object stimulus, averaged across cells and trials. **(f–j)** Same as (a–e) for adaptation-testing session, monkey M, AM.

Minor concerns:

(14) The terms “Rapid” and “concerted” are frequently used as part of the conclusions, but neither is really shown except qualitatively. Going just by examples, one could point to cell 8 in Fig. 3g, which does not switch in the same 20-ms window as the other cells. It will be useful to more precisely define and quantify what rapid and concerted mean.

To quantify what we mean by “rapid” and “concerted,” we plotted the distribution of face axis switch times across the population, defined as the time point when the face–object axis correlation drops below half of its peak value for each unit. The distribution was tightly clustered, with a standard deviation of 12.21 ms, indicating concerted timing across neurons. The mean switch time was 95.16 ms—closely aligned with the average response latency across units, which was 76.9 ms (calculated as the half-peak time of each unit’s response; vertical black dashed line in **Reviewer Fig. 26**). Together, these

findings demonstrate that the face axis change occurs in a temporally precise and rapid manner across the population.

Reviewer Figure 26 (replicated as Extended Data Fig. 11). Axis change time of the population is concerted.

(15) How does prosopagnosia indicate a face detection gate (lines 46–50)? The same evidence is consistent with face-specific regions always (i.e., in an ungated way) doing detailed feature processing.

We acknowledge the reviewer’s point that prosopagnosia could be consistent with face-specific regions always engaging in detailed feature processing, rather than indicating a face detection gate. We referenced prosopagnosia in our brief review of evidence for specialized mechanisms because, historically it has been a key piece of evidence raised to support the existence of specialized face areas. However, we agree that holistic processing and microstimulation results are more directly relevant to our claims.

To address this, we have deleted the part mentioning prosopagnosia:

~~"This idea originated from the behavioral observation that humans are much better at discriminating face parts presented in the context of a whole face (Fig. 1a, top right inset), as well as the neuropsychological observation of "prosopagnosia," a highly specific deficit in face recognition that can be caused by a focal lesion of the temporal lobe^{26,27}."~~

This addition ensures that the interpretation of prosopagnosia aligns with the broader context of our discussion on face-specific mechanisms.

(16) On lines 61–64, it is relevant to mention that general-purpose deep nets are better models of “face” neuron responses than face-only deep nets (e.g., Vinken et al 2023).

We thank the reviewer for this excellent suggestion. We have added the following to manuscript:

"Somewhat surprisingly, these general-purpose neural networks outperform models trained exclusively on face identities in predicting face cell responses. This suggests that face encoding may be governed by general computational principles rather than specialized, face-specific mechanisms^{22,30–32}."

Referee #3 (Remarks to the Author):

In this manuscript, the authors investigate the important question of whether face-selective cells use general (not specific to faces) versus face-specific mechanisms.

The research question is highly relevant and of significant interest to the community, especially given the renewed interest in whether face cells use a domain-general or domain-specific code (Vinken et al., 2023). The results support the conclusion that face cells exhibit a dynamic code that changes from early to late components. Overall, these findings are highly relevant and supported by a large number of data analyses. Moreover, several control analyses were performed to rule out possible alternative explanations. Methodologically, the manuscript employs state-of-the-art approaches. While the writing is clear, some conclusions appear insufficiently supported by the analyses performed. I concur with the novelty of revealing a dynamic coding for face cells. However, as a reader, I often found myself with lingering questions that, if addressed, could significantly enhance its potential impact. Below, I discuss these points.

1) Feature dimensions underlying early and late components:

The manuscript raises a crucial question regarding the underlying feature dimensions for the initial domain-general and subsequent domain-specific face code proposed by the authors. The absence of specific analyses addressing this question leaves a gap in understanding.

I give you some examples.

-You propose that face-selective cells are initially agnostic to the category of face. This conclusion suggests that the type of coding allowing discrimination between faces and objects is not based on the category of an object. However, this conclusion is not supported by specific analyses. What is the feature dimension underlying the single-axis encoding? In Extended Data Fig. 9b, examples of face and object images projecting maximally onto the extremes of the first two components show an interesting animacy distinction: high-projection objects are all animate, and low-projection objects are all inanimate. If animacy (in line with Bao et al., 2020) is a relevant axis in the object space for initial discrimination between faces and objects, can a cell encoding animacy be considered agnostic to the category of faces? First, it needs to understand that something is a face before concluding that it is animate. In other words, the examples shown in Extended Data Fig. 9b do not seem to fully support the idea of agnostic cells. It would be interesting to see examples of face and object images projecting maximally onto the extremes of the first two components taken at different time windows as shown in Figure 3f. Such visualization could offer valuable insights into the dimensions underlying the domain-general code of face cells as proposed by the authors.

We apologize for the confusion. In the paper, we stated: *“Intuitively, the preferred axis is the direction of the response gradient of a cell (i.e., the vector pointing from stimuli in the object space that elicit weak responses to stimuli that elicit strong responses). In this*

framework, face cells—just like other IT cells—encode objects by projecting incoming stimuli onto their preferred axis agnostic to their category^{6,22}.” What we meant by this is that the cells use the same axis to project all images, regardless of their category. However, this axis itself has a direction within the feature space and thus reflects the cell’s feature selectivity. As a result, a cell can still distinguish faces from objects using this initial axis, which is absolutely category selective—even though it is not yet specialized for detailed face discrimination. We further explain that a cell’s preferred axis dictates its category selectivity under the domain-general framework in the manuscript:

“Consistent with the idea that IT computations are domain general, IT cortex has been shown to contain a topographic map of object space, defined by the top two principal components of the DNN feature representation^{6,32} (Fig. 1c). Within this map, face patches occupy one quadrant, while patches selective for other object categories span the remaining quadrants. This largescale organization further suggests that face cells are not special, but together with other IT cells, are representing a shared object space.”

The reviewer’s suggestion to examine the images projecting maximally onto the extremes of the first two components at different time windows is an excellent one. To address this, we performed PCA on the population responses to both face and object stimuli, then computed the preferred face and object axes for the first two principal components. We next visualized the images that projected maximally onto the extremes of these axes at early and late time windows (**Reviewer Fig. 27**).

Our results show that at early latencies (e.g., 100–120 ms), both the face and object axes for the first two PCs point toward face-like features—indicating that early population coding emphasizes properties that distinguish faces from objects. At later latencies (e.g., 140–160 ms), the images projecting maximally onto the object axes remain largely unchanged. However, the face axes exhibit a striking reversal: the new preferred directions point away from canonical faces and toward more stubby, non-animate object-like stimuli, while faces now project weakly onto these axes.

These findings are consistent with the conclusions from the manuscript and support the idea that face-selective cells initially use a domain-general axis optimized for face detection, i.e., to distinguish faces from objects. Over time, the tuning reverses to enable domain-specific processing, supporting fine-grained identity discrimination. Importantly, this dynamic shift occurs only for face axes, while object axes remain stable throughout, underscoring the specificity of this temporal reconfiguration for face processing.

We have added this paragraph to the manuscript:

“We further visualized the images that project maximally and minimally onto the face and object axes of the first two response principal components (PCs) at different time points (Extended Data Fig. 13, see Methods). At early latencies (e.g., 60–80 ms), both the face and object axes pointed toward canonical faces, suggesting that early population coding emphasizes features distinguishing faces from objects. By later latencies (e.g., 140–160 ms), the object axes remained stable, but the face axes reversed direction—now pointing toward more stubby, non-animate, object-like stimuli. Canonical faces, by contrast, projected weakly onto these late face axes.”

a Response PC1

Object axis

Face axis

Reviewer Figure 27 (replicated as Extended Data Fig. 13). Maximally projecting images onto the object and face axis of the first two response components at different time windows. **(a)** Images projecting maximally onto the extremes of the object (top) and face (bottom) axis of the first response component of ML at different time windows. **(b)** Same as (a) but for the second response component of ML.

-You conclude that 'Newly emergent face encoding axes improve face discrimination'. [line 455: 'Overall, these results show that the drastic changes in feature tuning occurring at ~100 ms have a functional consequence, markedly improving the ability of the neural population to perform fine face discrimination']. To conclude this, one needs to show that late components [120-140ms] better discriminate face identities relative to early components [80-100ms]. The analyses performed (Fig 5) to support this statement do not directly test this question. For instance, Figure 5b demonstrates that while long latencies appear to yield slightly better reconstructions, short latencies also produce very good results. Further, combining short and long latencies gives the best possible reconstruction. How does this fit with the idea that at short latencies face cells are agnostic to faces and encode feature dimensions common to faces and objects? How do these results support the complete change of neural code proposed by the authors? Wouldn't

you expect a substantial difference between early and late latencies in their face reconstruction ability? Also, here you use very short latencies [50-75ms] but from your previous analyses, it seems that the object and face axes align at 80-100ms.

First, we would like to clarify a subtle but important point regarding the reviewer's interpretation. The reviewer is correct in summarizing our core idea as: "at short latencies, face cells are agnostic to faces and encode feature dimensions common to faces and objects." However, we wish to clarify what we mean by "agnostic." Specifically, we do *not* mean that face cells ignore category altogether at early latencies, but rather that the same encoding axis is used for both face and object stimuli. This axis is still meaningful and informative—it enables discrimination between faces and objects and, to some extent, between individual faces, since different faces still project differently along it. Face detection is a non-trivial process, and this early, shared object axis captures critical features for identifying faces, as illustrated by the maximally projected images described earlier. Therefore, we do expect some degree of face discrimination even at short latencies.

Regarding the analyses in Fig. 5, we agree that a more direct quantification of face discrimination at early versus late latencies would help strengthen our claim. To address this, we tracked face categorization (face vs. object) and face discrimination (between identities) using 20-ms sliding windows (**Reviewer Fig. 28a**). This analysis revealed two key findings: (1) Face discrimination accuracy rises significantly later than face categorization accuracy. (2) The rise in discrimination accuracy occurs around 100 ms, coinciding with the observed axis change. These results provide direct support for our interpretation that the newly emergent face axes enhance fine face discrimination. Specifically, while the initial object axis supports both detection and coarse discrimination, the emergence of dedicated face axes at later latencies enables improved identity-level discrimination—a shift from a domain-general to a domain-specific coding regime.

However, one might still question whether the observed increase in face discrimination accuracy is due to a *refinement of tuning in low-dimensional features*, rather than the *emergence of new tuning in higher dimensions*. To directly test this, we designed a new experiment aimed at "*knocking out*" the contribution of low-dimensional features and assessing whether the performance gain at later latencies still persists. We generated 2,000 synthetic faces using the face model from Chang et al. (2016), which defines a 50-dimensional face feature space (comprising 25 shape and 25 appearance dimensions). In this face set, we strongly constrained variance in the first 5 shape and 5 appearance dimensions, limiting them to 5 discrete values (400 faces per position), while allowing higher-dimensional features to vary freely (**Reviewer Fig. 28b**). If face discrimination were driven mainly by low-dimensional features, such a manipulation would significantly impair neural discrimination. However, we observed the opposite: discrimination performance still increased at later latencies, despite low-dimensional features being tightly controlled. This provides strong evidence that higher-dimensional features play a critical role in enabling fine face discrimination during later stages (**Reviewer Fig. 28c**).

Finally, regarding the choice of time windows, we want to clarify that we never use the [50–75 ms] window in isolation. Our short-latency window is a concatenation of [50–75 ms] and [75–100 ms], which effectively covers [50–100 ms]. The reason we split the window is to allow fair comparison with the short+long combined window, which by design spans two separate time intervals. Using a [50–100 ms] window directly would result in half the number of training samples (or free parameters) compared to the combined window, due to the way linear regression is fit. This imbalance would bias results in favor of the combined window. By splitting the short and long windows into matched sub-windows, we ensure that comparisons across conditions are methodologically balanced and statistically fair.

Reviewer Figure 28 (Replicated as Extended Data Fig. 25). Decoding time course of synthetic faces. (a) Time course of face categorization (face vs. object) and face identity discrimination, computed using 20-ms sliding population responses from all ML cells to realistic faces. (b) Schematic of synthetic face generation. The first 5 shape PCs and first 5 appearance PCs were restricted to 5 fixed values each (in a 10D subspace), while higher-dimensional features were

sampled randomly. Thus, variations within each triad of faces arise solely from higher-order features. (c) Time course for categorization and identification using the low-dimension–restricted synthetic face set. The orange line shows identity decoding using all synthetic faces. Gray dashed lines represent decoding for subsets of faces grouped by each of the five lower-dimensional positions.

We have added these paragraphs to the manuscript:

"We also analyzed face category and identity discrimination performance directly from the population responses using a 20-ms sliding window (see Methods), without relying on reconstruction. This analysis confirmed that face discrimination accuracy increased at later time points, peaking after the rise in categorization accuracy, consistent with a temporal shift from coarse to fine coding^{3,15} (Extended Data Fig. 24c)."

"In the previous section, we demonstrated emergence of new tuning in higher dimensions at long latency, and here, we demonstrated a corresponding increase in face discrimination accuracy at the same time. Are the two directly related, or is the latter solely caused by tuning changes in low dimensions? To test this, we designed an experiment to "knock out" the contribution of low-dimensional features and assess whether the performance gain at later latencies still persists. We created synthetic faces whose variation along low-level features was tightly constrained (see Methods). Despite this constraint, discrimination performance still improved at later time points, indicating that the late-phase enhancement is not solely dependent on low-dimensional features. This supports the idea that the observed improvement is, at least in part, driven by new tuning to higher-dimensional features essential for fine face discrimination (Extended Data Fig. 25)."

And also mentioned it in the discussion:

"Tsao et al. came to a similar finding, comparing time courses for decoding face/object category versus individual face identity in face patch ML15. We have also confirmed that face categorization peaks earlier than face discrimination using the present dataset (Extended Fig. 24c)."

Related to this you write: [line 333: 'Each of the four cells showed new tuning to different features, e.g., $v\vec{v} \perp _$ for Cell 1 varied from small inter-eye distance and sharp chin to large inter-eye distance and round chin']. This statement lacks support from specific analyses. Upon reviewing these images, the primary distinction I observe is the variation in luminance, transitioning from bright to dark from left to right. How does this emphasis on a low-level feature align with the suggestion that the face code during late latencies is specific to face identity?

We thank the reviewer for their careful observation. It is true that for three of the example cells, the primary orthogonal axis showed a gradient in complexion. However, this is not inconsistent with our conclusions. Complexion is an important factor for discriminating between faces and is a feature used more for within-category (face identity) distinctions than for differentiating faces from objects. Thus, it makes sense for complexion to emerge as a new tuning direction during later latencies. Additionally, we emphasize that there are many detailed features beyond complexion that also contribute to the new tuning directions. For example, as we noted, features like inter-eye distance and chin shape also change along these new tuning axes. These subtle variations, along with complexion, collectively determine the new tuning directions of the cells and highlight the transition from general face detection to specific face identity coding during late latencies.

Overall, throughout the manuscript, I was expecting analyses to try to understand what the underlying dimensions of the domain-general and domain-specific code are. In my view, such an investigation would increase the potential impact of this work.

We appreciate the reviewer's thoughtful comments regarding the importance of understanding the underlying dimensions of the domain-general and domain-specific

codes, and we agree that such analyses enhance the impact of the work. In response, we pursued two complementary approaches to address this question.

First, we demonstrated that the newly emergent high-dimensional features are functionally meaningful, as they contribute to improved face discrimination accuracy (**Reviewer Fig. 28a**). To further validate this, we conducted a new experiment in which we strongly constrained low-dimensional feature variance while allowing only higher-dimensional features to vary. Even under these conditions, we observed a significant increase in face discrimination performance at later latencies (**Reviewer Fig. 28b, c**), directly confirming that high-dimensional features play a critical role in fine face discrimination.

Second, we aimed to directly visualize the emergent tuning dimensions, providing an intuitive view of what those higher dimensions represent. In **Fig. 4h–k**, we synthesized faces along the new tuning directions that emerge at later time windows. These feature gradients clearly illustrate examples of finer-scale facial features—such as inter-eye distance, chin shape, and complexion—that are selectively encoded in the higher dimensions.

Together, these analyses show not only that higher-dimensional tuning arises over time, but also that it is functionally relevant and interpretable, offering insight into how the face-selective population transitions from domain-general to domain-specific encoding.

2) Comparisons between AM and ML.

The manuscript's observation regarding the largely similar pattern of dynamics in face patch AM and ML, albeit with longer latency in AM, raises intriguing questions about potential differences between these regions. Does it mean that AM represents faces in the same way as ML but just slightly later? According to these results, AM appears to be a sort of copy of ML. This is rather unexpected and in contrast with previous findings from the same lab (Tsao et al., 2010). It would be extremely interesting to further investigate potential differences between AM and ML representations. By inspecting Extended Data Fig. 11, as the authors point out, there seem to be already some interesting differences between ML and AM (line 262: 'There was also a weak, brief second period of realignment at ~160 ms not observed in ML'). I was expecting more analyses in this direction given that data has been acquired from both areas.

We thank the reviewer for raising this excellent point. A more detailed comparison between face patches AM and ML is indeed valuable and provides a deeper understanding of the dynamics within the face-processing system.

Initially, we were also surprised to find that AM exhibits axis change behavior, implying that AM goes through a face detection stage similar to ML. However, this redundancy could be advantageous — each patch may independently apply its own face detection gate to ensure that only valid faces are further processed. Such redundancy would enhance robustness. Moreover, anatomically, the face patch system is not a strictly

feedforward hierarchy: AM receives substantial direct input from PL, bypassing ML entirely (Grimaldi & Tsao, Neuron 2016).

As noted, one key difference is the re-alignment period observed in AM's face axis dynamics (**Reviewer Fig. 29d**), which is not present in ML (**Reviewer Fig. 29c**). To investigate the origin of this difference, we examined the time course of the raw population responses for both ML and AM (**Reviewer Fig. 29a, b, e**). In AM, the second increase in axis alignment (**Reviewer Fig. 29g**) coincides with a second wave of population response. We also recorded from AM in a third monkey (Monkey M), and observed the same phenomenon: a second peak in the population response accompanied by a re-alignment of the face axis (**Reviewer Fig. 29f, h**). This replication suggests that the effect is a robust and characteristic feature of AM.

Beyond this re-alignment effect, we identified additional differences in terms of shape vs. appearance tuning: Prior work from our lab has suggested that ML neurons are more tuned to shape, whereas AM neurons are more tuned to appearance (Chang et al., 2017). To test this in our data, we transformed the AlexNet-based feature space into the shape-appearance space defined in Chang et al., 2017 and performed axis mapping by regressing neural responses onto these new features. We then decomposed each cell's face axis into shape and appearance components and defined a shape preference index as the relative magnitude of the shape vs. appearance axis. Using this index, we confirmed our lab's previous finding that ML neurons show stronger shape tuning than AM neurons when using time-averaged responses (**Reviewer Fig. 29i**).

It is important to note that in the present study, we only presented frontal-view faces, limiting the extent to which view-dependent differences between ML and AM can be assessed. Other than these, ML and AM appear to show qualitatively similar dynamics in the current dataset.

Together, these analyses highlight important functional distinctions between ML and AM, even though both areas exhibit the axis change phenomenon. They support a model in which both regions contribute to dynamic, fine-grained face encoding, but do so along distinct representational axes aligned with their unique roles in the face-processing hierarchy.

Reviewer Figure 29. Comparison between ML and AM. **(a)** Mean response time course to each face and object stimulus in ML in monkey A, averaged across cells. **(b)** Same as (a) for AM in monkey A. **(c)** Cosine similarity between the overall object axis for each cell (computed using a time window of 50-220 ms) and its time-varying face axes, sorted from top to bottom according to face selectivity d' (left). **(d)** Same as (c) for AM in monkey A. **(e)** Cell-averaged response of ML and AM across all face or object stimuli in monkey A. **(f)** Cell-averaged response of AM across all face or object stimuli in monkey M. **(g)** Correlation between face and object axes for ML and AM as a function of time in monkey A. **(h)** Correlation between face and object axes for AM as a function of time in monkey M. **(i)** Shape preference indices for ML and AM of monkey A of time-averaged responses.

3) AlexNet trained on ImageNet

The authors used AlexNet trained on ImageNet to generate a 60-dimensional space capturing features of both faces and objects (or only faces for the second part investigating domain-specific coding) to compute the object and the face axes. Thus, what comes after is constrained by the assumption that AlexNet is a good model to capture the object and face space in primates.

While I acknowledge that CNNs are currently considered the best models for primate perception, there is also ample evidence highlighting their limitations. One of the most striking differences between biological vision and convolutional neural networks (CNNs) is their bias toward image texture over object shape (Geirhos et al., 2018). Moreover, a CNN trained on ImageNet does not inherently learn face-specific features since faces are not included in the training set. This aspect becomes particularly relevant in the second part of the analyses where the authors test the domain-specific code for faces.

Thus, overall, the use of AlexNet trained on ImageNet raises concerns regarding its ability to capture face-specific features. Given these limitations, especially concerning face-specific features, utilizing a face identity-trained network such as VGG-face could offer more relevant insights, particularly in the context of the domain-specific code for faces.

We thank the reviewer for raising this important point and for suggesting the use of a face-identity-trained network such as VGG-Face. While AlexNet trained on ImageNet has been widely used and is currently considered one of the best available models for capturing object and face representations in primates, we acknowledge its limitations, particularly concerning its texture bias and lack of explicit face-specific training.

To address this concern, we implemented a ResNet-50 model trained on the VGGFace2 dataset (Cao et al, 2018. arXiv) model as suggested by the reviewer. Using features extracted from the last fully connected layer of VGG-Face, we repeated the analyses presented in Figure 4. Importantly, we were able to replicate all the key conclusions using this alternative model. These findings indicate that the observed neural dynamics are robust and not specific to the choice of model, reinforcing the validity of our conclusions. We appreciate the reviewer's suggestion, which further strengthens the generalizability of our results.

Reviewer Figure 30 (Replicated as Extended Data Fig. 22). Axis similarity matrix of VGG face feature encoding models. **(a)** Matrix of face axis weights in VGG-FACE face space for each cell, computed using a short (80-100 ms, top), long-early (120-140 ms, middle), and long-late (160-180 ms, bottom) latency window. **(b)** Matrix of object axis weights computed at three different latencies; conventions same as (a). **(c)** Purple (green) distribution shows cosine similarities across units between face (object) axis weights at short and long-early latency. All dimensions (1-60) were used to compute cosine similarities shown here. **(d)** Same as (c), using higher face space dimensions 6-60 to compute cosine similarities for each cell. **(e)** Same as (c), showing cosine similarities across units between face (object) axis weights at long-early and long-late latencies (120-140 ms and 160-180 ms), using all 60 dimensions. **(f)** Explained variance in z-

scored population responses to faces at short (80-100 ms, orange) and long (120-140 ms, blue) latencies, as a function of number of PCs. Shaded area indicates shuffling controls.

We have added the following description in the text:

"To test whether our findings depended on the choice of feature space, we repeated the analyses using a ResNet-50 model trained on the VGGFace2 dataset⁴¹—a deep network specifically optimized for face identification (see Methods). We successfully replicated all key results, confirming that the observed axis dynamics are robust across different feature representations (Extended Fig. 22)."

4) Some conclusions are not supported by sufficient evidence.

In my opinion, it might be beneficial to reconsider some of the conclusions that seem to surpass the support provided by the analyses. For instance, the sentence in the abstract, 'new tuning developed to multiple higher feature space dimensions supporting fine face discrimination', appears to lack sufficient evidence from my perspective. To strengthen this conclusion, it may be advisable to conduct a direct test of face identity discrimination at both early and late latencies, and possibly include a comparison between ML and AM.

We thank the reviewer for raising this important concern. We agree that directly demonstrating identity discrimination at early versus late latencies would provide stronger support for our conclusion. As noted in our response to Question #1, we performed a sliding-window decoding analysis using 20 ms windows to measure face identity discrimination accuracy over time. We found that discrimination performance increased around the time of axis change, thereby confirming that axis change indeed facilitates improved identity-level discrimination (**Reviewer Fig. 28** above).

To further strengthen this conclusion, we also designed an additional experiment to directly test whether higher-dimensional features are responsible for the improvement. In this experiment, we highly constrained the variance of low-dimensional features in a synthetic face stimulus set and observed whether discrimination performance could still improve at later latencies. Remarkably, even when low-dimensional variance was tightly controlled, face discrimination accuracy still increased at late latencies, indicating that higher-dimensional feature space indeed contributes to fine face discrimination (**Reviewer Fig. 28** above).

Together, these analyses provide direct and convergent support for the conclusion that new tuning to multiple higher-dimensional features emerges over time, enabling more precise identity-level encoding. We have revised the abstract to reflect this point more accurately and grounded in the data.

Additional comments:

[line 322: 'Fig. 4f shows that indeed, more dimensions were required to explain 90% of response variance at long compared to short latency (96 dim for long, 82 dim for short)']. Is this difference significant?

Thank you for the question, and we apologize for the oversight. To assess the significance of this difference, we conducted a bootstrap analysis. Specifically, we randomly selected

1,000 faces from the 2,050 available face stimuli, repeated this sampling 100 times, and calculated the number of dimensions required to explain 90% of the response variance at both long (120-140 ms) and short (80-100 ms) latencies for each sample.

The results confirm that the explained response variance at long latency requires significantly more dimensions than at short latency. Paired t-tests confirmed that these differences were significant:

Monkey A, ML:

mean \pm s.d.: 78.5 ± 2.85 vs. 90.8 ± 5.28 , Paired t-test, $t(99) = -21.52$, $p < 0.05$.

Monkey A, AM:

mean \pm s.d.: 48.6 ± 2.21 vs. 52.3 ± 2.98 , Paired t-test, $t(99) = -9.69$, $p < 0.05$.

Monkey J, ML:

mean \pm s.d.: 78.8 ± 4.47 vs. 90.5 ± 4.53 , Paired t-test, $t(99) = -20.54$, $p < 0.05$.

This supports our conclusion that higher-dimensional feature spaces are increasingly important for face processing at later stages. We have updated all relevant figures and texts in the manuscript to the shuffling version.

Reviewer Figure 31 (Replicated as Fig. 4f, Extended Data Fig. 2p, Extended Data Fig. 23f). Explained variance in population responses at short (yellow) and long (blue) latencies as a function of number of PCs. More dimensions were required to explain 90% of response variance at long compared to short latency. Graded areas from random shuffling of subsets of stimuli used.

[line 116: 'Our findings challenge the notion that object recognition can be explained by largely feedforward processes, instead revealing an essential role for recurrent dynamics in core visual processing']. How does this statement relate to the fact that you created the object and face axes based on a feedforward model?

We thank the reviewer for raising this important question. A long-held belief in the field is that feedforward models, such as convolutional neural networks (CNNs), are sufficient to account for the majority of response dynamics observed in IT neurons (DiCarlo, J. J., & Cox, D. D. 2007, DiCarlo, J. J. et al 2012, Khaligh-Razavi et al. 2014, Yamins et al 2014, Cadieu et al. 2014). We do indeed create the axes based on a feedforward model, as it remains the best available model for describing IT neurons. However, our findings reveal a crucial limitation of this framework: while the initial axes describe the population's responses, the axes for faces change drastically over time. This dynamic axis change is entirely unexpected under a purely feedforward framework and underscores the essential role of recurrent dynamics in core object recognition. Our results demonstrate that even in tasks traditionally thought to involve minimal recurrence, such as core object recognition, recurrent processing is critical. This goes far beyond what a feedforward model can achieve and highlights the need for more advanced models to incorporate recurrent mechanisms to fully explain IT cortex dynamics.

Extended Data Fig. 6 (d and f) shows the distribution of face axes computed for stimulus-shuffled data. These results seem to suggest that shuffling the data does not affect the face axes' length. This is not what I would expect if these face axes project toward dimensions relevant to face identity discrimination.

We thank the reviewer for pointing out this detail. The observed lack of a clear difference in face axis length after shuffling in now **Extended Data Fig. 4 (d and f)** does require clarification. If an axis explains a very high variance of the data, shuffling would indeed result in a noticeable shortening of the axis length. However, in the case in **Extended Data Fig. 4**, the shuffling was performed on the Gaussian subset of stimuli, not the original full stimulus set. Because the Gaussian subset contains fewer stimuli, the fit quality (as measured by R2) is lower, which minimizes the difference in axis length after shuffling.

Despite this, the R2 for the Gaussian subset fit remains significantly higher than for the shuffled data (**Reviewer Fig. 32a**). Furthermore, when we analyze the axis length using the original full stimulus set, we observe a much clearer reduction in axis length after shuffling (**Reviewer Fig. 32b** for the original data and **Reviewer Fig. 32c** for the shuffled data). This demonstrates that both face and object axes become shorter after shuffling, confirming the significance of the axis mapping and its relevance to face identity discrimination.

Reviewer Figure 32. Axis length change for original mapping. **(a)** Left: Explained variance of Gaussian subset of object responses for each ML cell using the object axis (green) together with explained variance for stimulus-shuffled data (gray). Right: Explained variance of Gaussian subset of face responses for each cell using the face axis (purple) together with explained variance for stimulus-shuffled data (gray). **(b)** Distribution of object (left, green) and face (right, purple) axes, projected onto the top two dimensions of the 60-d object space. **(c)** Same as (b) but for shuffled stimulus.

Referee #1

My only final comment is that it would be nice if the authors could briefly discuss their findings reported in Nature last year (She et al Nature 2024), where they describe a rotation of the feature axis at long latencies for familiar compared to non-familiar faces, which seems quite relevant within the context of the current paper (showing a switch of axis for individual discrimination vs categorization).

We thank the reviewer for highlighting the relevance of She et al. (*Nature*, 2024). That study asked how familiarity is represented in IT and reported a late-latency rotation of the face-feature axis specifically for familiar faces, displacing those representations off the unfamiliar face manifold.

Our findings are complementary but distinct in two ways:

1. **Functional context.** She et al. examined a familiarity-dependent transformation that may facilitate memory retrieval. In contrast, our study reveals a task-general and stimulus-driven phenomenon present for unfamiliar faces (and not non-face objects) that improves individual-level discrimination while preserving category selectivity, addressing the long-standing categorization-versus-discrimination debate.
2. **Geometry of the change.** In She et al., the axis rotates off the manifold. Here, the transformation is structured: low-dimensional components reverse, while higher-dimensional components reorient, creating new tuning directions that specifically enhance identity discrimination.

We now cite She et al. (Nature 2024) in the Discussion:

“These findings demonstrate that the neural code for faces is not fixed but can change completely within just 20 ms (Fig. 3b-h), complementing a recent report of long-latency axis change triggered by stimulus familiarity⁴².”